# On the Generalization Ability of Next-Token-Prediction Pretraining

**Zhihao Li**[1]  **Xue Jiang**[2][3]  **Liyuan Liu**[1]  **Xuelin Zhang**[1]  **Hong Chen**[1][4]  **Feng Zheng**[2]

## Abstract

Large language models (LLMs) have demonstrated remarkable potential in handling natural language processing (NLP) tasks and beyond. LLMs usually can be categorized as transformer decoder-only models (DOMs), utilizing Next-Token-Prediction (NTP) as their pretraining methodology. Despite their tremendous empirical successes, the theoretical understanding of how NTP pre-training affects the model's generalization behavior is lacking. To fill this gap, we establish the fine-grained generalization analysis for NTP pre-training based on Rademacher complexity, where the dependence between tokens is also addressed. Technically, a novel decomposition of Rademacher complexity is developed to study DOMs from the representation learner and the token predictor, respectively. Furthermore, the upper bounds of covering number are established for multi-layer and multi-head transformer-decoder models under the Frobenius norm, which theoretically pioneers the incorporation of mask matrix within the self-attention mechanism. Our results reveal that the generalization ability of NTP pre-training is affected quantitively by the number of token sequences $N$, the maximum length of sequence $m$, and the count of parameters in the transformer model $\Theta$. Additionally, experiments on public datasets verify our theoretical findings. Our code is available at https://github.com/Lizeihao/MININTP.

## 1. Introduction

Large Language Models (LLMs) have emerged as powerful generative models in solving sequence-to-sequence (seq2seq) tasks (Ott et al., 2019), which not only have achieved tremendous progress in various NLP tasks (Malach, 2023), but also have realized remarkable performance in other domains (Li et al., 2024). Surprisingly, several existing LLMs, such as GPT3 (Brown et al., 2020), OPT (Zhang et al., 2022), BLOOM (Workshop et al., 2023), Llama (Touvron et al., 2023), Deepseek (Liu et al., 2024a) and Qwen (Yang et al., 2025), share two common characteristics: (i) Employing a decoder-only architecture based on the masked-self-attention(Vaswani et al., 2017). (ii) Adopting the unsupervised pre-training method of Next-Token-Prediction (NTP) (see Figure 1 (a)), which is to predict the next token based on all previous context tokens in each step (Qi et al., 2020). The predominant expense in training a large language model is typically incurred during the pre-training phase (Zhao et al., 2024). Consequently, it is very important to examine the DOMs-based NTP pre-training.

Recently, there have been increasing efforts to evaluate the DOMs-based NTP pre-training empirically. Shlegeris et al. (2022) found that language models are consistently better than humans at NTP tasks by performing two distinct experiments. Malach (2023) demonstrated when trained on Chain-of-Thought data, even a linear next-token predictor can possess high fitting ability. Bachmann & Nagarajan (2024) designed a minimal planning task and demonstrated that NTP pre-training cannot accurately predict the first position in some tasks. Li et al. (2024) utilized a single self-attention layer to explore the mechanics of NTP. While these works justify the use of NTP pre-training in the corresponding regimes, they do not provide a rigorous analysis of the training mechanism, especially from the perspective of generalization theory. This motivates a natural question:

*"Can we establish the generalization analysis of NTP pretraining, which probably explains the effects of model parameters?"*

This paper answers the above question positively. The DOMs (see Figure 1 (b)) usually consist of two components (Zhao et al., 2024): multiple layers of transformer-decoder-blocks, shortened as Representation-Learner (R-L), and a task-specific processor, noted by Token-Predictor (T-

---

[1]College of Informatics, Huazhong Agricultural University [2]Department of Computer Science and Engineering, Southern University of Science and Technology [3]Department of Computer Science, Hong Kong Baptist University [4]Engineering Research Center of Intelligent Technology for Agriculture, Ministry of Education, China. Correspondence to: Hong Chen <chenh@mail.hzau.edu.cn>.

*Proceedings of the 42$^{nd}$ International Conference on Machine Learning*, Vancouver, Canada. PMLR 267, 2025. Copyright 2025 by the author(s).

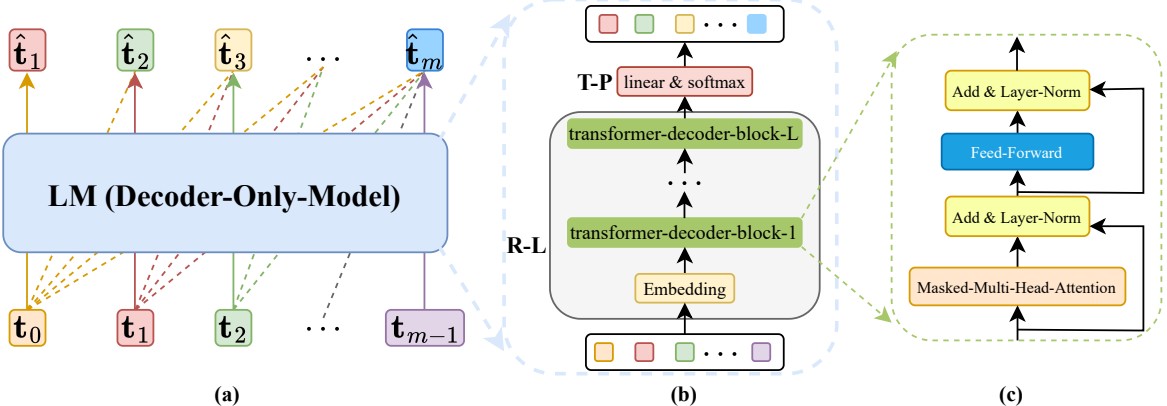

Figure 1. (a): How NTP works utilizing decoder-only model (DOM), for every input token $\mathbf{t}_j (0 \le j < m)$, we can get an output token $\hat{\mathbf{t}}_{j+1}$ whose label is $\mathbf{t}_{j+1}$, and the dashed line here represents the context used. (b): The architecture of DOM, which is consistent with the GPT-3 (Brown et al., 2020). (c): The architecture of transformer-decoder-block (Vaswani et al., 2017).

P). Similar to Zhang et al. (2024a); Deng et al. (2024), we abstract the DOMs into a token-based composite function made up of two separate functions for R-L and T-P, respectively. We consider the dependence between tokens and utilize $\varphi$-mixing to delineate the inter-token dependencies, which is a commonly used tool in non-independent scenes (Masuda, 2007; Mohri & Rostamizadeh, 2010; McDonald et al., 2015; Wong et al., 2020). We then propose a theoretical framework for NTP from the perspective of statistical learning. To bound the excess risk of NTP, we introduce the concept of Rademacher complexity for composite function classes and propose a decomposition law, as stated in Proposition 4.8. We then establish two distinct bounds on the generalization capability of NTP, depending on whether Proposition 4.8 is applied.

To further assess the impact of model parameters of DOMs on NTP, we provide a refined estimation of the covering number for multi-layer, multi-head transformer-decoder blocks. We are the first to consider the mask matrix in self-attention, which is crucial for NTP. We then use the covering number of DOMs to get the corresponding Rademacher complexity upper bound by utilizing the theory of Bartlett et al. (2017); Lin & Zhang (2019) and establish the generalization bounds for DOMs-based NTP pre-training. Our results primarily encompass three key parameters: the number of token sequences $N$, the maximum sequence length $m$, and the count of transformer model parameters $\Theta$. Our generalization bound can be expressed as $\mathcal{O}\big(\sqrt{\Theta/Nm} + \sqrt{1/m}\big)$, where $\mathcal{O}\big(\sqrt{\Theta/Nm}\big)$ signifies the generalization capability across token sequences, and $\mathcal{O}\big(\sqrt{1/m}\big)$ denotes the generalization capability among individual tokens. Our bounds remain valid even with modifications to the structure of the transformer-decoder block. Our main contributions are summarized as follows:

- *A novel Rademacher complexity decomposition method:* We consider the dependence between tokens and provide a theoretical framework for NTP pre-training (Section 3). On this basis, we establish the Rademacher complexity upper bounds of excess risk by a novel Rademacher complexity decomposition method (Section 4.1), which shows that the generalization performance of NTP pre-training is related to both sequences and tokens.

- *A refined covering number for multi-layer, multi-head transformer-decoder models:* We establish bounds for the covering number of a function space derived from a multi-layer, multi-head transformer-decoder model based on masked-self-attention (Section 4.2). Unlikely the previous works, our theoretical results are the first to consider the mask matrix in self-attention based on the metric induced by the Frobenius norm.

- *A generalization bound for DOMs-based NTP pre-training:* We use the Rademacher complexity upper bound and covering number to establish the generalization theory of DOMs-based NTP pre-training (Section 4.3). Theoretical results imply that the generalization bound mainly depends on: the number of token sequences $N$, the maximum length of the token sequence $m$, and the number of model parameters $\Theta$. Data experiments in Section 5 verify our theoretical findings.

## 2. Related Work

**Next-Token-Prediction (NTP).** Beyond its prominence in NLP (Moon et al., 2021; De Souza Pereira Moreira et al., 2021), NTP has found applications in diverse domains, including object recognition (Yue et al., 2024), sensorimotor trajectory prediction (Radosavovic et al., 2024), autonomous driving (Wu et al., 2024; Jia et al., 2024), and code-related

*Table 1.* Generalization bounds for transformer-based models in different pre-training scenarios. ($N$: the number of token sequences, $m$: the maximum sequence length, $T$: the number of prompts, $L$: the number of transformer layers, $d$: the model dimension of transformer, $\Theta \approx 12Ld^2$: the number of transformer model parameters, $C$: the constant bigger than 1).

| Ref. | Scenario | Technique | Bound |
|------|----------|-----------|-------|
| Edelman et al. (2022) | seq2seq Pretraining | Rademacher Complexity | $\mathcal{O}\left(\sqrt{\frac{C^{\mathcal{O}(L)}\ln(Nmd)}{N}}\right)$ |
| Li et al. (2023) | ICL Pretraining | Stability | $\mathcal{O}\left(\frac{C\ln N}{\sqrt{NT}}\right)$ |
| Zhang et al. (2023) | ICL Pretraining | Operator Approximation | $\mathcal{O}\left(\frac{L^2d^2\ln(1+NTC)}{\sqrt{NT}}\right)$ |
| Deng et al. (2024) | MAE Pretraining | Rademacher Complexity | $\mathcal{O}\left(\frac{L(mC)^{\mathcal{O}(L)}\ln d}{N}\right)$ |
| Ours | NTP Pretraining | Rademacher Complexity | $\mathcal{O}\left(\sqrt{\frac{\Theta\ln(1+NmC)}{Nm}}+\sqrt{\frac{C}{m}}\right)$ |

tasks (Izadi et al., 2022; Kim et al., 2021; Qi et al., 2024). While vision applications remain less explored, Kilian et al. (2024) demonstrated that NTP excels in prompt adherence and throughput efficiency for image synthesis, though diffusion models achieve superior image quality and lower latency. Extensions to standard NTP include multi-token prediction (Qi et al., 2020; Gloeckle et al., 2024) and Diffusion Forcing (Chen et al., 2024), a hybrid training paradigm combining NTP with diffusion for sequence generation.

Theoretical insights reveal NTP's foundational capabilities: Malach (2023) proved autoregressive NTP can approximate complex functions using a novel length-complexity measure. Thrampoulidis (2024) identified an implicit bias toward structured solutions in gradient-based NTP optimization at low training loss. Flemings et al. (2024) proposed PMixED, a differentially private protocol for LLM-based NTP. Madden et al. (2024) established memory capacity bounds for decoder-only transformers in NTP. Li et al. (2024) showed self-attention learns token-retrieval automata via token-priority graphs.

**Generalization theory for pre-training and transformer-based models.** Generalization characterizations of pre-training have been stated for many learning paradigms, such as curriculum learning (Zhou et al., 2022), transfer learning (Tripuraneni et al., 2020; Xu & Tewari, 2021; Lotfi et al., 2022), reinforcement learning (RL) (Ye et al., 2023; Lin et al., 2023), etc. Moreover, Zhang et al. (2024a) constructed the generalization theory for supervised pre-training and fine-tuning to explore the trade-off between intra-class and inter-class diversity in pre-training datasets. Deng et al. (2024) developed a generalization bound for the unsupervised pre-training of Masked Autoencoder (MAE), their results are mainly based on the covering number theory of the transformer-encoder models, which was established by Edelman et al. (2022).

For the generalization of transformer-based models, Deora et al. (2023) derived generalization bounds for the single-layer muti-head-attention models based on the stability of SGD Lei & Ying (2020); Zhang et al. (2024b). Recently, theoretical understandings have been provided for the generalization ability of In-context Learning (ICL). The ICL pre-training is investigated theoretically from the aspects of multi-task learning (Li et al., 2023) and Markov processes (Zhang et al., 2023), respectively. Notably, Lotfi et al. (2023) derived the first non-vacuous generalization bounds for pre-trained LLMs. Later, Lotfi et al. (2024) introduced a martingale-based bound that captures token-level dependencies.

Table 1 highlights the key differences of our theoretical result by comparing it with the most related progresses in (Edelman et al., 2022; Li et al., 2023; Zhang et al., 2023; Deng et al., 2024).

**Notation.** Throughout our paper, we denote set $\{1, \cdots, n\}$ as $[n]$. And for a matrix $\mathbf{W}$, $\|\mathbf{W}\|_{\ell_\infty} := \max_{i,j}|\mathbf{W}_{i,j}|$.

## 3. Preliminaries

This section introduces the framework of NTP pre-training and defines the architecture of DOMs.

### 3.1. Problem Setting

Consider a set of tokens $\mathcal{T}$ whose vocabulary size is $n_v = |\mathcal{T}|$. Given a pre-training dataset $D = \{\mathbf{X}_i\}_{i=1}^N \subseteq \mathcal{X}$, where $\mathbf{X}_i$ denotes the $i$-th token sequence, $\mathcal{X}$ is an instance domain such as sentences. We assume there exists an unknown distribution $\mathcal{D}$ that $\{\mathbf{X}_i\}_{i=1}^N \sim \mathcal{D}$ and all the sequences are independent of each other. Each sequence $\mathbf{X}_i$ is composed of $m$ tokens $\{\mathbf{t}_1^i, \cdots, \mathbf{t}_m^i\} \subseteq \mathcal{T}$, where $\mathbf{t}_j^i \in \mathbb{R}^{n_v}$ denotes the $j$-th token of $i$-th sequence, and $m$ denotes the maximum input length of a language model $\mathbf{LM}$. Note that the token sequences we consider here have all undergone a series of preprocessing operations, such as cropping, masking, and patching, so all sequences have the same length. We denote $\mathbf{T}_j^i = \{\mathbf{t}_0^i, \mathbf{t}_1^i, \cdots, \mathbf{t}_{j-1}^i\}$ as the context of $\mathbf{t}_j^i$, where

$\mathbf{t}_0^i = \mathbf{t}_0 \in \mathcal{T}$ denotes the given begin sign $<|\, \mathbf{im\_start}\, |>$ for all sequences. Note that $\mathbf{T}_0^i$ is the empty context for $\mathbf{t}_0^i$.

**Next-Token-Prediction.** For NTP, we abstract the model $\mathbf{LM} : \mathcal{X} \times \mathcal{T} \to \mathcal{T}$ as an algorithm that maps the context $\mathbf{T}_{j-1}$ to a function $\mathbf{LM}\,(\mathbf{T}_{j-1}, \cdot)$, similar idea can be found in Li et al. (2023). Then, we input the token $\mathbf{t}_{j-1}$ to the above function, which will get a response:

$$\hat{\mathbf{t}}_j = \mathbf{LM}\,(\mathbf{T}_{j-1}, \mathbf{t}_{j-1}),$$

we hope that $\hat{\mathbf{t}}_j$ can be as close to $\mathbf{t}_j$ as possible. More details about the next-token-prediction can be found in Figure 1. Note that the model $\mathbf{LM}$ belongs to decoder-only model (DOM), which is usually composed of a Representation-Learner (R-L) $h \in \mathcal{H} \subseteq \{\mathcal{T} \to \mathcal{I}\}$ and a Token-Predictor (T-P) $g \in \mathcal{G} \subseteq \{\mathcal{I} \to \mathcal{T}\}$, where $\mathcal{I}$ denotes a hidden representation space. We can represent the model $\mathbf{LM}$ via

$$\mathbf{LM}\,(\mathbf{T}_{j-1}, \mathbf{t}_{j-1}) = g\,(h\,(\mathbf{T}_{j-1}, \mathbf{t}_{j-1})).$$

Denote $\mathbf{z}_j^i = (\mathbf{T}_j^i, \mathbf{t}_j^i)$ as the $j$-th training sample of $i$-th sequence, where $\mathbf{T}_j^i = \{\mathbf{T}_{j-1}^i, \mathbf{t}_{j-1}^i\}$. Then the empirical risk based on NTP with the $i$-th sequence can be defined as

$$\hat{\mathcal{L}}_{\mathbf{X}_i}\,(g \circ h) := \frac{1}{m} \sum_{j \in [m]} \ell\,(g \circ h(\mathbf{z}_j^i), \mathbf{z}_j^i), \qquad (1)$$

where $\ell\big(g \circ h(\cdot),\, \cdot\big) : \mathcal{X} \times \mathcal{X} \to \mathbb{R}$ represents the pre-training loss function, usually cross-entropy loss, and

$$\ell(g \circ h(\mathbf{z}_j^i), \mathbf{z}_j^i) := \ell\Big(g\big(h(\mathbf{T}_{j-1}^i, \mathbf{t}_{j-1}^i)\big), \mathbf{t}_j^i\Big)$$

denotes the loss of sample $\mathbf{z}_j^i$. Pre-training based on NTP is to train each token sequence in the dataset $D$ according to formula (1), further obtaining the optimal R-L and T-P. We can denote the objective function based on the empirical risk minimization (ERM) as

$$\min_{g \in \mathcal{G}, h \in \mathcal{H}} \hat{\mathcal{L}}_D\,(g \circ h) := \frac{1}{N} \sum_{i \in [N]} \hat{\mathcal{L}}_{\mathbf{X}_i}\,(g \circ h). \qquad (2)$$

Let $\mathcal{L}_{\phi_i}\,(g \circ h) = \mathbb{E}\big[\hat{\mathcal{L}}_{\mathbf{X}_i}\,(g \circ h)\big]$ denote the population risk of $\hat{\mathcal{L}}_{\mathbf{X}_i}\,(g \circ h)$, and

$$\mathcal{L}_D\,(g \circ h) = \mathbb{E}\big[\hat{\mathcal{L}}_D\,(g \circ h)\big] = \frac{1}{N} \sum_{i \in [N]} \mathcal{L}_{\phi_i}\,(g \circ h)$$

be the population risk of $\hat{\mathcal{L}}_D\,(g \circ h)$. Then, the excess risk for NTP pre-training task can be represented as

$$\mathcal{E}_D\big(\hat{g}, \hat{h}\big) := \mathcal{L}_D\big(\hat{g} \circ \hat{h}\big) - \min_{g \in \mathcal{G}, h \in \mathcal{H}} \mathcal{L}_D\,(g \circ h), \qquad (3)$$

where $\hat{g} \in \mathcal{G}$ and $\hat{h} \in \mathcal{H}$ denote the optimal R-L and T-P we learned by solving (2) respectively.

## 3.2. Decoder-only Models

For a given token sequence $\mathbf{X} = [\mathbf{t}_1, \cdots, \mathbf{t}_m] \in \mathbb{R}^{m \times n_v}$, we denote $\mathbf{Z} = [\mathbf{t}_0, \mathbf{t}_1, \cdots, \mathbf{t}_{m-1}] \in \mathbb{R}^{m \times n_v}$ as the input matrix, which contains all the context information. We consider the $L$-layer and $H$-head decoder-only transformer model as the R-L, which mainly consists of one Embedding-layer and $L$ layer transformer-decoder-block (see Figure 1 (b)). We use $d$ to denote the model dimension, $d_k = d/H$ denotes the attention dimension, and $d_f = 4d$ denotes the feed-forward dimension throughout the paper.

**Embedding.** Token vectors are one-hot vectors, which are in discrete form. We need to convert the discrete vectors into continuous vectors first through the Embedding operation:

$$\mathbf{Z}^0 = \mathrm{Embedding}(\mathbf{Z}) := \mathbf{Z}\mathbf{W}_e + \mathbf{W}_p,$$

where $\mathbf{Z}^0$ denotes the embedded token sequence of $\mathbf{Z}$, $\mathbf{W}_e \in \mathbb{R}^{n_v \times d}$ denotes the token-embedding matrix, and $\mathbf{W}_p \in \mathbb{R}^{m \times d}$ denotes the position-embeding matrix.

**Transformer-decoder-block.** Let $\Pi_{\mathrm{norm}}$ denote the Layer-normlization operator, and $\sigma$ denote the non-linear activation function ReLU. Denote $\mathrm{TF}_{\mathcal{W}^l}$ as the $l$-th layer transformer-decoder-block with parameter set

$$\mathcal{W}^l = \big\{\mathbf{W}_{\mathrm{F1}}^l, \mathbf{W}_{\mathrm{F2}}^l, \{\mathbf{W}_{O_h}^l, \mathbf{W}_{Q_h}^l, \mathbf{W}_{K_h}^l, \mathbf{W}_{V_h}^l\}_{h=1}^H\big\},$$

where $\mathbf{W}_{\mathrm{F1}}^l \in \mathbb{R}^{d \times d_f}$, $\mathbf{W}_{\mathrm{F2}}^l \in \mathbb{R}^{d_f \times d}$, $\mathbf{W}_{O_h}^l \in \mathbb{R}^{d \times d}$, $\mathbf{W}_{Q_h}^l \in \mathbb{R}^{d \times d_k}$, $\mathbf{W}_{K_h}^l \in \mathbb{R}^{d \times d_k}$, $\mathbf{W}_{V_h}^l \in \mathbb{R}^{d \times d}$ and $\mathbf{Z}^l = \mathrm{TF}_{\mathcal{W}^l}\big(\mathbf{Z}^{l-1}\big)$ $(l \geq 1)$ denotes the output of $l$-th layer block, which can be formulated by

$$\mathrm{TF}_{\mathcal{W}^l}\big(\mathbf{Z}^{l-1}\big) = \Pi_{\mathrm{norm}}\left(\sigma\big(\mathbf{Y}^l \mathbf{W}_{\mathrm{F1}}^l\big)\mathbf{W}_{\mathrm{F2}}^l + \mathbf{Y}^l\right),$$
$$\mathbf{Y}^l = \Pi_{\mathrm{norm}}\left(\sum_{h \in [H]} \mathbf{A}_h^l \mathbf{W}_{O_h}^l + \mathbf{Z}^{l-1}\right). \qquad (4)$$

Here $\mathbf{A}_h^l$ denotes the masked self-attention of the $h$-th head.

**Masked Self-attention.** Denote softmax as the row-wise softmax operator, $\mathbf{Q}_h^l = \mathbf{Z}^{l-1}\mathbf{W}_{Q_h}^l$, $\mathbf{K}_h^l = \mathbf{Z}^{l-1}\mathbf{W}_{K_h}^l$, $\mathbf{V}_h^l = \mathbf{Z}^{l-1}\mathbf{W}_{V_h}^l$, we have

$$\mathbf{A}_h^l = \mathrm{softmax}\left(\frac{\mathbf{Q}_h^l (\mathbf{K}_h^l)^\top + \mathbf{M}}{\sqrt{d_k}}\right)\mathbf{V}_h^l, \qquad (5)$$

where $\mathbf{M} \in \mathbb{R}^{m \times m}$ is a mask matrix defined as

$$\mathbf{M}_{ij} = \begin{cases} 0, & j \leq i \\ -\infty, & j > i \end{cases}.$$

Then, the R-L can be mathematically formulated by

$$h(\mathbf{Z}) := TF_{\mathcal{W}^L}\Big(\ldots TF_{\mathcal{W}^1}\big(\mathrm{Embedding}(\mathbf{Z})\big)\Big). \qquad (6)$$

The T-P is composed of a linear projection and $\mathrm{softmax}$:

$$g\left(h\left(\mathbf{Z}\right)\right) = \mathrm{softmax}\left(h\left(\mathbf{Z}\right)\mathbf{W}^P\right), \mathbf{W}^P \in \mathbb{R}^{d \times n_v}. \quad (7)$$

## 4. Main Results

Note that the tokens in one sequence are dependent, which we usually call a non-i.i.d. process. We introduce the concept of $\varphi$-mixing processes to characterize the dependency relationship between tokens.

**Definition 4.1.** Let $\mathbf{T} = \{\mathbf{t}_j\}_{j=-\infty}^{\infty}$ be a stationary process (Hirschfeld, 1935). $\mathbf{T}$ is said to be exponentially $\varphi$-mixing (Dobrushin, 1956) if there exist some constants $\varphi_0 > 0$, $\varphi_1 > 0$ and $r > 0$ such that the $\varphi$-mixing coefficient

$$\varphi(k) := \sup_n \sup_{\substack{A \in \sigma_{n+k}^{\infty} \\ B \in \sigma_{-\infty}^{n}}} |\Pr[A \mid B] - \Pr[A]|$$

$$\leq \varphi_0 \exp\left(-\varphi_1 k^r\right), \forall k \in \mathbb{N}^*,$$

where $\sigma_j^i$ denotes the $\sigma$-algebra generated by the random variables $\mathbf{t}_i, \cdots, \mathbf{t}_j$.

Based on the above definition, we now make the following assumption on dataset $D$.

**Assumption 4.2.** Assume that $\mathbf{X}_i = \{\mathbf{t}_1^i, \cdots, \mathbf{t}_m^i\}$ is generated by a $\varphi$-mixing distribution $\phi_i$ for all $i$, and there exists an unknown distribution $\mathcal{U}$ such that $U = \{\phi_i\}_{i=1}^N \sim \mathcal{U}$.

*Remark* 4.3. The above assumption is widely adopted in the study of non-i.i.d. processes such as Ralaivola et al. (2010); Heinrich & Pawlas (2013); Vankadara et al. (2022); Liu et al. (2025b). In Definition 4.1, $\lim_{k \to +\infty} \varphi(k) \to 0$ means that A and B will become independent as $k$ increases. When A and B represent two different sentences, the farther the distance between A and B is (coming from two different articles), the smaller the correlation between A and B will be. Therefore, Assumption 4.2 is reasonable.

**Assumption 4.4.** There exists a constant $B_\ell \in \mathbb{R}^+$ satisfying $|\ell(\hat{\mathbf{t}}, \mathbf{t})| \leq B_\ell$ for any $\hat{\mathbf{t}}, \mathbf{t} \in \mathcal{T}$, and $\ell$ is $G_\ell$-Lipschitz w.r.t. $\hat{\mathbf{t}}$.

Assumption 4.4 is commonly used in learning theory (Bartlett & Mendelson, 2002; Shalev-Shwartz & Ben-David, 2014; Liu et al., 2022; Deng et al., 2024; Liu et al., 2025a).

**Definition 4.5** (Discrepancy measure). Given the set of distributions $U = \{\phi_i\}_{i=1}^N$, we define its discrepancy as

$$\mathrm{disc}(U) := \sup_{k \in [N]} \frac{1}{N} \sum_{i \in [N]} \|\phi_i - \phi_k\|_{\mathrm{TV}},$$

where $\|\phi_i - \phi_k\|_{\mathrm{TV}} = \sup_{\mathbf{t} \in \mathcal{T}} |\phi_i(\mathbf{t}) - \phi_k(\mathbf{t})|$ denotes total variation distance between two distributions.

As shown in (Kuznetsov & Mohri, 2020; Wang et al., 2022a), the closer the distributions in set $U$ are, the smaller $\mathrm{disc}(U)$ will be. In particular, $\mathrm{disc}(U) = 0$ when $\phi_1 = \cdots = \phi_N$.

### 4.1. Rademacher Complexity Upper Bounds

To mitigate the excess risk defined in Equation (3), we introduce a metric for assessing the complexity of a function class, known as Rademacher complexity (Mohri & Rostamizadeh, 2008).

**Definition 4.6** (Rademacher complexity). Given a sample set $S = \{z_1, ..., z_n\} \subseteq \mathcal{Z}$ and a function class $\mathcal{F} : \mathcal{Z} \to \mathbb{R}$, the empirical Rademacher complexity of $\mathcal{F}$ is defined as

$$\hat{\mathfrak{R}}_S(\mathcal{F}) := \mathbb{E}_{\boldsymbol{\varepsilon}}\left[\sup_{f \in \mathcal{F}} \frac{1}{n} \sum_{i \in [n]} \varepsilon_i f(z_i)\right], \quad (8)$$

where $\boldsymbol{\varepsilon} = \{\varepsilon_1, ..., \varepsilon_n\}$ are i.i.d. Rademacher random variables satisfying $\mathbb{P}(\varepsilon_i = 1) = \mathbb{P}(\varepsilon_i = -1) = 0.5, i \in [n]$.

Due to the dependence of tokens within a sequence, and the independence of distinct sequences, it is essential to establish two separate measures of Rademacher complexity. By setting $S = D$ and $\mathcal{F} = \ell \circ \mathcal{G} \circ \mathcal{H}$ in Equation (8), we can define the empirical Rademacher complexity of the composite function class $\ell \circ \mathcal{G} \circ \mathcal{H}$ for $D$ as

$$\hat{\mathfrak{R}}_D(\ell \circ \mathcal{G} \circ \mathcal{H}) := \mathbb{E}_{\boldsymbol{\varepsilon}}\left[\sup_{g \in \mathcal{G}, h \in \mathcal{H}} \frac{1}{N} \sum_{i \in [N]} \varepsilon_i \hat{\mathcal{L}}_{\mathbf{X}_i}(g \circ h)\right].$$

For ease of representation, we denote $\ell_j^i = \ell\left(g \circ h(\mathbf{z}_j^i), \mathbf{z}_j^i\right)$. Referring to the definition of Rademacher complexity of multi-task learning in Wang et al. (2022b), we can also consider all token sequences, and define the following multi-sequence Rademacher complexity:

$$\tilde{\mathfrak{R}}_D(\ell \circ \mathcal{G} \circ \mathcal{H}) := \mathbb{E}_{\boldsymbol{\varepsilon}}\left[\sup_{g \in \mathcal{G}, h \in \mathcal{H}} \frac{1}{Nm} \sum_{i \in [N]} \sum_{j \in [m]} \varepsilon_{ij} \ell_j^i\right].$$

*Remark* 4.7. The Rademacher complexity defined here closely resembles the "representation-induced Rademacher complexity" delineated in Deng et al. (2024). However, a notable distinction exists: the innermost function in their composite function is fixed, whereas in our definition, it encompasses the entire function class $\mathcal{H}$.

We primarily focus on the performance of R-L because it applies to various downstream scenarios after pre-training, whereas T-P changes as downstream tasks evolve. Consequently, we propose decomposing the Rademacher complexity of $\mathcal{G} \circ \mathcal{H}$ into the complexities of the individual function classes $\mathcal{G}$ and $\mathcal{H}$. This approach allows for a more precise analysis of the influence exerted by $\mathcal{H}$, defined as follows:

**Proposition 4.8** (Rademacher complexity decomposition). *Let $\mathcal{F} : \mathcal{Z} \to \mathbb{R}$ be a composite function satisfying $\mathcal{F} = \ell \circ \mathcal{G} \circ \mathcal{H}$, where $\ell$ is a loss function and $\mathcal{H}, \mathcal{G}$ are function classes. Given a sample set $S = \{z_1, ..., z_n\} \subseteq \mathcal{Z}$, for any $g \in \mathcal{G}$ satisfying $G_g$-Lipschitz w.r.t. $h \in \mathcal{H}$ and $\ell$ satisfying $G_\ell$-Lipschitz w.r.t. $g \circ h \in \mathcal{G} \circ \mathcal{H}$, we have*

$$\hat{\mathfrak{R}}_S(\ell \circ \mathcal{G} \circ \mathcal{H}) \leq G_\ell G_g \hat{\mathfrak{R}}_S(\mathcal{H}) + G_\ell \hat{\mathfrak{R}}_S(\mathcal{G} \circ \hat{h}),$$

*where $\hat{h}$ is any given function in $\mathcal{H}$.*

**Theorem 4.9.** *Given a pre-training dataset $D$ containing $N$ token sequences $\{\mathbf{X}_i\}_{i=1}^N \subseteq \mathcal{X}$, satisfying the distribution conditions in Assumption 4.2. Denote $\hat{g}$ and $\hat{h}$ as the optimal R-L and T-P derived by solving Equation (2), respectively. Then, under Assumption 4.4, for some $\varphi_0 > 0$, $\varphi_1 > 0$ and $r > 0$, there holds*

$$\mathcal{E}_\mathcal{D}(\hat{g},\hat{h}) \leq \underbrace{6\tilde{\mathfrak{R}}_D\left(\ell \circ \mathcal{G} \circ \mathcal{H}\right) + B_\ell \sqrt{\frac{8\ln\frac{4}{\delta}}{N}}}_{\textbf{I}}$$

$$+ \underbrace{B_\ell \sqrt{\frac{\|\Delta_m\|_\infty^2 \log\frac{2}{\delta}}{2m}}}_{\textbf{II}} + 4B_\ell \operatorname{disc}(U),$$

*with probability at least $1 - \delta$, where $\|\Delta_m\|_\infty \leq 1 + 2\sum_{k=1}^m \varphi(k)$ and $\varphi(k) \leq \varphi_0 \exp\left(-\varphi_1 k^r\right)$, $\forall k \in [m]$.*

The proofs of Proposition 4.8 and Theorem 4.9 are provided in Appendix A and Appendix B, respectively.

*Remark* 4.10. Item **I** represents the generalization error of NTP pre-training on the dataset $D$, reflecting the model's ability to generalize to unseen token sequences and its overall generalization capability within the sequence space. Item **II** denotes the average generalization capability on individual token sequences $\mathbf{X}_i$, indicating the model's local generalization ability within the token space. The last item, $\operatorname{disc}(U)$ (see Definition 4.5), reflects the influence of the quality of the pre-training dataset. Since $\varphi_0$, $\varphi_1$, and $r$ are all greater than 0, the positive series $\sum_{k=1}^\infty \varphi(k)$ is convergent. Therefore, there exists a constant $C_{\varphi_1, \varphi_2, r} > 0$ such that $\|\Delta_m\|_\infty^2 \leq C_{\varphi_1, \varphi_2, r}$. For simplicity, we will use the constant $C_{\varphi, r}$ to represent the upper bound of $\|\Delta_m\|_\infty^2$ in the subsequent analysis.

The following corollary can be derived by combining Theorem 4.9 and Proposition 4.8.

**Corollary 4.11.** *Under the same assumptions as Theorem 4.9, if $g$ is $G_g$-Lipschitz w.r.t. $h$ for any $g \in \mathcal{G}, h \in \mathcal{H}$, there exists a constant $C_{\varphi, r} > 0$ such that the following inequality holds with probability at least $1 - \delta$:*

$$\mathcal{E}_\mathcal{D}(\hat{g},\hat{h}) \leq \underbrace{6G_\ell G_g \tilde{\mathfrak{R}}_D(\mathcal{H})}_{\textbf{(I)}} + \underbrace{6G_\ell \tilde{\mathfrak{R}}_D(\mathcal{G} \circ \hat{h})}_{\textbf{(II)}}$$

$$+ B_\ell \sqrt{\frac{8\ln\frac{4}{\delta}}{N}} + B_\ell \sqrt{\frac{C_{\varphi,r} \log\frac{2}{\delta}}{2m}} + 4B_\ell \operatorname{disc}(U).$$

*Remark* 4.12. Item **(I)** is exclusively associated with the complexity of R-L, while item **(II)** depends solely on the complexity of T-P. These two items operate independently, allowing for separate analysis of the effects of R-L and T-P on generalization performance. This independence also simplifies the process of replacing T-P, as it only requires redefining item **(II)**.

### 4.2. Capacity of Transformer-decoder Models

To investigate the effect of the parameters within the transformer-decoder model (i.e., R-L) on the generalization performance of NTP, we use the covering number to quantify the Rademacher complexity of R-L. We begin by providing a general definition of the covering number.

**Definition 4.13** ($\epsilon$-cover and covering number). Denote $(U, \|\cdot\|)$ as a metric space and $V \subseteq U$. For any $\epsilon > 0$, $V$ is called an $\epsilon$-cover of $U$ if for any $u \in U$, there exists $v \in V$ such that $\|u - v\| \leq \epsilon$. The covering number of $(U, \|\cdot\|)$ is the cardinality of the smallest $\epsilon$-cover, which is defined by

$$\mathcal{N}(U, \epsilon, \|\cdot\|) := \min\{|V| : V \text{ is the } \epsilon\text{-cover of } U\}.$$

**Assumption 4.14.** Assume that

- $\Pi_{\text{norm}}$ is $G_\pi$-Lipschitz with the $\ell_2$-norm, i.e., $\forall \mathbf{t}_1, \mathbf{t}_2 \in \mathbb{R}^d, \|\Pi_{\text{norm}}(\mathbf{t}_1) - \Pi_{\text{norm}}(\mathbf{t}_2)\|_{\ell_2} \leq G_\pi \|\mathbf{t}_1 - \mathbf{t}_2\|_{\ell_2}$.

- $\forall l \in [L]$ and $h \in [H]$, there exists constants $C_l$ such that $\left\|\mathbf{Q}_h^l(\mathbf{K}_h^l)^\top / \sqrt{d_k}\right\|_{\ell_\infty} \leq C_l$.

- $\forall l \in [L], \mathbf{W}^l \in \mathcal{W}^l$, there exists constants $B_l$ satisfying $\|\mathbf{W}^l\|_F \leq B_l$.

The second assumption in Assumption 4.14 is reasonable due to the presence of the scaling factor $\sqrt{d_k}$ in the self-attention mechanism. The first and third assumptions have been previously used in the analysis of the transformer covering number (Edelman et al., 2022; Deng et al., 2024).

Since the learnable parameters of the Transformer model are all fully connected layer parameters, we introduce the following lemma proposed by Lin & Zhang (2019):

**Lemma 4.15.** *Let $\mathbf{X} \in \mathbb{R}^{n \times d_{in}}$ be a given input matrix with a bounded Frobenius norm, and $\mathbf{W} \in \mathbb{R}^{d_{in} \times d_{out}}$ such that $\|\mathbf{W}\|_F \leq a$. Then, we have*

$$\ln \mathcal{N}\left(\{\mathbf{XW} : \|\mathbf{W}\|_F \leq a\}, \epsilon, \|\cdot\|_F\right)$$

$$\leq d_{in} d_{out} \ln\left(1 + \frac{2a\|\mathbf{X}\|_F}{\epsilon}\right).$$

Lemma 4.15 emphasizes the impact of model parameters on the covering number, aligning with our research objectives. Based on this lemma, we provide the upper bound of the logarithmic covering number for masked self-attention as follows:

**Lemma 4.16** (Simplification of Lemma C.10). *Given an input sequence $S = \{\mathbf{Z}_1, \dots, \mathbf{Z}_N\} \in \mathbb{R}^{m \times d}$, denote $\mathbf{Z}_{[N]} = [\mathbf{Z}_1, \dots, \mathbf{Z}_N] \in \mathbb{R}^{Nm \times d}$ as the concatenated data matrix. Consider the masked self-attention head $\mathbf{A}(\cdot)$ (ignore the layer and head indices) defined in Equation (5), the corresponding function class can be defined as:*

$$\mathcal{H}_S^{\mathbf{A}} := \{\mathbf{Z} \mapsto \mathbf{A}(\mathbf{Z}) : \|\mathbf{W}_Q, \mathbf{W}_K, \mathbf{W}_V\|_F \leq B\}.$$

*Then, we have*

$$\ln \mathcal{N}\left(\mathcal{H}_S^{\mathbf{A}}, \epsilon, \|\cdot\|_F\right) \leq dd_k \ln\left(1 + \frac{B^3\|\mathbf{Z}^*\|_F\|\mathbf{Z}_{[N]}\|_F^2}{\sqrt{d_k}\epsilon}\right)$$
$$+ d^2 \ln\left(1 + \frac{e^C B\sqrt{N\ln m}\|\mathbf{Z}_{[N]}\|_F}{\epsilon}\right),$$

*where* $\|\mathbf{Z}^*\|_F = \max_{i\in[N]}\|\mathbf{Z}_i\|_F$.

Then, based on Lemma 4.16, we can further obtain the upper bound of the logarithmic covering number for a single-layer transformer-decoder-block as follows:

**Proposition 4.17.** *For the transformer-decoder-block* $\mathrm{TF}_{\mathcal{W}}$ *(ignoring the layer indices) defined in Equation* (4)*, the corresponding function class can be defined as*

$$\mathcal{H}_S^{TF} := \{\mathbf{Z} \mapsto \mathrm{TF}_{\mathcal{W}}(\mathbf{Z}) : \|\mathbf{W}\|_F \leq B, \mathbf{W} \in \mathcal{W}\}.$$

*Then we can get the following covering number bound:*

$$\ln\mathcal{N}\left(\mathcal{H}_S^{TF}, \epsilon, \|\cdot\|_F\right) \lesssim 4d^2(H+3)\ln\left(1 + \frac{\omega}{\epsilon}\right),$$

*where* $\omega = G_\pi^2 B^2(B^2+1)(e^C B^2 H\sqrt{N\ln m}+1)\|\mathbf{Z}_{[N]}\|_F$.

The following two lemmas mainly explore the Lipschitz continuity of the transformer decoder model.

**Lemma 4.18.** *For a single transformer-decoder-block* $\mathrm{TF}_{\mathcal{W}}(\cdot)$ *parameterized by* $\mathcal{W}$ *(ignore the layer indices), let* $\mathbf{Z}, \hat{\mathbf{Z}} \in \mathbb{R}^{m\times d}$ *be any input matrixes. Then, there holds*

$$\left\|\mathrm{TF}_{\mathcal{W}}(\mathbf{Z}) - \mathrm{TF}_{\mathcal{W}}\left(\hat{\mathbf{Z}}\right)\right\|_F$$
$$\lesssim G_\pi^2(B^2+1)\left(e^C B^2 Hmd+1\right)\left\|\mathbf{Z} - \hat{\mathbf{Z}}\right\|_F.$$

**Lemma 4.19.** *Let* $\mathbf{Z}^l \in \mathbb{R}^{m\times d}$ *as the output matrix of the* $l$*-th layer decoder-block, we have:*

$$\|\mathbf{Z}^l\|_F \leq \prod_{j\in[l]} G_\pi^2(B_j^2+1)\left(e^{C_j}B_j^2 H\sqrt{\ln m}+1\right)\|\mathbf{Z}^0\|_F.$$

Based on Proposition 4.17 and Lemmas 4.18 and 4.19 (proved in Appendix C.3), we obtain the following logarithmic covering number upper bound for the R-L.

**Theorem 4.20.** *Let* $D = \{\mathbf{X}_i\}_{i=1}^N$ *be a a dataset containing* $N$ *token sequences and let* $\mathbf{Z}_{[N]} = [\mathbf{Z}_1, \ldots, \mathbf{Z}_N] \in \mathbb{R}^{Nm\times n_v}$ *be the input matrix generated from D, and denote* $\mathbf{Z}_{[N]}^0 \in \mathbb{R}^{Nm\times d}$ *as the embedded matrix. The function class of the R-L defined in Equation* (6) *can be defined as*

$$\mathcal{H} := \{\mathbf{Z} \mapsto h(\mathbf{Z}) : \|\mathbf{W}^l\|_F \leq B_l, \mathbf{W}^l \in \mathcal{W}^l, \forall l \in [L]\}.$$

*Then, under Assumption 4.14, we have*

$$\ln\mathcal{N}\left(\mathcal{H}, \epsilon, \|\cdot\|_F\right) \leq \frac{\Theta H}{L}\sum_{l=1}^L \ln\left(1 + \frac{LB_l^2 s_L\|\mathbf{Z}_{[N]}^0\|_F}{\epsilon}\right),$$

*where* $\Theta \approx 12Ld^2$ *is the number of model parameters and*

$$s_L := \prod_{l\in[L]} G_\pi^2(B_l^2+1)\left(e^{C_l}B_l^2 H\sqrt{N}md+1\right).$$

*Remark* 4.21. As demonstrated in Bartlett et al. (2017), the logarithm of the covering number of $\mathcal{H}$ under the infinite-norm is bounded above by that under the Frobenius-norm, i.e., $\ln\mathcal{N}(\mathcal{H}, \epsilon, \|\cdot\|_{\ell_\infty}) \leq \ln\mathcal{N}(\mathcal{H}, \epsilon, \|\cdot\|_F)$. However, our approach bounds the Frobenius-norm covering number with a smaller order of $\mathcal{O}(L)$ compared to the orders $C^{\mathcal{O}(L)}$ (where $C > 1$) reported in Edelman et al. (2022); Deng et al. (2024). This indicates that our method offers a significant advantage over the previous studies.

### 4.3. Generalization Bounds for DOMs

Inspired by Bartlett et al. (2017), we use the covering number to deduce the upper bound of Rademacher complexity. Then, the excess risk for NTP pre-training can be bounded by integrating Corollary 4.11 and Theorem 4.20.

**Theorem 4.22.** *Let* $\mathbf{Z}_{[N]} \in \mathbb{R}^{Nm\times n_v}$ *be the input sequences generated from dataset D. Denote* $\hat{g}$ *and* $\hat{h}$ *as the optimal R-L and T-P learned from Equation* (2)*, respectively. Then, under Assumptions 4.2, 4.4 and 4.14, there exists a constant* $C_{\varphi,r} > 0$ *such that the following inequality holds with probability at least* $1 - \delta$:

$$\mathcal{E}_{\mathcal{D}}(\hat{f}, \hat{h}) \lesssim \mathcal{O}\left(\sqrt{\frac{\Theta dH\tau_1}{Nm}}\right) + G_\ell\sqrt{\frac{dn_v}{Nm}}$$
$$+ B_\ell\left(\sqrt{\frac{8\ln\frac{4}{\delta}}{N}} + \sqrt{\frac{C_{\varphi,r}\log\frac{2}{\delta}}{2m}} + 4\,\mathrm{disc}(U)\right),$$

*where* $\Theta$ *is the number of model parameters, and* $\tau_1 = \ln(1+\rho_L s_L)$*, with* $\rho_L = \sum_{l=1}^L B_l^2$ *and constant* $s_L$ *defined in Theorem 4.20.*

*Remark* 4.23. We focus on three parameters: the number of token sequences $N$, the maximum length of the token sequence $m$, and the number of model parameters $\Theta$. Our bound is $\mathcal{O}(\sqrt{\Theta/Nm} + \sqrt{C_{\varphi,r}/m})$, where $\mathcal{O}(\sqrt{\Theta/Nm})$ reflects the generalization ability between token sequences, and $\mathcal{O}(\sqrt{C_{\varphi,r}/m})$ reflects the generalization capacity among tokens within a sequence. Here, $C_{\varphi,r}$ is a constant related to the $\varphi$-mixing coefficient, indicating the distribution quality of a single token sequence, while $\mathrm{disc}(U)$ reflects the overall distribution quality of all token sequences. Our bound captures the impact of both dataset quality and individual sample quality on generalization performance. Unlike previous works (see Table 1), we show that effective generalization requires both a larger $N$ and a larger $m$, enabling the model to generalize across both sequence space and token space. Additionally, as $\Theta$ increases, the total number of tokens $Nm$ should also increase to achieve better generalization.

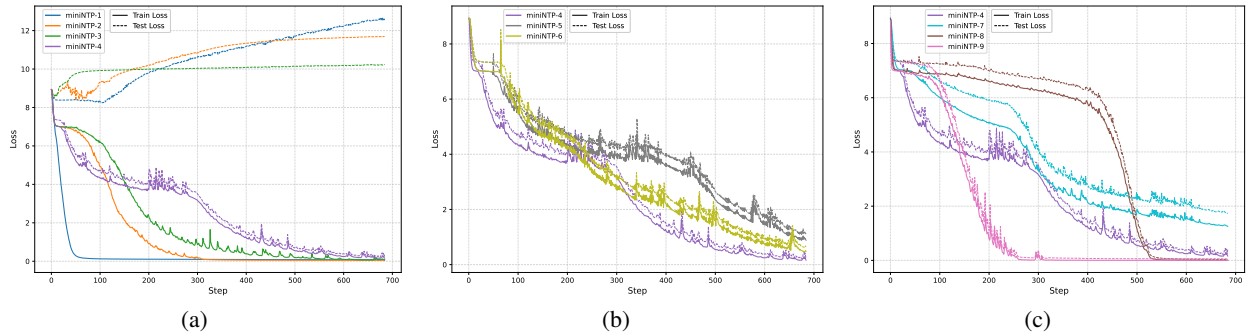

*Figure 2.* Experiments on MiniMind and DAMO_NLP datasets.

*Remark* 4.24. It should be noted that our bounds remain valid even when modifications are made to the structure of the transformer-decoder block. For instance, replacing post-layer Normalization with pre-layer Normalization or substituting the ReLU activation function with SiLU. This is attributable to the fact that such modifications exclusively affect the logarithmic terms $\rho_L$ and $s_L$.

## 5. Experiments

To validate the theoretical contribution of this paper, specifically, Theorem 4.22, we performed a set of NTP pre-training experiments in DOMs. These experiments were designed to systematically examine the influence of model parameters and sample size on generalization performance.

### 5.1. Setup

**Model Architecture.** Our architecture largely follows the GPT-2 framework, with two key modifications: (1) the adoption of the SiLU activation function and (2) RMSNorm normalization (Zhang & Sennrich, 2019). As demonstrated in Remark 4.24, these adjustments do not compromise the validity of our theoretical framework.

**Datasets.** For pretraining, we employ the MiniMind dataset[1], while our test set consists of 8,192 samples (with a maximum sequence length of $m \leq 512$) carefully selected from the DAMO_NLP dataset[2]. Both datasets belong to the category of Chinese text generation datasets. Due to the consistent pretraining corpus, we adopted the same tokenizer as MiniMind[3], preserving a vocabulary size of $n_v = 6400$.

**Training Protocol.** Our training methodology follows the approach outlined in MiniMind. To optimize efficiency, we employ FlashAttention (Dao et al., 2022) for accelerated attention computation and conduct distributed training on

---

[1] https://www.modelscope.cn/datasets/gongjy/minimind_dataset
[2] https://www.modelscope.cn/datasets/DAMO_NLP/lcsts_test_set
[3] https://github.com/jingyaogong/minimind

8× NVIDIA A800-80GB GPUs using DeepSpeed-Zero2 (Rajbhandari et al., 2020). For optimization, we utilized the AdamW (Loshchilov & Hutter, 2017) optimizer, combined with a cosine learning rate scheduler that includes a 20-step warm-up phase during the initial training stage.

Full specifications for model architecture, dataset preprocessing, and training configurations are detailed in Table 3.

### 5.2. Main Results

**Maximum sequence length $m$.** As demonstrated in Figure 2(a), we performed experiments with varying maximum sequence lengths $m$ (64, 128, 256, 512) in the training dataset while holding all other parameters constant. Notably, the test dataset retained a fixed maximum sequence length of 512 across all evaluations. While models trained on shorter sequences converge more rapidly, they exhibit limited generalization capability when applied to longer sequences. This observation accounts for the monotonically increasing test loss trend observed for miniNTP-1, 2, and 3 as the training sequence length decreases. As shown in Table 2, models trained on shorter sequences deliver strong performance on test cases with similarly short sequences. Importantly, models trained on longer sequences maintain robust performance even when evaluated on shorter sequences, highlighting their adaptability.

*Table 2.* Test sample perplexity (PPL) variations across models under variable maximum sequence lengths ($m$).

| Model | $m = 64$ | $m = 128$ | $m = 256$ | $m = 512$ |
|---|---|---|---|---|
| miniNTP-1 | 1.10 | 69.35 | 1578.14 | 316024.25 |
| miniNTP-2 | 1.24 | 1.31 | 224.25 | 130613.71 |
| miniNTP-3 | 1.03 | 1.05 | 1.08 | 24343.04 |
| miniNTP-4 | 1.03 | 1.16 | 1.24 | 1.49 |

**The number of sequences $N$.** For models with identical architectural configurations, we demonstrate a enhancement in generalization performance with increasing training sequence quantity. As illustrated in Figure 2(b), while holding model parameters, training hyperparameters, and maximum sequence length ($m = 512$) constant, we evaluated perfor-

*Table 3.* Model architectures, training data specifications, hyperparameter configurations, and test PPL ($m = 512$).

| Model | $\Theta$ | $L$ | $H$ | $d$ | $m$ | $N\%$ | Batch Size | Learning Rate | PPL |
|---|---|---|---|---|---|---|---|---|---|
| miniNTP-1 | 0.029B | 8 | 8 | 512 | 64 | 100 | 0.5M | 5.0e-4 | 316024.25 |
| miniNTP-2 | 0.029B | 8 | 8 | 512 | 128 | 100 | 0.5M | 5.0e-4 | 130613.71 |
| miniNTP-3 | 0.029B | 8 | 8 | 512 | 256 | 100 | 0.5M | 5.0e-4 | 24343.04 |
| miniNTP-4 | 0.029B | 8 | 8 | 512 | 512 | 100 | 0.5M | 5.0e-4 | 1.49 |
| miniNTP-5 | 0.029B | 8 | 8 | 512 | 512 | 50 | 0.5M | 5.0e-4 | 3.17 |
| miniNTP-6 | 0.029B | 8 | 8 | 512 | 512 | 75 | 0.5M | 5.0e-4 | 1.95 |
| miniNTP-7 | 0.002B | 6 | 4 | 128 | 512 | 100 | 0.5M | 1.0e-3 | 5.76 |
| miniNTP-8 | 0.09B | 12 | 12 | 768 | 512 | 100 | 0.5M | 6.0e-4 | 1.13 |
| miniNTP-9 | 0.31B | 24 | 16 | 1024 | 512 | 100 | 0.5M | 3.0e-4 | 1.05 |

mance across 50%, 75%, and 100% subsets of the complete pretraining dataset. The experimental results reveal that diminishing training data volume not only compromises model generalization but also significantly impairs convergence characteristics and training stability.

**Model size $\Theta$.** Our analysis in Figure 2(c) evaluates how model parameter size ($\Theta$) affects generalization performance under fixed training sequence count ($N$) and maximum length ($m$), with configurations adapted from Biderman et al. (2023). Early training (step<100) shows smaller models converge faster, but prolonged training demonstrates larger models achieve superior convergence rates and lower final losses. As predicted by Theorem 4.20, this divergence arises from capacity limits: smaller models saturate earlier while larger ones continue learning from additional tokens.

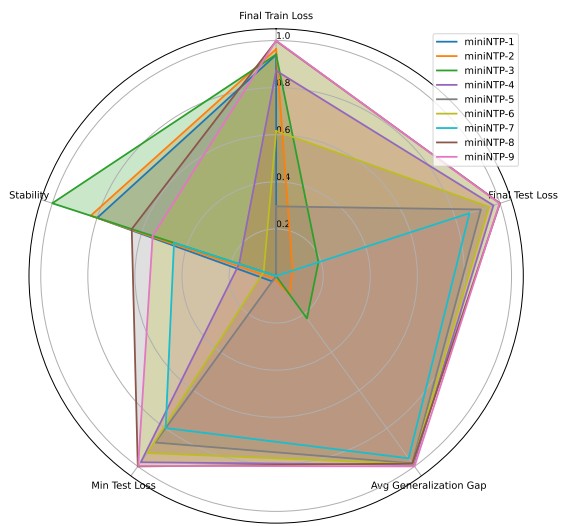

*Figure 3.* Model generalization ability radar chart.

Figure 3 provides a comprehensive visualization comparing the nine models across five distinct evaluation dimensions, with radial distance from the center indicating performance quality in each dimension. The analysis reveals that extending both the maximum sequence length ($m$) and the number of training sequences ($N$) improves model gen-

eralization in the NTP pretraining task. Notably, increasing $m$ produces more substantial performance gains than expanding $N$, though this comes with increased training complexity. Additionally, while larger model parameters facilitate learning richer token representations and elevate the model's capacity ceiling, this architectural expansion should be matched with a corresponding increase in total token quantity to mitigate overfitting risks.

## 6. Conclusion

This paper presents a generalization error analysis for next-token prediction pre-training, a widely used paradigm in large language models. Our theoretical results enhance the understanding of how model parameters influence generalization ability. We find that generalization depends on the number of token sequences, the maximum sequence length, and the number of parameters in the transformer model. Empirical evaluations confirm our theoretical findings through data experiments.

## 7. Future Work

In the rapidly advancing field of large language models, theoretical foundations remain underdeveloped. While our study addresses part of this research gap, we identify several promising avenues for future work. First, although our $\varphi$-mixing data modeling approach demonstrates theoretical validity, empirical verification in practical applications requires further investigation. Beyond mixing processes, developing language-specific data distributions could provide deeper insights into how linguistic properties affect model behavior. Second, this work primarily uses Rademacher complexity for theoretical analysis, other frameworks like stability-based (Liu et al., 2024b; Zhang et al., 2024b) or information-theoretic (Lu & Van Roy, 2019; Livni, 2023) methods are viable alternatives. Finally, given the anticipated evolution toward unified multimodal architectures, extending this research to incorporate diverse data modalities represents a crucial direction for future exploration.

## Acknowledgments

This work is supported by the National Natural Science Foundation of China (NSFC) (Nos. 62376104 and 12426512) and the Open Research Fund of Engineering Research Center of Intelligent Technology for Agriculture, Ministry of Education (No. ERCITA-KF002).

## Impact Statement

This paper presents work whose goal is to advance the field of Machine Learning. There are many potential societal consequences of our work, none which we feel must be specifically highlighted here.

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

# Appendix

## Summary & Technical Route

This paper conducts a comprehensive theoretical analysis by focusing on the pre-training methodology of Large Language Models (LLMs) known as Next-Token-Prediction (NTP). It categorizes LLMs as transformer decoder-only models (DOMs) and delves into the empirical successes of NTP despite a lack of theoretical understanding. This work also establishes a theoretical framework to analyze the generalization behavior of NTP pre-training. We introduce a novel decomposition of Rademacher complexity to study the representation-learner and token-predictor components of DOMs. The paper also addresses the dependence between tokens using $\varphi$-mixing, a tool commonly used in non-independent scenarios, to delineate inter-token dependencies. This approach allows for a fine-grained analysis of the generalization ability of NTP pre-training, considering the model's structure and the nature of the training data.

The technical route of the paper involves developing a theoretical framework for NTP from a statistical learning perspective. This work proposes a decomposition law for Rademacher complexity to bound the excess risk of NTP and establish different bounds on the generalization capability. We refine the estimation of the covering number for multi-layer multi-head transformer-decoder models, pioneering the incorporation of the mask matrix within the self-attention mechanism under the Frobenius norm. This paper uses the covering number to derive the corresponding Rademacher complexity upper bound, extending the theory of (Bartlett et al., 2017) and (Lin & Zhang, 2019) to establish fine-grained generalization bounds for DOMs-based NTP pre-training. The results are expressed in terms of key parameters that affect the generalization ability, providing a clear and quantifiable understanding of how NTP pre-training behaves in practice.

## Outline of the Appendix

The appendix is mainly structured as follows,

- Section A: Proof of the Proposition 4.8.

- Section B: Proof of the Theorem 4.9.

- Section C: Capacity of DOMs.

    - Section C.1: Introduction to the model architecture.
    - Section C.2: Restatement to some useful lemmas.
    - Section C.3: Proof of the Proposition C.11.
    - Section C.4: Proof of the Theorem 4.20.

- Section D: Proof of the Theorem 4.22.

## A. Proof of the Proposition 4.8

*Proof.*

$$
\begin{aligned}
\hat{\mathfrak{R}}_S(\ell \circ \mathcal{G} \circ \mathcal{H}) &= \mathbb{E}_{\boldsymbol{\varepsilon}} \left[ \sup_{f \in \mathcal{F}} \frac{1}{n} \sum_{i \in [n]} \varepsilon_i \ell \circ g \circ h\,(z_i) \right] \\
&\overset{(a)}{\leq} G_\ell \mathbb{E}_{\boldsymbol{\varepsilon}} \left[ \sup_{g \in \mathcal{G}, h \in \mathcal{H}} \frac{1}{n} \sum_{i \in [n]} \varepsilon_i \left\| g \circ h\,(z_i) \right\| \right] \\
&\leq G_\ell \mathbb{E}_{\boldsymbol{\varepsilon}} \left[ \sup_{g \in \mathcal{G}, h \in \mathcal{H}} \frac{1}{n} \sum_{i \in [n]} \varepsilon_i \left\| g \circ h\,(z_i) - g \circ \hat{h}\,(z_i) \right\| \right] + G_\ell \mathbb{E}_{\boldsymbol{\varepsilon}} \left[ \sup_{g \in \mathcal{G}} \frac{1}{n} \sum_{i \in [n]} \varepsilon_i \left\| g \circ \hat{h}\,(z_i) \right\| \right] \\
&\overset{(b)}{\leq} G_\ell G_g \mathbb{E}_{\boldsymbol{\varepsilon}} \left[ \sup_{h \in \mathcal{H}} \frac{1}{n} \sum_{i \in [n]} \varepsilon_i \left\| h\,(z_i) - \hat{h}\,(z_i) \right\| \right] + G_\ell \hat{\mathfrak{R}}_S(\mathcal{G} \circ \hat{h}) \\
&\leq G_\ell G_g \mathbb{E}_{\boldsymbol{\varepsilon}} \left[ \sup_{h \in \mathcal{H}} \frac{1}{n} \sum_{i \in [n]} \varepsilon_i \left\| h\,(z_i) \right\| \right] + G_\ell G_g \mathbb{E}_{\boldsymbol{\varepsilon}} \left[ \frac{1}{n} \sum_{i \in [n]} \varepsilon_i \left\| \hat{h}\,(z_i) \right\| \right] + G_\ell \hat{\mathfrak{R}}_S(\mathcal{G} \circ \hat{h}) \\
&\overset{(c)}{=} G_\ell G_g \hat{\mathfrak{R}}_S(\mathcal{H}) + G_\ell \hat{\mathfrak{R}}_S(\mathcal{G} \circ \hat{h}),
\end{aligned}
$$

where $\hat{h}$ is a any given function in $\mathcal{H}$. Here $(a)$ is by Ledoux-Talagrand contraction inequality, $(b)$ uses the *Lipschitz* conditions of $g$, and $(c)$ uses the property that the Rademacher random variables $\boldsymbol{\varepsilon}$ are $i.i.d.$ with zero mean. □

## B. Proof of Theorem 4.9

Firstly, we give two necessary assumptions which have been mentioned before.

**Assumption B.1.** Assume that $\mathbf{X}_i = \left[ z_1^i, \cdots, z_m^i \right] \in \mathbb{R}^{m \times n_v}$ is generated by a $\varphi$-mixing distribution $\phi_i$, and there exists an unknown distribution $\mathcal{U}$ such that $U = \{\phi_i\}_{i=1}^N \sim \mathcal{U}$.

**Assumption B.2.** Assume there exists a constant $B_\ell \in \mathbb{R}^+$ satisfying $|\ell(\hat{\mathbf{t}}, \mathbf{t})| \leq B_\ell$ for any $\hat{\mathbf{t}}, \mathbf{t} \in \mathcal{T}$, and $\ell$ is $G_\ell$-Lipschitz w.r.t. $\hat{\mathbf{t}}$.

Then, we introduce some related lemmas which will be used in our proof. Since $\{\mathbf{X}_i\}_{i=1}^N$ are independent of each other, $\{\hat{\mathcal{L}}_{\mathbf{X}_i}(g \circ h)\}_{i=1}^N$ are also independent of each other, so the generalization error based on the dataset $D$ can be bounded by the following common theorem.

**Lemma B.3** (Shalev-Shwartz & Ben-David (2014)). *Given a dataset $D = \{\mathbf{X}_i\}_{i=1}^N \overset{i.i.d.}{\sim} \mathcal{D}$, if loss function $\ell : \mathcal{T} \times \mathcal{T} \to [-B_\ell, B_\ell]$, $B_\ell \in \mathbb{R}^+$, then with probability at least $1 - \delta$, the following inequality holds for any $h \in \mathcal{H}$ and $g \in \mathcal{G}$:*

$$
\mathcal{L}_{\mathcal{D}}(g \circ h) \leq \hat{\mathcal{L}}_D(g \circ h) + 2\hat{\mathfrak{R}}_D(\ell \circ \mathcal{G} \circ \mathcal{H}) + 4B_\ell \sqrt{\frac{2 \log \frac{4}{\delta}}{N}}.
$$

For the non-independent case, Mohri & Rostamizadeh (2010) gave a Rademacher complexity bound under the $\varphi$-mxing distribution. Therefore, under Assumption B.1, we can define the generalization error based on a single token sequence $\mathbf{X}_i$, mainly using the following theorem:

**Lemma B.4** (Mohri & Rostamizadeh (2010)). *Given a token sequence $\mathbf{X}_i = [\mathbf{t}_1^i, \cdots, \mathbf{t}_m^i]$ and loss function $\ell : \mathcal{T} \times \mathcal{T} \to [-B_\ell, B_\ell]$, $B_\ell \in \mathbb{R}^+$, if $\{\mathbf{t}_j^i\}_{j=1}^m$ follow a $\varphi$-mxing distribution $\phi_i$, then for some $\varphi_0 > 0$, $\varphi_1 > 0$ and $r > 0$, with probability at least $1 - \delta$, the following inequality holds for any $h \in \mathcal{H}$ and $g \in \mathcal{G}$:*

$$
\left| \mathcal{L}_{\phi_i}(g \circ h) - \hat{\mathcal{L}}_{\mathbf{X}_i}(g \circ h) \right| \leq B_\ell \sqrt{\frac{\|\Delta_m\|_\infty^2 \log \frac{2}{\delta}}{2m}},
$$

*where $\|\Delta_m\|_\infty \leq 1 + 2 \sum_{k=1}^m \varphi(k)$, and $\varphi(k) \leq \varphi_0 \exp\left(-\varphi_1 k^r\right)$ for all $k \in [m]$.*

**Lemma B.5.** *For the two Rademacher complexity over the dataset D mentioned before, we have the following inequality:*

$$\hat{\mathfrak{R}}_D\left(\ell \circ \mathcal{G} \circ \mathcal{H}\right) \le 3\tilde{\mathfrak{R}}_D\left(\ell \circ \mathcal{G} \circ \mathcal{H}\right).$$

*Proof.* Let $\boldsymbol{\varepsilon}' = \{\varepsilon_i'\}_{i=1}^N, \boldsymbol{\varepsilon}'' = \{\varepsilon_j''\}_{j=1}^m, \boldsymbol{\varepsilon} = \{\{\varepsilon_{ij}\}_{j=1}^m\}_{i=1}^N$ be three *i.i.d.* Rademacher random variable collections, and $\boldsymbol{\varepsilon}', \boldsymbol{\varepsilon}''$ are independent of each other. For ease of representation, we denote $\ell_j^i = \ell\left(g \circ h(\mathbf{z}_j^i), \mathbf{z}_j^i\right)$. Then, we have:

$$\hat{\mathfrak{R}}_D\left(\ell \circ \mathcal{G} \circ \mathcal{H}\right) = \mathbb{E}_{\boldsymbol{\varepsilon}'}\left[\sup_{g \in \mathcal{G}, h \in \mathcal{H}} \frac{1}{N}\sum_{i \in [N]} \varepsilon_i'\left(\frac{1}{m}\sum_{j \in [m]} \ell\left(g \circ h(\mathbf{z}_j^i), \mathbf{z}_j^i\right)\right)\right]$$

$$= \mathbb{E}_{\boldsymbol{\varepsilon}'}\left[\sup_{g \in \mathcal{G}, h \in \mathcal{H}} \frac{1}{Nm}\sum_{i \in [N]}\sum_{j \in [m]}\left(\varepsilon_i'\ell_j^i - \varepsilon_j''\ell_j^i + \varepsilon_j''\ell_j^i\right)\right]$$

$$= \mathbb{E}_{\boldsymbol{\varepsilon}''}\left[\mathbb{E}_{\boldsymbol{\varepsilon}'}\left[\sup_{g \in \mathcal{G}, h \in \mathcal{H}} \frac{1}{Nm}\sum_{i \in [N]}\sum_{j \in [m]}\left(\left(\varepsilon_i' - \varepsilon_j''\right)\ell_j^i + \varepsilon_j''\ell_j^i\right)\right]\right]$$

$$\le \underbrace{\mathbb{E}_{\boldsymbol{\varepsilon}''}\left[\mathbb{E}_{\boldsymbol{\varepsilon}'}\left[\sup_{g \in \mathcal{G}, h \in \mathcal{H}} \frac{1}{Nm}\sum_{i \in [N]}\sum_{j \in [m]}\left(\varepsilon_i' - \varepsilon_j''\right)\ell_j^i\right]\right]}_{(I)} + \underbrace{\mathbb{E}_{\boldsymbol{\varepsilon}''}\left[\mathbb{E}_{\boldsymbol{\varepsilon}'}\left[\sup_{g \in \mathcal{G}, h \in \mathcal{H}} \frac{1}{Nm}\sum_{i \in [N]}\sum_{j \in [m]} \varepsilon_j''\ell_j^i\right]\right]}_{(II)}.$$

For part $(I)$, we denote $\hat{\boldsymbol{\varepsilon}} = \{\{\hat{\varepsilon}_{ij}\}_{j=1}^m\}_{i=1}^N$, where $\hat{\varepsilon}_{ij} = \frac{1}{2}\left(\varepsilon_i' - \varepsilon_j''\right)$. It's easy to get $\hat{\boldsymbol{\varepsilon}}$ are *i.i.d.* random variables, and the distribution is:

$$p\left(\hat{\varepsilon}_{ij}\right) = \begin{cases} 1/4, & \hat{\varepsilon}_{ij} = 1 \\ 1/2, & \hat{\varepsilon}_{ij} = 0 \\ 1/4, & \hat{\varepsilon}_{ij} = -1 \end{cases}.$$

Then, by the independence of $\boldsymbol{\varepsilon}'$ and $\boldsymbol{\varepsilon}''$, we have:

$$(I) = 2\mathbb{E}_{\boldsymbol{\varepsilon}''}\left[\mathbb{E}_{\boldsymbol{\varepsilon}'}\left[\sup_{g \in \mathcal{G}, h \in \mathcal{H}} \frac{1}{Nm}\sum_{i \in [N]}\sum_{j \in [m]} \frac{1}{2}\left(\varepsilon_i' - \varepsilon_j''\right)\ell_j^i\right]\right]$$

$$= 2\mathbb{E}_{\hat{\boldsymbol{\varepsilon}}}\left[\sup_{g \in \mathcal{G}, h \in \mathcal{H}} \frac{1}{Nm}\sum_{i \in [N]}\sum_{j \in [m]} \hat{\varepsilon}_{ij}\ell_j^i\right]$$

$$\le 2\mathbb{E}_{\boldsymbol{\varepsilon}}\left[\sup_{g \in \mathcal{G}, h \in \mathcal{H}} \frac{1}{Nm}\sum_{i \in [N]}\sum_{j \in [m]} \varepsilon_{ij}\ell_j^i\right].$$

For part $(II)$, we denote $\tilde{\boldsymbol{\varepsilon}} = \{\{\tilde{\varepsilon}_{ij}\}_{j=1}^m\}_{i=1}^N$, where $\tilde{\varepsilon}_{ij} = \varepsilon_i'\varepsilon_j''$. It's easy to get $\tilde{\boldsymbol{\varepsilon}}$ are *i.i.d.* Rademacher random variables. We have:

$$(II) = \mathbb{E}_{\boldsymbol{\varepsilon}'}\left[\mathbb{E}_{\boldsymbol{\varepsilon}''}\left[\sup_{g \in \mathcal{G}, h \in \mathcal{H}} \frac{1}{Nm}\sum_{i \in [N]}\sum_{j \in [m]} \varepsilon_j''\ell_j^i\right]\right]$$

$$= \mathbb{E}_{\boldsymbol{\varepsilon}'}\left[\mathbb{E}_{\boldsymbol{\varepsilon}''}\left[\sup_{g \in \mathcal{G}, h \in \mathcal{H}} \frac{1}{Nm}\sum_{i \in [N]}\sum_{j \in [m]} \varepsilon_i'\varepsilon_j''\ell_j^i\right]\right]$$

$$= \mathbb{E}_{\tilde{\boldsymbol{\varepsilon}}}\left[\sup_{g \in \mathcal{G}, h \in \mathcal{H}} \frac{1}{Nm}\sum_{i \in [N]}\sum_{j \in [m]} \tilde{\varepsilon}_{ij}\ell_j^i\right]$$

$$= \mathbb{E}_{\boldsymbol{\varepsilon}}\left[\sup_{g \in \mathcal{G}, h \in \mathcal{H}} \frac{1}{Nm}\sum_{i \in [N]}\sum_{j \in [m]} \varepsilon_{ij}\ell_j^i\right].$$

Therefore we can get: $\hat{\mathfrak{R}}_D \left( \ell \circ \mathcal{G} \circ \mathcal{H} \right) \leq (I) + (II) \leq 3\tilde{\mathfrak{R}}_D \left( \ell \circ \mathcal{G} \circ \mathcal{H} \right).$ □

**Lemma B.6** (Levin & Peres (2017)). *Given two probability measures $\mathcal{T}_1$ and $\mathcal{T}_2$ over instance space $\mathcal{X}$, the following equality holds:*

$$\|\mathcal{T}_1 - \mathcal{T}_2\|_{TV} = \frac{1}{2} \sum_{\mathbf{z} \in \mathcal{X}} |\mathcal{T}_1(\mathbf{z}) - \mathcal{T}_2(\mathbf{z})|.$$

We then give some necessary symbol descriptions as follows:

$$g^*, h^* = \arg \min_{g \in \mathcal{G}, h \in \mathcal{H}} \mathcal{L}_\mathcal{D} \left( g \circ h \right), \tag{9}$$

$$k = \arg \max_{i \in [N]} \mathcal{L}_{\phi_i} \left( g^* \circ h^* \right). \tag{10}$$

In the above symbols, $g^*$ and $h^*$ in (9) are the Token-Predictor and Representation-Learner that minimize the expected risk, exactly the best $g$ and $h$ that we hope to learn through (2). $k$ represents the subscript that maximizes the expected risk based on distribution $\mathcal{T}_k$ ($k \in [N]$) when using $g^*$ and $h^*$, therefore $\mathcal{T}_k$ represents the worst distribution in $\mathcal{U} = \{\mathcal{T}_k\}_{k=1}^N$. Based on the above lemmas and notations, we begin the proof of Theorem 1.

*Proof.* We first perform an error decomposition on the excess risk defined in (3):

$$\begin{aligned}
\mathcal{E}_\mathcal{D}(\hat{g}, \hat{h}) &= \mathcal{L}_\mathcal{D}(\hat{g} \circ \hat{h}) - \mathcal{L}_\mathcal{D} \left( g^* \circ h^* \right) \\
&= \mathcal{L}_\mathcal{D}(\hat{g} \circ \hat{h}) - \hat{\mathcal{L}}_D(\hat{g} \circ \hat{h}) + \hat{\mathcal{L}}_D(\hat{g} \circ \hat{h}) - \mathcal{L}_\mathcal{D} \left( g^* \circ h^* \right) \\
&\leq \underbrace{\mathcal{L}_\mathcal{D}(\hat{g} \circ \hat{h}) - \hat{\mathcal{L}}_D(\hat{g} \circ \hat{h})}_{\text{I}} + \underbrace{\hat{\mathcal{L}}_D \left( g^* \circ \hat{h} \right) - \mathcal{L}_\mathcal{D} \left( g^* \circ h^* \right)}_{\text{II}},
\end{aligned}$$

**Bounding** I : According to Lemma B.3, we can get

$$\begin{aligned}
\text{I} &= \mathcal{L}_\mathcal{D}(\hat{g} \circ \hat{h}) - \hat{\mathcal{L}}_D(\hat{g} \circ \hat{h}) \\
&\leq 2\hat{\mathfrak{R}}_D \left( \ell \circ \mathcal{G} \circ \mathcal{H} \right) + 4B_\ell \sqrt{\frac{2 \log \frac{4}{\delta}}{N}},
\end{aligned}$$

**Bounding** II :

$$\begin{aligned}
\text{II} &= \hat{\mathcal{L}}_D \left( g^* \circ \hat{h} \right) - \mathcal{L}_\mathcal{D} \left( g^* \circ h^* \right) \\
&= \frac{1}{N} \sum_{i \in [N]} \left( \hat{\mathcal{L}}_{\mathbf{x}^i} \left( g^* \circ \hat{h} \right) - \mathcal{L}_{\mathcal{T}^i} \left( g^* \circ h^* \right) \right) \\
&= \frac{1}{N} \sum_{i \in [N]} \left( \hat{\mathcal{L}}_{\mathbf{x}^i} \left( g^* \circ \hat{h} \right) - \mathcal{L}_{\phi_i} \left( g^* \circ \hat{h} \right) + \mathcal{L}_{\phi_i} \left( g^* \circ \hat{h} \right) - \mathcal{L}_{\phi_i} \left( g^* \circ h^* \right) \right) \\
&\leq \underbrace{\frac{1}{N} \sum_{i \in [N]} \left| \hat{\mathcal{L}}_{\mathbf{x}^i} \left( g^* \circ \hat{h} \right) - \mathcal{L}_{\phi_i} \left( g^* \circ \hat{h} \right) \right|}_{\text{III}} + \underbrace{\frac{1}{N} \sum_{i \in [N]} \left( \mathcal{L}_{\phi_i} \left( g^* \circ \hat{h} \right) - \mathcal{L}_{\phi_i} \left( g^* \circ h^* \right) \right)}_{\text{IV}},
\end{aligned}$$

**Bounding** III : According to Lemma B.4, we can get

$$\begin{aligned}
\text{III} &= \frac{1}{N} \sum_{i \in [N]} \left| \hat{\mathcal{L}}_{\mathbf{X}_i} \left( g^* \circ \hat{h} \right) - \mathcal{L}_{\phi_i} \left( g^* \circ \hat{h} \right) \right| \\
&\leq B_\ell \sqrt{\frac{\|\Delta_m\|_\infty^2 \log \frac{2}{\delta}}{2m}},
\end{aligned}$$

**Bounding** IV :

$$
\begin{aligned}
\text{IV} &= \frac{1}{N} \sum_{i \in [N]} \left( \mathcal{L}_{\phi_i} \left( g^* \circ \hat{h} \right) - \mathcal{L}_{\phi_i} \left( g^* \circ h^* \right) \right) \\
&\leq \frac{1}{N} \sum_{i \in [N]} \left( \mathcal{L}_{\mathcal{T}_k} \left( g^* \circ h^* \right) - \mathcal{L}_{\phi_i} \left( g^* \circ h^* \right) \right) + \frac{1}{N} \sum_{i \in [N]} \left( \mathcal{L}_{\phi_i} \left( g^* \circ \hat{h} \right) - \mathcal{L}_{\mathcal{T}_k} \left( g^* \circ \hat{h} \right) \right) \\
&\leq \frac{1}{N} \sum_{i \in [N]} \left( \sum_{z \in \mathcal{X}} |\mathcal{T}_k(z) - \phi_i(z)| \cdot \ell \left( g^* \circ h^*(z), z \right) \right) + \frac{1}{N} \sum_{i \in [N]} \left( \sum_{z \in \mathcal{X}} |\phi_i(z) - \mathcal{T}_k(z)| \cdot \ell \left( g^* \circ \hat{h}(z), z \right) \right) \\
&\leq \frac{2 B_\ell}{N} \sum_{i \in [N]} \left( \sum_{z \in \mathcal{X}} |\phi_i(z) - \mathcal{T}_k(z)| \right) \\
&\overset{(i)}{=} \frac{4 B_\ell}{N} \sum_{i \in [N]} \| \phi_i - \mathcal{T}_k \|_{\text{TV}} \\
&\leq 4 B_\ell \, \text{disc}(U),
\end{aligned}
$$

where $(i)$ is by Lemma B.6. Combining the above processes and by Lemma B.5, Theorem 1 is obtained. $\qquad \square$

## C. Capacity of DOMs

### C.1. Model architecture

In this section, we describe the architecture and function class of the decoder-only transformer model in detail. Given a pre-training dataset $D = \{ \mathbf{X}_i \}_{i=1}^N \subseteq \mathbb{R}^{m \times n_v}$ containing $N$ token sequences, where $m$ represents the maximum word vector length and $n_v$ represents the vocabulary size. We can get $N$ input matrixes $\{ \mathbf{Z}_i \}_{i=1}^N \subseteq \mathbb{R}^{m \times n_v}$. We first introduce two normalization operations that will be used. For a given matrix $\mathbf{A} \in \mathbb{R}^{n \times m}$, we denote $\text{softmax}(\cdot)$ as the row-wise softmax operator, which can be defined as:

$$
\text{softmax}(\mathbf{A})_{i,j} = \frac{\exp(\mathbf{A}_{i,j})}{\sum_{j' \in [m]} \exp(\mathbf{A}_{i,j'})}. \tag{11}
$$

Let $\Pi_{\text{norm}}(\cdot)$ denote the Layer-norm operator (Ba et al., 2016), which can be defined as:

$$
\Pi_{\text{norm}}(\mathbf{A})_{i,j} = \frac{\mathbf{A}_{i,j} - \mu}{\delta}, where \begin{cases} \mu &= \frac{1}{m} \sum_{j \in m} \mathbf{A}_{i,j} \\ \delta &= \sqrt{\frac{1}{m} \sum_{j \in [m]} (\mathbf{A}_{i,j} - \mu)^2} \end{cases}. \tag{12}
$$

We consider a $L$-layer and $H$-head decoder-only transformer model as our Representation-Learner $h(\cdot)$, which mainly consists of one Embedding-layer and $L$ layer transformer-decoder-block. We use $d$ to denote the model dimension, $d_k = d/H$ denotes the attention dimension, and $d_f = 4d$ denotes the feed-forward dimension throughout the paper. Given a token sequence $\mathbf{Z} \in \mathbb{R}^{m \times n_v}$ as input matrix, the $l$th layer's output is:

$$
\mathbf{Z}^l = \begin{cases} \text{Embedding}(\mathbf{Z}), & l = 0 \\ \text{TF}_{\mathcal{W}^l}(\mathbf{Z}^{l-1}), & l \in [L] \end{cases}, \tag{13}
$$

here $\text{Embedding}(\mathbf{Z}) = \mathbf{Z} \mathbf{W}_e + \mathbf{W}_p$, where $\mathbf{W}_e \in \mathbb{R}^{n_v \times d}$ denotes the token-embedding matrix, $\mathbf{W}_p \in \mathbb{R}^{m \times d}$ denotes the position-embedding matrix. Note that matrices $\mathbf{W}_e$ and $\mathbf{W}_p$ are learnable, but we can also directly use the pre-trained $\mathbf{W}_e$ and calculate $\mathbf{W}_p$ using sine and cosine functions, the specific calculation method can be found in ($Vaswani\,et\,al.$, 2017). To simplify our analysis, we choose the latter.
$\text{TF}_{\mathcal{W}^l}(\cdot)$ denotes the $l$-th layer transformer-decoder-block with

$$
\mathcal{W}^l = \left\{ \begin{matrix} \mathbf{W}_{\text{F1}}^l, \mathbf{W}_{\text{F2}}^l, \left\{ \mathbf{W}_{O_h}^l \right\}_{h=1}^H, \left\{ \mathbf{W}_{Q_h}^l \right\}_{h=1}^H, \left\{ \mathbf{W}_{K_h}^l \right\}_{h=1}^H, \left\{ \mathbf{W}_{V_h}^l \right\}_{h=1}^H \\ \in \mathbb{R}^{d \times d_f}, \mathbb{R}^{d_f \times d}, \mathbb{R}^{d \times d}, \mathbb{R}^{d \times d_k}, \mathbb{R}^{d \times d_k}, \mathbb{R}^{d \times d} \end{matrix} \right\} \tag{14}
$$

as parameters, which can be defined as:

$$\text{TF}_{\mathcal{W}^l}\left(\mathbf{Z}^{l-1}\right) = \Pi_{\text{norm}}\left(\text{FFN}\left(\mathbf{Y}^l\right)\right),$$
$$\mathbf{Y}^l = \Pi_{\text{norm}}\left(\text{MHA}\left(\mathbf{Z}^{l-1}\right)\right), \tag{15}$$

where $\text{FFN}\left(\cdot\right)$ denotes the Feed-Forward Neural Network with Residual Connections:

$$\text{FFN}\left(\mathbf{Y}^l\right) = \sigma\left(\mathbf{Y}^l \mathbf{W}_{\text{F1}}^l\right)\mathbf{W}_{\text{F2}}^l + \mathbf{Y}^l, \tag{16}$$

where $\sigma\left(\cdot\right)$ denotes the activation function, and we use ReLU throughout the paper. $\text{MHA}\left(\cdot\right)$ denotes the Masked-Mutil-Head-Attention with Residual Connections:

$$\text{MHA}\left(\mathbf{Z}^{l-1}\right) = \sum_{h \in [H]} \mathbf{A}_h^l\left(\mathbf{Z}^{l-1}\right)\mathbf{W}_{O_h}^l + \mathbf{Z}^{l-1}, \tag{17}$$

here $\mathbf{A}_h^l\left(\cdot\right)$ denotes the Self-Attention head:

$$\mathbf{A}_h^l\left(\mathbf{Z}^{l-1}\right) = \text{softmax}\left(\frac{\mathbf{Q}_h^l\left(\mathbf{K}_h^l\right)^\top + \mathbf{M}}{\sqrt{d_k}}\right)\mathbf{V}_h^l, \tag{18}$$

where $\mathbf{Q}_h^l = \mathbf{Z}^{l-1}\mathbf{W}_{Q_h}^l$, $\mathbf{K}_h^l = \mathbf{Z}^{l-1}\mathbf{W}_{K_h}^l$, $\mathbf{V}_h^l = \mathbf{Z}^{l-1}\mathbf{W}_{V_h}^l$ denote $Q$, $K$, $V$ matrix respectively, and

$$\mathbf{M} = \begin{pmatrix} 0 & -\infty & -\infty & \cdots & -\infty \\ 0 & 0 & -\infty & \cdots & -\infty \\ \vdots & \vdots & \vdots & \ddots & \vdots \\ 0 & 0 & 0 & \cdots & 0 \end{pmatrix} \in \mathbb{R}^{m \times m} \tag{19}$$

is a given mask matrix.

Therefore the Representation-Learner can be defined as $h(\mathbf{X}) = \mathbf{Z}^L$. And the hypothesis class of $h(\mathbf{X})$ is defined as:

$$\mathcal{H} = \left\{ \begin{matrix} \mathbf{X} \mapsto TF_{\mathbf{W}^L}\left(TF_{\mathbf{W}^{L-1}}\ldots TF_{\mathbf{W}^1}\left(\text{Embedding}\left(\mathbf{X}\right)\right)\right): \\ \left\|\mathbf{W}_{\text{F1}}^l\right\|_F, \left\|\mathbf{W}_{\text{F2}}^l\right\|_F, \left\|\mathbf{W}_{O_h}^l\right\|_F, \left\|\mathbf{W}_{Q_h}^l\right\|_F, \left\|\mathbf{W}_{K_h}^l\right\|_F, \left\|\mathbf{W}_{V_h}^l\right\|_F \leq B_l, \forall l \in [L], \forall h \in [H] \end{matrix} \right\}. \tag{20}$$

The Token-Predictor has many options, there we use a simple linear projection layer and $\text{softmax}$:

$$g\left(h\left(\mathbf{X}\right)\right) = \text{softmax}(\mathbf{Z}^L \mathbf{W}^P), \tag{21}$$

where $\mathbf{W}^P \in \mathbb{R}^{d \times n_v}$.

**Assumption C.1.** Assume that

- $\Pi_{\text{norm}}$ is $G_\pi$-Lipschitz with the $\ell_2$-norm, i.e., $\forall \mathbf{t}_1, \mathbf{t}_2 \in \mathbb{R}^d$,

$$\left\|\Pi_{\text{norm}}\left(\mathbf{t}_1\right) - \Pi_{\text{norm}}\left(\mathbf{t}_2\right)\right\|_{\ell_2} \leq G_\pi \|\mathbf{t}_1 - \mathbf{t}_2\|_{\ell_2}.$$

- $\forall l \in [L]$ and $h \in [H]$, there exists constants $C_l$ such that

$$\left\|\mathbf{Q}_h^l(\mathbf{K}_h^l)^\top / \sqrt{d_k}\right\|_{\ell_\infty} \leq C_l.$$

- $\forall l \in [L]$, $\mathbf{W}^l \in \mathcal{W}^l$, there exists constants $B_l$ satisfying

$$\|\mathbf{W}^l\|_F \leq B_l.$$

## C.2. Useful Lemmas

Here, we use the covering number to bound Rademacher complexity, so we first introduce the definition of the covering number and provide some useful lemmas and propositions.

**Definition C.2** ($\epsilon$-cover and covering numbe). Denote $(U, \|\cdot\|)$ as a normed space and $V \subseteq U$. For any $\epsilon > 0$, $V$ is called an $\epsilon$-cover of $U$ if for any $u \in U$, there exists $v \in V$ such that $\|u - v\| \leq \epsilon$. The covering number of the normed space $(U, \|\cdot\|)$ is the cardinality of the smallest $\epsilon$-cover, which is defined by $\mathcal{N}(U, \epsilon, \|\cdot\|) := \min\{|V| : V \text{ is an } \epsilon\text{-cover of } U\}$.

**Lemma C.3** (Lemma 9 of Lin & Zhang (2019)). *Let $W := \{w : w \in \mathbb{R}^d, \|w\|_2 \leq a\}$, then for any $\epsilon > 0$, we have*

$$\ln \mathcal{N}(W, \epsilon, \|\cdot\|_2) \leq d \ln\left(1 + \frac{2a}{\epsilon}\right).$$

**Lemma C.4** (Lemma 10 of Lin & Zhang (2019)). *Let $\mathbf{X} \in \mathbb{R}^{n \times d_{in}}$ be a given input matrix with bounded F-norm, and $\mathbf{W} \in \mathbb{R}^{d_{in} \times d_{out}}$ satisfying $\|\mathbf{W}\|_F \leq a$, then*

$$\ln \mathcal{N}(\{\mathbf{XW} : \|\mathbf{W}\|_F \leq a\}, \epsilon, \|\cdot\|_F) \leq d_{in} d_{out} \ln\left(1 + \frac{2a\|\mathbf{X}\|_F}{\epsilon}\right).$$

*Proof.* Let $\hat{\mathbf{W}}$ be the $\epsilon$-cover of $\{\mathbf{W} : \|\mathbf{W}\|_F \leq a\}$ such that $\|\mathbf{W} - \hat{\mathbf{W}}\|_F \leq \epsilon$, then

$$\|\mathbf{XW} - \mathbf{X}\hat{\mathbf{W}}\|_F \leq \|\mathbf{X}\|_F \|\mathbf{W} - \hat{\mathbf{W}}\|_F \leq \epsilon \|\mathbf{X}\|_F.$$

This means that any $\epsilon$-cover of $\{\mathbf{W} : \|\mathbf{W}\|_F \leq a\}$ is also an $\epsilon\|\mathbf{X}\|_F$-cover for $\{\mathbf{XW} : \|\mathbf{W}\|_F \leq a\}$, we have

$$\ln \mathcal{N}(\{\mathbf{XW} : \|\mathbf{W}\|_F \leq a\}, \epsilon, \|\cdot\|_F) \leq \ln \mathcal{N}\left(\{\mathbf{W} : \|\mathbf{W}\|_F \leq a\}, \frac{\epsilon}{\|\mathbf{X}\|_F}, \|\cdot\|_F\right).$$

We denote $\bar{\mathbf{W}} \in \mathbb{R}^{d_{in} d_{out}}$ as the one dimensional vector which is obtained by reshaping $\mathbf{W}$. Then by Lemma C.3, we have

$$\ln \mathcal{N}\left(\{\mathbf{W} : \|\mathbf{W}\|_F \leq a\}, \frac{\epsilon}{\|\mathbf{X}\|_F}, \|\cdot\|_F\right) \leq \ln \mathcal{N}\left(\{\bar{\mathbf{W}} : \|\bar{\mathbf{W}}\|_2 \leq a\}, \frac{\epsilon}{\|\mathbf{X}\|_F}, \|\cdot\|_F\right)$$
$$\leq d_{in} d_{out} \ln\left(1 + \frac{2a\|\mathbf{X}\|_F}{\epsilon}\right).$$

$\square$

**Lemma C.5** (Extension of (Bartlett et al., 2017)). *Let $\mathcal{F}$ be a real-valued function class taking values in $[0, c]$, and assume that $\mathbf{0} \in \mathcal{F}$. Then the empirical Rademacher complexity of $\mathcal{F}$ can be bounded as:*

$$\hat{\mathfrak{R}}_S(\mathcal{F}) \leq \inf_{\alpha > 0}\left(\frac{4\alpha}{\sqrt{n}} + \frac{12}{n}\int_\alpha^{c\sqrt{n}} \sqrt{\ln \mathcal{N}(\mathcal{F}_{|S}, \epsilon, \|\cdot\|_2)} d\epsilon\right),$$

*where $S$ represents the dataset containing $n$ samples.*

**Lemma C.6.** *Assume that $f(x)$ is a continouous function on $[a, b]$ satisfying $f(x) > 0$, and $g(x)$ is a continouous concave(downward) function on the range of $f(x)$. Then we have:*

$$\frac{1}{b-a}\int_a^b g(f(x))dx \leq g\left(\frac{1}{b-a}\int_a^b f(x)dx\right).$$

*Proof.* We divide the interval $[a, b]$ into $n$ eaqual parts, let $x_i = a + \frac{i}{n}(b-a)(i = 0, 1, 2, \cdots, n)$, then $\Delta_i = x_i - x_{i-1} = \frac{b-a}{n}(i = 1, 2, \cdots, n)$. Since $g(x)$ is a concave function, we can use Jensen's inequality to get:

$$\frac{1}{b-a}\sum_{i=1}^n g(f(x_i))\Delta_i = \sum_{i=1}^n \frac{1}{n}g(f(x_i)) \leq g\left(\sum_{i=1}^n \frac{1}{n}f(x_i)\right) = g\left(\frac{1}{b-a}\sum_{i=1}^n f(x_i)\Delta_i\right).$$

Combining the integrability of continuous functions and the definition of integral, let $n \to \infty$ in the above formula, we can get the result. $\square$

**Lemma C.7.** *Let* $\Pi_{norm}(\cdot)$ *be the layer normlization operator defined in (12), for any matrix* $\mathbf{X} \in \mathbb{R}^{m \times d}$*, we have* $\|\Pi_{norm}(\mathbf{X})\|_F \leq \sqrt{md}.$

*Proof.*

$$\|\Pi_{\mathrm{norm}}(\mathbf{X})\|_F = \sqrt{\sum_{i \in [m]} \sum_{j \in [d]} \frac{\mathbf{X}_{i,j}^2}{\xi^2 + \frac{1}{d} \sum_{j' \in [d]} \mathbf{X}_{i,j'}^2}}$$

$$\leq \sqrt{\sum_{i \in [m]} \sum_{j \in [d]} \frac{\mathbf{X}_{i,j}^2}{\frac{1}{d} \sum_{j' \in [d]} \mathbf{X}_{i,j'}^2}}$$

$$= \sqrt{md}$$

$\square$

**Lemma C.8.** *Let* $\mathrm{softmax}(\cdot)$ *be the row-wise softmax function defined as (11), for any matrix* $\mathbf{X} \in \mathbb{R}^{m \times m}$ *obeying* $\|\mathbf{X}\|_{\ell_\infty} \leq C$*, we have:*

$$\|\mathrm{softmax}(\mathbf{X})\|_F \leq e^C \text{ and } \|\mathrm{softmax}(\mathbf{X} + \mathbf{M})\|_F \leq e^C \sqrt{\ln m},$$

*where* $\mathbf{M}$ *is the given mask matrix defined in (19).*

*Proof.* Denote $\mathbf{X} = (x_{ij})_{m \times m}$ and $\mathrm{softmax}(\mathbf{X}) = (y_{ij})_{m \times m}$, we have:

$$y_{ij} = \frac{e^{x_{ij}}}{\sum_{j' \in [m]} e^{x_{ij'}}} \leq \frac{e^C}{me^{-C}} = \frac{e^{2C}}{m}.$$

Then we can get:

$$\|\mathrm{softmax}(\mathbf{X})\|_F = \sqrt{\sum_{i \in [m]} \sum_{j \in [m]} y_{ij}^2} \leq \sqrt{\sum_{i \in [m]} \sum_{j \in [m]} y_{ij} \frac{e^{2C}}{m}} = e^C,$$

here we uses $\sum_{j \in [m]} y_{ij} = 1, \forall i \in [m]$. When adding the mask matrix $\mathbf{M}$, we have:

$$\mathbf{X} + \mathbf{M} = \begin{pmatrix} x_{11} & -\infty & -\infty & \cdots & -\infty \\ x_{21} & x_{22} & -\infty & \cdots & -\infty \\ \vdots & \vdots & \vdots & \ddots & \vdots \\ x_{m1} & x_{m2} & x_{m3} & \cdots & x_{mm} \end{pmatrix}, \quad \mathrm{softmax}(\mathbf{X} + \mathbf{M}) = \begin{pmatrix} y'_{11} & 0 & 0 & \cdots & 0 \\ y'_{21} & y'_{22} & 0 & \cdots & 0 \\ \vdots & \vdots & \vdots & \ddots & \vdots \\ y'_{m1} & y'_{m2} & y'_{m3} & \cdots & y'_{mm} \end{pmatrix},$$

where

$$y'_{kj} = \begin{cases} \frac{e^{x_{kj}}}{\sum_{j' \in [k]} e^{x_{kj'}}} & , j \leq k \\ 0 & , j > k \end{cases}.$$

Similarly we can get:

$$\|\mathrm{softmax}(\mathbf{X} + \mathbf{M})\|_F \leq \sqrt{\sum_{k \in [m]} \sum_{j \in [k]} y'_{kj} \frac{e^{2C}}{k}} = e^C \sqrt{\sum_{k \in [m]} \frac{1}{k}} \lesssim e^C \sqrt{\ln m}.$$

Here we use the fact: $(1 + \frac{1}{2} + \cdots + \frac{1}{m} - \ln m) \to \gamma$, where $\gamma \approx 0.577218$ called Euler constant. $\square$

**Lemma C.9.** *The* $\mathrm{softmax}$ *is* $G_s - Lipschitz$ *in the* $\ell_2 - norm$*, and* $G_s \leq \frac{4\sqrt{3}}{9}$*, which means for any* $\mathbf{x}, \mathbf{y} \in \mathbb{R}^m$*, we have:*

$$\|\mathrm{softmax}(\mathbf{x}) - \mathrm{softmax}(\mathbf{y})\|_{\ell_2} \leq \frac{4\sqrt{3}}{9} \|\mathbf{x} - \mathbf{y}\|_{\ell_2}.$$

*Proof.* Let $e_j$ denote the $j$th element of $\mathrm{softmax}(\mathbf{x})$, the Jacobian satisfies:

$$\|J(\mathbf{x})\|_F = \left\|\mathrm{diag}(\mathrm{softmax}(\mathbf{x})) - \mathrm{softmax}(\mathbf{x})\,\mathrm{softmax}(\mathbf{x})^\top\right\|_F$$

$$= \sqrt{\sum_{i\in[m]}\sum_{j\in[m]} \left(e_i\left(\mathbb{I}\left[i=j\right] - e_j\right)\right)^2}$$

$$= \sqrt{\sum_{i\in[m]} e_i^2 \left(1 - e_i + \sum_{j\neq i} e_j\right)^2}$$

$$= \sqrt{\sum_{i\in[m]} 4e_i^2 \left(1 - e_i\right)^2}$$

$$= \sqrt{\sum_{i\in[m]} 4e_i \left(e_i^3 - 2e_i^2 + e_i\right)}$$

$$\leq \sqrt{\frac{16}{27}\sum_{i\in[m]} e_i}$$

$$= \frac{4\sqrt{3}}{9},$$

where the last inequality uses the fact: $x^3 - 2x^2 + x \leq \frac{4}{9}, x \in [0,1]$.

Denote $\mathbf{z} = \mathbf{y} - \mathbf{x}$, according to the definition of derivative, we have:

$$\lim_{\delta\to 0} \frac{\mathrm{softmax}(\mathbf{x} + \delta\mathbf{z}) - \mathrm{softmax}(\mathbf{x})}{\delta} = J(\mathbf{x})\mathbf{z}.$$

Integrating along $\delta = 0$ to $1$ under $\|J(\mathbf{x})\mathbf{z}\|_{\ell_2} \leq \frac{4\sqrt{3}}{9}\|\mathbf{z}\|_{\ell_2}$ can obtain the result. $\qquad\square$

### C.3. Proof of Proposition 4.17

**Lemma C.10** (Covering number of masked self-attention head). *Given an input sequence $S = \{\mathbf{Z}_1, \ldots, \mathbf{Z}_N\} \in \mathbb{R}^{m\times d}$, denote $\mathbf{Z}_{[N]} = [\mathbf{Z}_1, \ldots, \mathbf{Z}_N] \in \mathbb{R}^{Nm\times d}$ as the concatenated data matrix. Consider the Self-Attention head $\mathbf{A}(\cdot)$ (ignore the layer and head indices) defined in Equation (18), the corresponding function class can be defined as:*

$$\mathcal{H}_S^{\mathbf{A}} := \left\{ \mathbf{A} = \begin{pmatrix} \mathbf{A}(\mathbf{Z}_1) \\ \vdots \\ \mathbf{A}(\mathbf{Z}_N) \end{pmatrix} : \mathbf{A}(\mathbf{Z}_i) = \mathrm{softmax}\left(\frac{\mathbf{Z}_i\mathbf{W}_Q\left(\mathbf{Z}_i\mathbf{W}_K\right)^\top + \mathbf{M}}{\sqrt{d_k}}\right)\mathbf{Z}_i\mathbf{W}_V \atop : \|\mathbf{W}_Q\|_F, \|\mathbf{W}_K\|_F, \|\mathbf{W}_V\|_F \leq B \right\},$$

*then we can get the following covering number bound:*

$$\ln\mathcal{N}\left(\mathcal{H}_S^{\mathbf{A}}, \epsilon, \|\cdot\|_F\right) \leq d^2\ln\left(1 + \frac{2e^C B\sqrt{N\ln m}\|\mathbf{Z}_{[N]}\|_F}{\epsilon}\right) + 2dd_k\ln\left(1 + \frac{8G_s B^3\|\mathbf{Z}^*\|_F\|\mathbf{Z}_{[N]}\|_F^2}{\sqrt{d_k}\epsilon}\right).$$

*Proof.* **Step 1:** Denote $\mathcal{C}_V$ as a $\epsilon_V$-cover of set $\mathcal{H}_S^{\mathbf{V}} := \{\mathbf{V} = \mathbf{Z}_{[N]}\mathbf{W}_V : \|\mathbf{W}_V\|_F \leq B\}$, then by Lemma B.6 we have:

$$\ln\mathcal{N}\left(\mathcal{H}_S^{\mathbf{V}}, \epsilon_V, \|\cdot\|_F\right) \leq d^2\ln\left(1 + \frac{2B\|\mathbf{Z}_{[N]}\|_F}{\epsilon_V}\right).$$

**Step 2:** Let $\mathcal{C}_Q$ to be a $\epsilon_Q$-cover of set $\mathcal{H}_{\mathbf{Z}}^{\mathbf{Q}} := \{\mathbf{Q} = \mathbf{Z}\mathbf{W}_Q : \|\mathbf{W}_Q\|_F \leq B\}$, and $\mathcal{C}_K$ to be a $\epsilon_K$-cover of set $\mathcal{H}_{\mathbf{Z}}^{\mathbf{K}} := \{\mathbf{K} = \mathbf{Z}\mathbf{W}_K : \|\mathbf{W}_K\|_F \leq B\}$. We can use $\mathcal{C}_Q$ and $\mathcal{C}_K$ to construct a set as following:

$$\mathcal{C}_{\mathbf{S}} := \left\{ \mathbf{S} = \begin{pmatrix} \mathbf{S}_1 & & \mathbf{0} \\ & \ddots & \\ \mathbf{0} & & \mathbf{S}_N \end{pmatrix} : \mathbf{S}_i = \mathrm{softmax}\left(\frac{\mathbf{Z}_i\mathbf{W}_Q\left(\mathbf{Z}_i\mathbf{W}_K\right)^\top + \mathbf{M}}{\sqrt{d_k}}\right) : \mathbf{W}_Q \in \mathcal{C}_Q, \mathbf{W}_K \in \mathcal{C}_K \right\}.$$

We consider the following set:

$$\mathcal{H}_S^{\mathbf{S}} := \left\{ \mathbf{S} = \begin{pmatrix} \mathbf{S}_1 & & \mathbf{0} \\ & \ddots & \\ \mathbf{0} & & \mathbf{S}_N \end{pmatrix} : \mathbf{S}_i = \mathrm{softmax}\left( \frac{\mathbf{Z}_i \mathbf{W}_Q \left(\mathbf{Z}_i \mathbf{W}_K\right)^\top + \mathbf{M}}{\sqrt{d_k}} \right) : \|\mathbf{W}_Q\|_F, \|\mathbf{W}_K\|_F \leq B \right\},$$

then for $\forall \epsilon > 0$ and any $\mathbf{S} \in \mathcal{H}_S^{\mathbf{S}}$, there exists $\hat{\mathbf{S}} \in \mathcal{C}_{\mathbf{S}}$ such that:

$$\left\| \mathbf{S} - \hat{\mathbf{S}} \right\|_F = \left\| \begin{pmatrix} \mathbf{S}_1 & & \mathbf{0} \\ & \ddots & \\ \mathbf{0} & & \mathbf{S}_N \end{pmatrix} - \begin{pmatrix} \hat{\mathbf{S}}_1 & & \mathbf{0} \\ & \ddots & \\ \mathbf{0} & & \hat{\mathbf{S}}_N \end{pmatrix} \right\|_F$$

$$= \sqrt{ \sum_{i \in [N]} \left\| \mathrm{softmax}\left( \frac{\mathbf{Z}_i \mathbf{W}_Q \left(\mathbf{Z}_i \mathbf{W}_K\right)^\top + \mathbf{M}}{\sqrt{d_k}} \right) - \mathrm{softmax}\left( \frac{\mathbf{Z}_i \hat{\mathbf{W}}_Q \left(\mathbf{Z}_i \hat{\mathbf{W}}_K\right)^\top + \mathbf{M}}{\sqrt{d_k}} \right) \right\|_F^2 }$$

$$\overset{(i)}{\leq} \frac{G_s}{\sqrt{d_k}} \sqrt{ \sum_{i \in [N]} \left\| \mathbf{Z}_i \mathbf{W}_Q \left(\mathbf{Z}_i \mathbf{W}_K\right)^\top - \mathbf{Z}_i \hat{\mathbf{W}}_Q \left(\mathbf{Z}_i \hat{\mathbf{W}}_K\right)^\top \right\|_F^2 }$$

$$\leq \frac{G_s}{\sqrt{d_k}} \sqrt{ \sum_{i \in [N]} \left( \left\| \left(\mathbf{Z}_i \mathbf{W}_Q - \mathbf{Z}_i \hat{\mathbf{W}}_Q\right) \left(\mathbf{Z}_i \mathbf{W}_K\right)^\top \right\|_F + \left\| \mathbf{Z}_i \hat{\mathbf{W}}_Q \left[ \left(\mathbf{Z}_i \mathbf{W}_K\right)^\top - \left(\mathbf{Z}_i \hat{\mathbf{W}}_K\right)^\top \right] \right\|_F \right)^2 }$$

$$\leq \frac{G_s B}{\sqrt{d_k}} \left(\epsilon_Q + \epsilon_K\right) \sqrt{ \sum_{i \in [N]} \|\mathbf{Z}_i\|_F^2 }$$

$$\leq \frac{G_s B}{\sqrt{d_k}} \left(\epsilon_Q + \epsilon_K\right) \|\mathbf{Z}_{[N]}\|_F$$

$$\overset{(ii)}{=} \epsilon.$$

Evoking Lemma C.9 we have $(i)$, and by setting $\epsilon_Q = \epsilon_K = \frac{\sqrt{d_k}}{2 G_s B \|\mathbf{Z}_{[N]}\|_F} \epsilon$ can get $(ii)$. Therefore $\mathcal{C}_{\mathbf{S}}$ is a cover of $\mathcal{H}_S^{\mathbf{S}}$, then by Lemma C.4 we have:

$$\ln \mathcal{N}\left(\mathcal{H}_S^{\mathbf{S}}, \epsilon, \|\cdot\|_F\right) \leq \ln |\mathcal{C}_{\mathbf{S}}| \leq \ln |\mathcal{C}_Q| + \ln |\mathcal{C}_K| \leq 2 d d_k \ln \left( 1 + \frac{4 G_s B^2 \|\mathbf{Z}^*\|_F \|\mathbf{Z}_{[N]}\|_F}{\sqrt{d_k} \epsilon} \right),$$

where $\|\mathbf{Z}^*\|_F = \max_{i \in [N]} \|\mathbf{Z}_i\|_F$.

**Step 3:** For every given $\hat{\mathbf{V}} \in \mathcal{C}_V$, we can construct the set $\mathcal{H}_S^{\mathbf{S}} \circ \hat{\mathbf{V}} := \left\{ \mathbf{S}\hat{\mathbf{V}} : \mathbf{S} \in \mathcal{H}_S^{\mathbf{S}} \right\}$ and $\mathcal{C}_{\mathbf{S}} \circ \hat{\mathbf{V}} := \left\{ \mathbf{S}\hat{\mathbf{V}} : \mathbf{S} \in \mathcal{C}_{\mathbf{S}} \right\}$, we denote $\mathcal{C}\left(\mathcal{C}_{\mathbf{S}} \circ \hat{\mathbf{V}}, \epsilon_S, \|\cdot\|_F\right)$ as a $\epsilon_S$-cover of $\mathcal{C}_{\mathbf{S}} \circ \hat{\mathbf{V}}$. Then for any $\mathbf{S}\hat{\mathbf{V}} \in \mathcal{H}_S^{\mathbf{S}} \circ \hat{\mathbf{V}}$, we can find $\hat{\mathbf{S}}\hat{\mathbf{V}} \in \mathcal{C}\left(\mathcal{C}_{\mathbf{S}} \circ \hat{\mathbf{V}}, \epsilon_S, \|\cdot\|_F\right)$ such that:

$$\left\| \mathbf{S}\hat{\mathbf{V}} - \hat{\mathbf{S}}\hat{\mathbf{V}} \right\|_F \leq \left\| \mathbf{S} - \hat{\mathbf{S}} \right\|_F \|\hat{\mathbf{V}}\|_F \leq \epsilon_S \|\mathbf{Z}_{[N]}\|_F B.$$

We can Choose $\epsilon_S = \frac{\epsilon_A}{\|\mathbf{Z}_{[N]}\|_F B}$ to get that $\mathcal{C}\left(\mathcal{C}_{\mathbf{S}} \circ \hat{\mathbf{V}}, \epsilon_S, \|\cdot\|_F\right)$ is a $\epsilon_A$-cover of $\mathcal{H}_S^{\mathbf{S}} \circ \hat{\mathbf{V}}$ which can be denoted as $\mathcal{C}\left(\mathcal{H}_S^{\mathbf{S}} \circ \hat{\mathbf{V}}, \epsilon_A, \|\cdot\|_F\right)$. Then we have:

$$\sup_{\hat{\mathbf{V}} \in \mathcal{C}_V} \ln \left| \mathcal{C}\left(\mathcal{H}_S^{\mathbf{S}} \circ \hat{\mathbf{V}}, \epsilon_A, \|\cdot\|_F\right) \right| \leq \sup_{\hat{\mathbf{V}} \in \mathcal{C}_V} \ln \left| \mathcal{C}\left(\mathcal{C}_{\mathbf{S}} \circ \hat{\mathbf{V}}, \epsilon_S, \|\cdot\|_F\right) \right| \leq 2 d d_k \ln \left( 1 + \frac{4 G_s B^3 \|\mathbf{Z}^*\|_F \|\mathbf{Z}_{[N]}\|_F^2}{\sqrt{d_k} \epsilon_A} \right).$$

Then we construct a set $\mathcal{C}_A$:

$$\mathcal{C}_A = \bigcup_{\hat{\mathbf{V}} \in \mathcal{C}_V} \mathcal{C}\left(\mathcal{H}_S^{\mathbf{S}} \circ \hat{\mathbf{V}}, \epsilon_A, \|\cdot\|_F\right),$$

which is easy to get:

$$\ln|\mathcal{C}_A| \le \ln|\mathcal{C}_V| + \sup_{\hat{\mathbf{V}} \in \mathcal{C}_V} \ln\left|\mathcal{C}\left(\mathcal{H}_S^{\mathbf{S}} \circ \hat{\mathbf{V}}, \epsilon_A, \|\cdot\|_F\right)\right|$$

$$\le d^2 \ln\left(1 + \frac{2B\|\mathbf{Z}_{[N]}\|_F}{\epsilon_V}\right) + 2dd_k \ln\left(1 + \frac{4G_s B^3 \|\mathbf{Z}^*\|_F \|\mathbf{Z}_{[N]}\|_F^2}{\sqrt{d_k}\epsilon_A}\right).$$

**Step 4:** Next we will proof that $\mathcal{C}_A$ covers $\mathcal{H}_S^{\mathbf{A}}$. For any $\mathbf{A} \in \mathcal{H}_S^{\mathbf{A}}$, we can find $\hat{\mathbf{A}} \in \mathcal{C}_A$ such that:

$$\begin{aligned}
\left\|\mathbf{A} - \hat{\mathbf{A}}\right\|_F &= \left\|\mathbf{SV} - \hat{\mathbf{S}}\hat{\mathbf{V}}\right\|_F \\
&\le \left\|\mathbf{SV} - \mathbf{S}\hat{\mathbf{V}}\right\|_F + \left\|\mathbf{S}\hat{\mathbf{V}} - \hat{\mathbf{S}}\hat{\mathbf{V}}\right\|_F \\
&\le \|\mathbf{S}\|_F \left\|\mathbf{V} - \hat{\mathbf{V}}\right\|_F + \epsilon_A \\
&= \sqrt{\sum_{i\in[N]} \|\mathbf{S}_i\|_F^2} \epsilon_V + \epsilon_A \\
&\le \sqrt{N \ln m} e^C \epsilon_V + \epsilon_A,
\end{aligned}$$

where the last inequality uses Lemma C.8. Then by setting $\epsilon_V = \frac{\epsilon}{2e^C\sqrt{N\ln m}}$, $\epsilon_A = \frac{\epsilon}{2}$, we can get:

$$\ln\mathcal{N}\left(\mathcal{H}_S^{\mathbf{A}}, \epsilon, \|\cdot\|_F\right) \le \ln|\mathcal{C}_A|$$

$$\le d^2 \ln\left(1 + \frac{2e^C B\sqrt{N\ln m}\|\mathbf{Z}_{[N]}\|_F}{\epsilon}\right) + 2dd_k \ln\left(1 + \frac{8G_s B^3 \|\mathbf{Z}^*\|_F \|\mathbf{Z}_{[N]}\|_F^2}{\sqrt{d_k}\epsilon}\right).$$

$$\square$$

**Proposition C.11** (Covering number of transformer-decoder-block). *Given an input sequence* $S = \{\mathbf{Z}_1, \ldots, \mathbf{Z}_N\} \in \mathbb{R}^{m \times d}$, *denote* $\mathbf{Z}_{[N]} = [\mathbf{Z}_1, \ldots, \mathbf{Z}_N] \in \mathbb{R}^{Nm \times d}$ *as the concatenated data matrix. Consider the transformer-decoder-block* $\mathrm{TF}(\cdot)$*(ignore the layer indices) defined in Equation* (15)*, the corresponding function class can be defined as:*

$$\mathcal{H}_S^{TF} := \left\{ \begin{aligned}
&\mathbf{Z} \mapsto \Pi_{norm}\left(\sigma\left(\mathbf{YW}_{\mathrm{F1}}\right)\mathbf{W}_{\mathrm{F2}} + \mathbf{Y}\right), \mathbf{Y} = \Pi_{norm}\left(\sum_{h\in H} \mathbf{A}_h(\mathbf{Z})\mathbf{W}_{O_h} + \mathbf{Z}\right), \\
&\mathbf{A}_h(\mathbf{Z}) = \mathrm{softmax}\left(\frac{\mathbf{ZW}_{Q_h}\left(\mathbf{ZW}_{K_h}\right)^\top + \mathbf{M}}{\sqrt{d_k}}\right)\mathbf{ZW}_{V_h} : \\
&\|\mathbf{W}_{\mathrm{F1}}\|_F, \|\mathbf{W}_{\mathrm{F2}}\|_F, \|\mathbf{W}_{O_h}\|_F, \|\mathbf{W}_{Q_h}\|_F, \|\mathbf{W}_{K_h}\|_F, \|\mathbf{W}_{V_h}\|_F \le B, \forall h \in [H]
\end{aligned} \right\},$$

*then we can get the following covering number bound:*

$$\ln\mathcal{N}\left(\mathcal{H}_S^{TF}, \epsilon, \|\cdot\|_F\right) \lesssim 4d^2(H+3)\ln\left(1 + \frac{G_\pi^2 B^2(B^2+1)\left(e^C B^2 H\sqrt{N\ln m} + 1\right)\|\mathbf{Z}_{[N]}\|_F}{\epsilon}\right).$$

*Proof.* **Step 1:** We first consider the function class of muti-head-attention $\mathrm{MHA}(\cdot)$, which can be defined as:

$$\mathcal{H}_S^{\mathbf{M_A}} := \left\{\mathbf{M_A} = \sum_{h\in H} \mathbf{A}_h \mathbf{W}_{O_h} + \mathbf{Z}_{[N]} : \mathbf{A}_h \in \mathcal{H}_S^{\mathbf{A}_h}, \|\mathbf{W}_{O_h}\|_F \le B, \forall h \in [H]\right\},$$

where

$$\mathcal{H}_S^{\mathbf{A}_h} := \left\{ \mathbf{A}_h = \begin{pmatrix} \mathbf{A}_h(\mathbf{Z}_1) \\ \vdots \\ \mathbf{A}_h(\mathbf{Z}_N) \end{pmatrix} : \mathbf{A}_h(\mathbf{Z}_i) = \mathrm{softmax}\left(\frac{\mathbf{Z}_i\mathbf{W}_{Q_h}\left(\mathbf{Z}_i\mathbf{W}_{K_h}\right)^\top + \mathbf{M}}{\sqrt{d_k}}\right)\mathbf{Z}_i\mathbf{W}_{V_h} \\ : \|\mathbf{W}_{Q_h}\|_F, \|\mathbf{W}_{K_h}\|_F, \|\mathbf{W}_{V_h}\|_F \le B \right\}.$$

For any $h \in [H]$, denote $\mathcal{C}_{A_h}$ as a $\epsilon_A$-cover of $\mathcal{H}_S^{\mathbf{A}_h}$, we select one element $\hat{\mathbf{A}}_h \in \mathcal{C}_{A_h}$ to construct the following set:

$$\hat{\mathbf{A}}_h \circ \mathbf{W}_{O_h} = \left\{ \hat{\mathbf{A}}_h \mathbf{W}_{O_h} : \|\mathbf{W}_{O_h}\|_F \leq B \right\},$$

let $\mathcal{C}\left( \hat{\mathbf{A}}_h \circ \mathbf{W}_{O_h}, \epsilon_d, \|\cdot\|_F \right)$ $\epsilon_d$-covers $\hat{\mathbf{A}}_h \circ \mathbf{W}_{O_h}$, we have:

$$\ln \left| \mathcal{C}\left( \hat{\mathbf{A}}_h \circ \mathbf{W}_{O_h}, \epsilon_d, \|\cdot\|_F \right) \right| \leq \sup_{\hat{\mathbf{A}}_h \in \mathcal{C}_{A_h}} \ln \mathcal{N}\left( \hat{\mathbf{A}}_h \circ \mathbf{W}_{O_h}, \epsilon_d, \|\cdot\|_F \right)$$

$$\leq \sup_{\hat{\mathbf{A}}_h \in \mathcal{C}_{A_h}} d^2 \ln \left( 1 + \frac{2B\|\hat{\mathbf{A}}_h\|_F}{\epsilon_d} \right)$$

$$\leq d^2 \ln \left( 1 + \frac{2e^C B^2 \sqrt{N \ln m} \|Z_{[N]}\|_F}{\epsilon_d} \right).$$

Now we can construct the $\epsilon_d$-cover of set $\left\{ \mathbf{A}_h \mathbf{W}_{O_h} : \mathbf{A}_h \in \mathcal{H}_S^{\mathbf{A}_h}, \|\mathbf{W}_{O_h}\|_F \leq B \right\}$ as :

$$\mathcal{C}_{head_h} = \bigcup_{\hat{\mathbf{A}}_h \in \mathcal{C}_{A_h}} \mathcal{C}\left( \hat{\mathbf{A}}_h \circ \mathbf{W}_{O_h}, \epsilon_d, \|\cdot\|_F \right).$$

Combined with Lemma C.10, it is easy to get the following covering number bound:

$$\ln \mathcal{N}\left( \mathcal{H}_S^{\mathbf{M_A}}, \epsilon_d, \|\cdot\|_F \right) \leq H \ln |\mathcal{C}_{head_h}|$$

$$\leq H \left( \ln |\mathcal{C}_{A_h}| + \sup_{\hat{\mathbf{A}}_h \in \mathcal{C}_{A_h}} \ln \left| \mathcal{C}\left( \hat{\mathbf{A}}_h \circ \mathbf{W}_{O_h}, \epsilon_d, \|\cdot\|_F \right) \right| \right)$$

$$\leq d^2 H \ln \left( 1 + \frac{2e^C B \sqrt{N \ln m} \|\mathbf{Z}_{[N]}\|_F}{\epsilon_A} \right) + 2dd_k H \ln \left( 1 + \frac{8G_s B^3 \|\mathbf{Z}^*\|_F \|\mathbf{Z}_{[N]}\|_F^2}{\sqrt{d_k} \epsilon_A} \right)$$

$$+ d^2 H \ln \left( 1 + \frac{2e^C B^2 \sqrt{N \ln m} \|Z_{[N]}\|_F}{\epsilon_d} \right).$$

**Step 2:** Now, we consider the Feed-Forward Neural Network FFN $(\cdot)$ defined in (16), which consists of two fully-connected layers. The hypothesis class of fully-connected layer 1 can be defined as:

$$\mathcal{H}_S^{\mathbf{F_1}} = \left\{ \mathbf{Y}_{[N]} \mathbf{W_{F1}} : \mathbf{Y}_{[N]} = \Pi_{\text{norm}}\left( \mathbf{M_A} \right), \mathbf{M_A} \in \mathcal{H}_S^{\mathbf{M_A}}, \|\mathbf{W_{F1}}\|_F \leq B \right\}.$$

Denote $\mathcal{C}_M$ as a $\epsilon_d$-cover of $\mathcal{H}_S^{\mathbf{M_A}}$, we select one element $\hat{\mathbf{M}}_\mathbf{A} \in \mathcal{C}_M$ to construct the following set:

$$\hat{\mathbf{M}}_\mathbf{A} \circ \mathbf{W_{F1}} = \left\{ \Pi_{\text{norm}}\left( \hat{\mathbf{M}}_\mathbf{A} \right) \mathbf{W_{F1}} : \|\mathbf{W_{F1}}\|_F \leq B \right\},$$

let $\mathcal{C}\left( \hat{\mathbf{M}}_\mathbf{A} \circ \mathbf{W_{F1}}, \epsilon_{F1}, \|\cdot\|_F \right)$ $\epsilon_{F1}$-covers $\hat{\mathbf{M}}_\mathbf{A} \circ \mathbf{W_{F1}}$, we have:

$$\ln \left| \mathcal{C}\left( \hat{\mathbf{M}}_\mathbf{A} \circ \mathbf{W_{F1}}, \epsilon_{F1}, \|\cdot\|_F \right) \right| \leq \sup_{\hat{\mathbf{M}}_\mathbf{A} \in \mathcal{C}_M} \ln \mathcal{N}\left( \hat{\mathbf{M}}_\mathbf{A} \circ \mathbf{W_{F1}}, \epsilon_{F1}, \|\cdot\|_F \right)$$

$$\leq \sup_{\hat{\mathbf{M}}_\mathbf{A} \in \mathcal{C}_M} dd_f \ln \left( 1 + \frac{2B\|\Pi_{\text{norm}}\left( \hat{\mathbf{M}}_\mathbf{A} \right)\|_F}{\epsilon_{F1}} \right)$$

$$\leq \sup_{\hat{\mathbf{A}}_h \in \mathcal{C}_{A_h}} dd_f \ln \left( 1 + \frac{2BG_\pi \left( H\|\hat{\mathbf{A}}_h\|_F B + \|\mathbf{Z}_{[N]}\|_F \right)}{\epsilon_{F1}} \right)$$

$$\leq dd_f \ln \left( 1 + \frac{2BG_\pi \left( e^C B^2 H \sqrt{N \ln m} + 1 \right) \|\mathbf{Z}_{[N]}\|_F}{\epsilon_{F1}} \right).$$

Now we can construct the $\epsilon_{F1}$-cover of $\mathcal{H}_S^{\mathbf{F_1}}$ as:

$$\mathcal{C}_{F_1} = \bigcup_{\hat{\mathbf{M}}_{\mathbf{A}} \in \mathcal{C}_M} \mathcal{C}\left(\hat{\mathbf{M}}_{\mathbf{A}} \circ \mathbf{W}_{\mathbf{F1}}, \epsilon_{F1}, \|\cdot\|_F\right).$$

We have the following covering number bound:

$$\ln|\mathcal{C}_{F_1}| \le \ln|\mathcal{C}_M| + \sup_{\hat{\mathbf{M}}_{\mathbf{A}} \in \mathcal{C}_M} \ln\left|\mathcal{C}\left(\hat{\mathbf{M}}_{\mathbf{A}} \circ \mathbf{W}_{\mathbf{F1}}, \epsilon_{F1}, \|\cdot\|_F\right)\right|$$

$$\le d^2 H \ln\left(1 + \frac{2e^C B \sqrt{N \ln m}\|\mathbf{Z}_{[N]}\|_F}{\epsilon_A}\right) + 2dd_k H \ln\left(1 + \frac{8G_s B^3 \|\mathbf{Z}^*\|_F \|\mathbf{Z}_{[N]}\|_F^2}{\sqrt{d_k}\epsilon_A}\right)$$

$$+ d^2 H \ln\left(1 + \frac{2e^C B^2 \sqrt{N \ln m}\|Z_{[N]}\|_F}{\epsilon_d}\right) + dd_f \ln\left(1 + \frac{2BG_\pi\left(e^C B^2 H \sqrt{N \ln m} + 1\right)\|\mathbf{Z}_{[N]}\|_F}{\epsilon_{F1}}\right).$$

**Step 3:** Next, we consider the hypothesis class of fully-connected layer 2:

$$\mathcal{H}_S^{\mathbf{F_2}} = \left\{\sigma\left(\mathbf{F}_{[\mathbf{N}]}\right)\mathbf{W}_{\mathbf{F2}} + \mathbf{Y}_{[\mathbf{N}]} : \mathbf{F}_{[\mathbf{N}]} \in \mathcal{H}_S^{\mathbf{F_1}}, \mathbf{Y}_{[\mathbf{N}]} = \Pi_{\text{norm}}\left(\mathbf{M}_{\mathbf{A}}\right), \mathbf{M}_{\mathbf{A}} \in \mathcal{H}_S^{\mathbf{M_A}}, \|\mathbf{W}_{\mathbf{F2}}\|_F \le B\right\}.$$

For every element $\hat{\mathbf{F}}_{[N]} \in \mathcal{C}_{F_1}$ and $\hat{\mathbf{M}}_{\mathbf{A}} \in \mathcal{C}_M$, we construct the following set:

$$\hat{\mathbf{F}}_{[N]} \circ \mathbf{W}_{\mathbf{F1}} \circ \hat{\mathbf{M}}_{\mathbf{A}} = \left\{\sigma\left(\hat{\mathbf{F}}_{[N]}\right)\mathbf{W}_{\mathbf{F2}} + \Pi_{\text{norm}}\left(\hat{\mathbf{M}}_{\mathbf{A}}\right) : \|\mathbf{W}_{\mathbf{F2}}\|_F \le B\right\},$$

let $\mathcal{C}\left(\hat{\mathbf{F}}_{[N]} \circ \mathbf{W}_{\mathbf{F1}} \circ \hat{\mathbf{M}}_{\mathbf{A}}, \epsilon_{F2}, \|\cdot\|_F\right)$ $\epsilon_{F2}$-covers $\hat{\mathbf{F}}_{[N]} \circ \mathbf{W}_{\mathbf{F1}} \circ \hat{\mathbf{M}}_{\mathbf{A}}$, we have:

$$\ln\left|\mathcal{C}\left(\hat{\mathbf{F}}_{[N]} \circ \mathbf{W}_{\mathbf{F1}} \circ \hat{\mathbf{M}}_{\mathbf{A}}, \epsilon_{F2}, \|\cdot\|_F\right)\right| \le \sup_{\hat{\mathbf{F}}_{[N]} \in \mathcal{C}_{F_1}, \hat{\mathbf{M}}_{\mathbf{A}} \in \mathcal{C}_M} \ln\mathcal{N}\left(\hat{\mathbf{F}}_{[N]} \circ \mathbf{W}_{\mathbf{F1}} \circ \hat{\mathbf{M}}_{\mathbf{A}}, \epsilon_{F2}, \|\cdot\|_F\right)$$

$$= \sup_{\hat{\mathbf{F}}_{[N]} \in \mathcal{C}_{F_1}} \ln\mathcal{N}\left(\hat{\mathbf{F}}_{[N]} \circ \mathbf{W}_{\mathbf{F1}}, \epsilon_{F2}, \|\cdot\|_F\right)$$

$$\le \sup_{\hat{\mathbf{F}}_{[N]} \in \mathcal{C}_{F_1}} dd_f \ln\left(1 + \frac{2B\|\hat{\mathbf{F}}_{[N]}\|_F}{\epsilon_{F2}}\right)$$

$$\le \sup_{\hat{\mathbf{M}}_{\mathbf{A}} \in \mathcal{C}_M} dd_f \ln\left(1 + \frac{2B^2\|\Pi_{\text{norm}}\left(\hat{\mathbf{M}}_{\mathbf{A}}\right)\|_F}{\epsilon_{F2}}\right)$$

$$\le \sup_{\hat{\mathbf{A}}_h \in \mathcal{C}_{A_h}} dd_f \ln\left(1 + \frac{2B^2 G_\pi\left(H\|\hat{\mathbf{A}}_h\|_F B + \|\mathbf{Z}_{[N]}\|_F\right)}{\epsilon_{F2}}\right)$$

$$\le dd_f \ln\left(1 + \frac{2B^2 G_\pi\left(e^C B^2 H \sqrt{N \ln m} + 1\right)\|\mathbf{Z}_{[N]}\|_F}{\epsilon_{F2}}\right).$$

Now we can construct the $\epsilon_{F2}$-cover of $\mathcal{H}_S^{\mathbf{F_2}}$ as:

$$\mathcal{C}_{F_2} = \bigcup_{\hat{\mathbf{F}}_{[N]} \in \mathcal{C}_{F_1}, \hat{\mathbf{M}}_{\mathbf{A}} \in \mathcal{C}_M} \mathcal{C}\left(\hat{\mathbf{F}}_{[N]} \circ \mathbf{W}_{\mathbf{F1}} \circ \hat{\mathbf{M}}_{\mathbf{A}}, \epsilon_{F2}, \|\cdot\|_F\right).$$

We have the following covering number bound:

$$\ln |\mathcal{C}_{F_2}| \leq \ln |\mathcal{C}_{F_1}| + \ln |\mathcal{C}_M| + \sup_{\hat{\mathbf{F}}_{[N]} \in \mathcal{C}_{F_1}, \tilde{\mathbf{M}}_{\mathbf{A}} \in \mathcal{C}_M} \ln \left| \mathcal{C} \left( \hat{\mathbf{F}}_{[N]} \circ \mathbf{W}_{\mathbf{F1}} \circ \hat{\mathbf{M}}_{\mathbf{A}}, \epsilon_{F2}, \|\cdot\|_F \right) \right|$$

$$\leq 2d^2 H \ln \left( 1 + \frac{2e^C B \sqrt{N \ln m} \|\mathbf{Z}_{[N]}\|_F}{\epsilon_A} \right) + 4dd_k H \ln \left( 1 + \frac{8 G_s B^3 \|\mathbf{Z}^*\|_F \|\mathbf{Z}_{[N]}\|_F^2}{\sqrt{d_k} \epsilon_A} \right)$$

$$+ 2d^2 H \ln \left( 1 + \frac{2e^C B^2 \sqrt{N \ln m} \|Z_{[N]}\|_F}{\epsilon_d} \right) + dd_f \ln \left( 1 + \frac{2BG_\pi \left( e^C B^2 H \sqrt{N \ln m} + 1 \right) \|\mathbf{Z}_{[N]}\|_F}{\epsilon_{F1}} \right)$$

$$+ dd_f \ln \left( 1 + \frac{2B^2 G_\pi \left( e^C B^2 H \sqrt{N \ln m} + 1 \right) \|\mathbf{Z}_{[N]}\|_F}{\epsilon_{F2}} \right).$$

**Step 4:** To get the covering number of $\mathcal{H}_S^{TF}$, we construct the following set:

$$\mathcal{C}_T = \left\{ \Pi_{\text{norm}} \left( \mathbf{F_2} \right) : \mathbf{F_2} \in \mathcal{C}_{F_2} \right\},$$

which is an $\epsilon$-cover of $\mathcal{H}_S^{TF}$ that can be verified. For any $\mathbf{Z}_{[N]} \in \mathcal{H}_S^{TF}$, we can find a $\hat{\mathbf{Z}}_{[N]} \in \mathcal{C}_T$ such that:

$$\left\| \mathbf{Z}_{[N]} - \hat{\mathbf{Z}}_{[N]} \right\|_F = \left\| \Pi_{\text{norm}} \left( \sigma \left( \mathbf{Y}_{[N]} \mathbf{W}_{\text{F1}} \right) \mathbf{W}_{\text{F2}} + \mathbf{Y}_{[N]} \right) - \Pi_{\text{norm}} \left( \sigma \left( \hat{\mathbf{Y}}_{[N]} \hat{\mathbf{W}}_{\text{F1}} \right) \hat{\mathbf{W}}_{\text{F2}} + \hat{\mathbf{Y}}_{[N]} \right) \right\|_F$$

$$\leq G_\pi \left\| \sigma \left( \mathbf{Y}_{[N]} \mathbf{W}_{\text{F1}} \right) \mathbf{W}_{\text{F2}} - \sigma \left( \mathbf{Y}_{[N]} \mathbf{W}_{\text{F1}} \right) \hat{\mathbf{W}}_{\text{F2}} \right\|_F$$

$$+ G_\pi \left\| \sigma \left( \mathbf{Y}_{[N]} \mathbf{W}_{\text{F1}} \right) \hat{\mathbf{W}}_{\text{F2}} - \sigma \left( \hat{\mathbf{Y}}_{[N]} \hat{\mathbf{W}}_{\text{F1}} \right) \hat{\mathbf{W}}_{\text{F2}} \right\|_F + G_\pi \left\| \mathbf{Y}_{[N]} - \hat{\mathbf{Y}}_{[N]} \right\|_F$$

$$\leq G_\pi \left( \epsilon_{F2} + B \left\| \mathbf{Y}_{[N]} \mathbf{W}_{\text{F1}} - \hat{\mathbf{Y}}_{[N]} \hat{\mathbf{W}}_{\text{F1}} \right\|_F + \left\| \mathbf{Y}_{[N]} - \hat{\mathbf{Y}}_{[N]} \right\|_F \right)$$

$$\leq G_\pi \left( \epsilon_{F2} + B \left\| \mathbf{Y}_{[N]} \mathbf{W}_{\text{F1}} - \mathbf{Y}_{[N]} \hat{\mathbf{W}}_{\text{F1}} \right\|_F + B \left\| \mathbf{Y}_{[N]} \hat{\mathbf{W}}_{\text{F1}} - \hat{\mathbf{Y}}_{[N]} \hat{\mathbf{W}}_{\text{F1}} \right\|_F + \left\| \mathbf{Y}_{[N]} - \hat{\mathbf{Y}}_{[N]} \right\|_F \right)$$

$$\leq G_\pi \left( \epsilon_{F2} + B \epsilon_{F1} + (B^2 + 1) \left\| \mathbf{Y}_{[N]} - \hat{\mathbf{Y}}_{[N]} \right\|_F \right).$$

For $\left\| \mathbf{Y}_{[N]} - \hat{\mathbf{Y}}_{[N]} \right\|_F$ we have:

$$\left\| \mathbf{Y}_{[N]} - \hat{\mathbf{Y}}_{[N]} \right\|_F = \left\| \Pi_{\text{norm}} \left( \sum_{h \in H} \mathbf{A}_h \mathbf{W}_{O_h} + \mathbf{Z}_{[N]} \right) - \Pi_{\text{norm}} \left( \sum_{h \in H} \hat{\mathbf{A}}_h \hat{\mathbf{W}}_{O_h} + \mathbf{Z}_{[N]} \right) \right\|_F$$

$$\leq G_\pi \sum_{h \in H} \left\| \mathbf{A}_h \mathbf{W}_{O_h} - \hat{\mathbf{A}}_h \hat{\mathbf{W}}_{O_h} \right\|_F$$

$$\leq G_\pi \sum_{h \in H} \left( \left\| \mathbf{A}_h \mathbf{W}_{O_h} - \mathbf{A}_h \hat{\mathbf{W}}_{O_h} \right\|_F + \left\| \mathbf{A}_h \hat{\mathbf{W}}_{O_h} - \hat{\mathbf{A}}_h \hat{\mathbf{W}}_{O_h} \right\|_F \right)$$

$$\leq G_\pi \left( H \epsilon_d + BH \epsilon_A \right).$$

In summary, we have:

$$\left\| \mathbf{Z}_{[N]} - \hat{\mathbf{Z}}_{[N]} \right\|_F \leq G_\pi \left( \epsilon_{F2} + B \epsilon_{F1} + G_\pi (B^2 + 1)(H \epsilon_d + BH \epsilon_A) \right)$$

$$= G_\pi \epsilon_{F2} + G_\pi B \epsilon_{F1} + G_\pi^2 (B^2 + 1) H \epsilon_d + G_\pi^2 (B^3 + B) H \epsilon_A.$$

We can conclude that $\mathcal{C}_T$ $\epsilon$ covers $\mathcal{H}_S^{TF}$ By setting

$$\epsilon_{F2} = \frac{\epsilon}{4 G_\pi}, \epsilon_{F1} = \frac{\epsilon}{4 G_\pi B}, \epsilon_d = \frac{\epsilon}{4 G_\pi^2 (B^2 + 1) H}, \epsilon_A = \frac{\epsilon}{4 G_\pi^2 (B^3 + B) H}.$$

From the definition of $\mathcal{C}_T$, we have:

$$\ln|\mathcal{C}_T| \leq \ln|\mathcal{C}_{F_2}|$$

$$\leq 4d^2 H \ln\left(1 + \frac{8G_s G_\pi^2 (B^4 + B^2) H\sqrt{N\ln m}\|\mathbf{Z}_{[N]}\|_F}{\epsilon}\right)$$

$$+ 4dd_k H \ln\left(1 + \frac{32e^C G_\pi^2 (B^6 + B^4) H\|\mathbf{Z}^*\|_F \|\mathbf{Z}_{[N]}\|_F^2}{\sqrt{d_k}\epsilon}\right)$$

$$+ 2dd_f \ln\left(1 + \frac{8G_\pi^2 B^2 \left(e^C B^2 H\sqrt{N\ln m} + 1\right)\|\mathbf{Z}_{[N]}\|_F}{\epsilon}\right)$$

$$\lesssim (4d^2 H + 4dd_k H + 2dd_f)\ln\left(1 + \frac{G_\pi^2 B^2 (B^2 + 1)\left(e^C B^2 H\sqrt{N\ln m} + 1\right)\|\mathbf{Z}_{[N]}\|_F}{\epsilon}\right).$$

Combining $d_k = d/H, d_f = 4d$, we have:

$$\ln\mathcal{N}\left(\mathcal{H}_S^{TF}, \epsilon, \|\cdot\|_F\right) \leq \ln|\mathcal{C}_T| \lesssim 4d^2(H+3)\ln\left(1 + \frac{G_\pi^2 B^2 (B^2 + 1)\left(e^C B^2 H\sqrt{N\ln m} + 1\right)\|\mathbf{Z}_{[N]}\|_F}{\epsilon}\right).$$

$\square$

**Lemma C.12.** *Let* $\mathbf{Z}_{[N]}^l \in \mathbb{R}^{Nm\times d}$ *be the lth layer's concatenated output matrix of our transformer model defined in Equation* (13), *we have:*

$$\left\|\mathbf{Z}_{[N]}^l\right\|_F \leq \prod_{j\in[l]} G_\pi^2 (B_j^2 + 1)\left(e^{C_j} B_j^2 H\sqrt{N\ln m} + 1\right)\|\mathbf{Z}_{[N]}^0\|_F.$$

*Furthermore, let* $\mathbf{Z}^l \in \mathbb{R}^{m\times d}$ *as the output matrix, we have:*

$$\left\|\mathbf{Z}^l\right\|_F \leq \prod_{j\in[l]} G_\pi^2 (B_j^2 + 1)\left(e^{C_j} B_j^2 H\sqrt{\ln m} + 1\right)\|\mathbf{Z}^0\|_F.$$

*Proof.*

$$\left\|\mathbf{Z}_{[N]}^l\right\|_F = \left\|\Pi_{\text{norm}}\left(\sigma\left(\mathbf{Y}_{[N]}^l \mathbf{W}_{F1}^l\right)\mathbf{W}_{F2}^l + \mathbf{Y}_{[N]}^l\right)\right\|_F$$

$$\overset{(i)}{\leq} G_\pi \left\|\sigma\left(\mathbf{Y}_{[N]}^l \mathbf{W}_{F1}^l\right)\mathbf{W}_{F2}^l + \mathbf{Y}_{[N]}^l\right\|_F$$

$$\leq G_\pi(B_l^2 + 1)\left\|\Pi_{\text{norm}}\left(\sum_{h\in[H]}\mathbf{A}_h^l\left(\mathbf{Z}_{[N]}^{l-1}\right)\mathbf{W}_{O_h}^l + \mathbf{Z}_{[N]}^{l-1}\right)\right\|_F$$

$$\leq G_\pi^2(B_l^2 + 1)\left(B_l\sum_{h\in[H]}\left\|\mathbf{S}_h^l \mathbf{Z}_{[N]}^{l-1}\mathbf{W}_{V_h}^l\right\|_F + \|\mathbf{Z}_{[N]}^{l-1}\|_F\right)$$

$$\overset{(ii)}{\leq} G_\pi^2(B_l^2 + 1)\left(e^{C_l} B_l^2 H\sqrt{N\ln m} + 1\right)\|\mathbf{Z}_{[N]}^{l-1}\|_F$$

$$\leq \prod_{j\in[l]} G_\pi^2(B_j^2 + 1)\left(e^{C_j} B_j^2 H\sqrt{N\ln m} + 1\right)\|\mathbf{Z}_{[N]}^0\|_F.$$

Here $(i)$ uses Assumption C.1, and $(ii)$ uses Lemma C.8. Similarly, we can get the second result by setting $N = 1$. $\square$

**Lemma C.13.** *For a single transformer-decoder-block* $\mathrm{TF}_{\mathcal{W}}(\cdot)$ *parameterized by* $\mathcal{W}$ *(ignore the layer indices), let* $\mathbf{Z}_{[N]} \in \mathbb{R}^{Nm \times d}$ *as the concatenated input matrix, we have:*

$$\left\| \mathrm{TF}_{\mathcal{W}}\left(\mathbf{Z}_{[N]}\right) - \mathrm{TF}_{\mathcal{W}}\left(\hat{\mathbf{Z}}_{[N]}\right) \right\|_F \lesssim G_\pi^2(B^2+1)\left(e^C B^2 H \sqrt{N} md + 1\right)\left\|\mathbf{Z}_{[N]} - \hat{\mathbf{Z}}_{[N]}\right\|_F.$$

*Furthermore, let* $\mathbf{Z} \in \mathbb{R}^{m \times d}$ *as the input matrix, we have:*

$$\left\| \mathrm{TF}_{\mathcal{W}}\left(\mathbf{Z}\right) - \mathrm{TF}_{\mathcal{W}}\left(\hat{\mathbf{Z}}\right) \right\|_F \leq G_\pi^2(B^2+1)\left(e^C B^2 Hmd + 1\right)\left\|\mathbf{Z} - \hat{\mathbf{Z}}\right\|_F.$$

*Proof.*

$$\left\| \mathrm{TF}_{\mathcal{W}}\left(\mathbf{Z}_{[N]}\right) - \mathrm{TF}_{\mathcal{W}}\left(\hat{\mathbf{Z}}_{[N]}\right) \right\|_F \leq G_\pi \left\| \sigma\left(\mathbf{Y}_{[N]}\mathbf{W}_{\mathrm{F1}}\right)\mathbf{W}_{\mathrm{F2}} + \mathbf{Y}_{[N]} - \sigma\left(\hat{\mathbf{Y}}_{[N]}\mathbf{W}_{\mathrm{F1}}\right)\mathbf{W}_{\mathrm{F2}} - \hat{\mathbf{Y}}_{[N]} \right\|_F$$

$$\leq G_\pi^2(B^2+1)\left\| \sum_{h \in [H]} \mathbf{A}_h\left(\mathbf{Z}_{[N]}\right)\mathbf{W}_{O_h} + \mathbf{Z}_{[N]} - \sum_{h \in [H]} \mathbf{A}_h\left(\hat{\mathbf{Z}}_{[N]}\right)\mathbf{W}_{O_h} - \hat{\mathbf{Z}}_{[N]} \right\|_F$$

$$\leq G_\pi^2(B^2+1)\left( B \sum_{h \in [H]} \left\| \mathbf{A}_h\left(\mathbf{Z}_{[N]}\right) - \mathbf{A}_h\left(\hat{\mathbf{Z}}_{[N]}\right) \right\|_F + \left\| \mathbf{Z}_{[N]} - \hat{\mathbf{Z}}_{[N]} \right\|_F \right).$$

For any $h \in [H]$, we have:

$$\left\| \mathbf{A}_h\left(\mathbf{Z}_{[N]}\right) - \mathbf{A}_h\left(\hat{\mathbf{Z}}_{[N]}\right) \right\|_F = \left\| \mathbf{S}_h \mathbf{Z}_{[N]}\mathbf{W}_{V_h} - \hat{\mathbf{S}}_h \hat{\mathbf{Z}}_{[N]}\mathbf{W}_{V_h} \right\|_F$$

$$\leq B\left( \left\| \mathbf{S}_h \mathbf{Z}_{[N]} - \hat{\mathbf{S}}_h \mathbf{Z}_{[N]} \right\|_F + \left\| \hat{\mathbf{S}}_h \mathbf{Z}_{[N]} - \hat{\mathbf{S}}_h \hat{\mathbf{Z}}_{[N]} \right\|_F \right)$$

$$\leq B\left( \sqrt{Nmd}\left\| \mathbf{S}_h - \hat{\mathbf{S}}_h \right\|_F + e^C \sqrt{N \ln m}\left\| \mathbf{Z}_{[N]} - \hat{\mathbf{Z}}_{[N]} \right\|_F \right).$$

For $\left\| \mathbf{S}_h - \hat{\mathbf{S}}_h \right\|_F$, we have:

$$\left\| \mathbf{S}_h - \hat{\mathbf{S}}_h \right\|_F = \sqrt{ \sum_{i \in [N]} \left\| \mathbf{S}_{hi} - \hat{\mathbf{S}}_{hi} \right\|_F^2 }$$

$$= \sqrt{ \sum_{i \in [N]} \left\| \mathrm{softmax}\left( \frac{\mathbf{Z}_i \mathbf{W}_{Qh}\left(\mathbf{Z}_i \mathbf{W}_{Kh}\right)^\top + \mathbf{M}}{\sqrt{d_k}} \right) - \mathrm{softmax}\left( \frac{\hat{\mathbf{Z}}_i \mathbf{W}_{Qh}\left(\hat{\mathbf{Z}}_i \mathbf{W}_{Kh}\right)^\top + \mathbf{M}}{\sqrt{d_k}} \right) \right\|_F^2 }$$

$$\leq \frac{G_s}{\sqrt{d_k}} \sqrt{ \sum_{i \in [N]} \left\| \mathbf{Z}_i \mathbf{W}_{Qh}\left(\mathbf{Z}_i \mathbf{W}_{Kh}\right)^\top - \hat{\mathbf{Z}}_i \mathbf{W}_{Qh}\left(\hat{\mathbf{Z}}_i \mathbf{W}_{Kh}\right)^\top \right\|_F^2 }$$

$$\leq \frac{G_s}{\sqrt{d_k}} \sqrt{ \sum_{i \in [N]} \left( \left\| \left(\mathbf{Z}_i \mathbf{W}_{Qh} - \hat{\mathbf{Z}}_i \mathbf{W}_{Qh}\right)\left(\mathbf{Z}_i \mathbf{W}_{Kh}\right)^\top \right\|_F + \left\| \hat{\mathbf{Z}}_i \mathbf{W}_{Qh}\left[ \left(\mathbf{Z}_i \mathbf{W}_{Kh}\right)^\top - \left(\hat{\mathbf{Z}}_i \mathbf{W}_{Kh}\right)^\top \right] \right\|_F \right)^2 }$$

$$\leq \frac{G_s B^2}{\sqrt{d_k}} \sqrt{ \sum_{i \in [N]} \left( \|\mathbf{Z}_i\|_F + \|\hat{\mathbf{Z}}_i\|_F \right)^2 \left\| \mathbf{Z}_i - \hat{\mathbf{Z}}_i \right\|_F^2 }$$

$$\leq \frac{2G_s B^2 \sqrt{md}}{\sqrt{d_k}} \sqrt{ \sum_{i \in [N]} \left\| \mathbf{Z}_i - \hat{\mathbf{Z}}_i \right\|_F^2 }$$

$$= \frac{2G_s B^2 \sqrt{md}}{\sqrt{d_k}} \left\| \mathbf{Z}_{[N]} - \hat{\mathbf{Z}}_{[N]} \right\|_F.$$

Combining the above results, we can get:

$$
\begin{aligned}
\left\| \mathrm{TF}_{\mathcal{W}}\left(\mathbf{Z}_{[N]}\right) - \mathrm{TF}_{\mathcal{W}}\left(\hat{\mathbf{Z}}_{[N]}\right) \right\|_F &\leq G_\pi^2(B^2+1)\left(e^C B^2 H\sqrt{N\ln m} + \frac{2G_s B^4 H\sqrt{N}md}{\sqrt{d_k}} + 1\right)\left\|\mathbf{Z}_{[N]} - \hat{\mathbf{Z}}_{[N]}\right\|_F \\
&\lesssim G_\pi^2(B^2+1)\left(e^C B^2 H\sqrt{N}md + 1\right)\left\|\mathbf{Z}_{[N]} - \hat{\mathbf{Z}}_{[N]}\right\|_F.
\end{aligned}
$$

$\square$

### C.4. Proof of Theorem 4.20

*Proof.* **Step 1:** We firstly define the class of $l$th layer's output as:

$$
\mathcal{H}^l = \left\{
\begin{aligned}
&\mathbf{X} \mapsto TF_{\mathbf{W}^l}\left(TF_{\mathbf{W}^{l-1}}\ldots TF_{\mathbf{W}^1}\left(\mathrm{Embedding}\left(\mathbf{X}\right)\right)\right): \\
&\left\|\mathbf{W}_{\mathrm{F1}}^j\right\|_F, \left\|\mathbf{W}_{\mathrm{F2}}^j\right\|_F, \left\|\mathbf{W}_{O_h}^j\right\|_F, \left\|\mathbf{W}_{Q_h}^j\right\|_F, \left\|\mathbf{W}_{K_h}^j\right\|_F, \left\|\mathbf{W}_{V_h}^j\right\|_F \leq B_j, \forall j \in [l], \forall h \in [H]
\end{aligned}
\right\}.
$$

For $l = 1$, we consider the set:

$$
\mathrm{TF}_1\left(\mathbf{Z}_{[N]}^0\right) := \left\{\mathrm{TF}_{\mathcal{W}^1}\left(\mathbf{Z}_{[N]}^0\right) : \mathbf{W}^1 \in \mathcal{W}^1\right\},
$$

where $\mathbf{Z}_{[N]}^0 = \mathrm{Embedding}\left(\mathbf{X}_{[N]}\right)$ represents the embedded token sequences, and $\mathcal{W}^l$ defined in (14). We denote $\mathcal{C}_1 = \mathcal{C}\left(\mathrm{TF}_1\left(\mathbf{Z}_{[N]}^0\right), \epsilon_1, \|\cdot\|_F\right)$ as the $\epsilon_1$-cover of $\mathrm{TF}_1\left(\mathbf{Z}_{[N]}^0\right)$. Then for $1 < l+1 \leq L$, let $\mathcal{C}_l$ be a cover of $\mathcal{H}^l$, we select one element $\hat{\mathbf{Z}}_{[N]}^l \in \mathcal{C}_l$ to construct the $\epsilon_{l+1}$-cover of following set:

$$
\mathrm{TF}_{l+1}\left(\hat{\mathbf{Z}}_{[N]}^l\right) := \left\{\mathrm{TF}_{\mathcal{W}^{l+1}}\left(\hat{\mathbf{Z}}_{[N]}^l\right) : \mathbf{W}^{l+1} \in \mathcal{W}^{l+1}\right\},
$$

here we denote $\mathcal{C}\left(\mathrm{TF}_{\mathcal{W}^{l+1}}\left(\hat{\mathbf{Z}}_{[N]}^l\right), \epsilon_{l+1}, \|\cdot\|_F\right)$ as the cover. Then by Lemma C.12 and Proposition C.11, we have:

$$
\begin{aligned}
\ln\left|\mathcal{C}\left(\mathrm{TF}_{\mathcal{W}^{l+1}}\left(\hat{\mathbf{Z}}_{[N]}^l\right), \epsilon_{l+1}, \|\cdot\|_F\right)\right| &\leq 4d^2(H+3)\ln\left(1 + \frac{G_\pi^2 B_{l+1}^2(B_{l+1}^2+1)\left(e^{C_{l+1}}B_{l+1}^2 H\sqrt{N}md + 1\right)\|\hat{\mathbf{Z}}_{[N]}^l\|_F}{\epsilon_{l+1}}\right) \\
&\leq 4d^2(H+3)\ln\left(1 + \frac{B_{l+1}^2 s_{l+1}\|\mathbf{Z}_{[N]}^0\|_F}{\epsilon_{l+1}}\right) \\
&:= \ln\mathcal{N}_{l+1},
\end{aligned}
$$

where $s_{l+1} := \prod_{j\in[l+1]} G_\pi^2(B_j^2+1)\left(e^{C_j}B_j^2 H\sqrt{N}md + 1\right)$.

**Step 2:** Now, we can cunstruct the cover of $\mathcal{H}^{l+1}$ as:

$$
\mathcal{C}_{l+1} = \bigcup_{\hat{\mathbf{Z}}_{[N]}^l \in \mathcal{C}_l} \mathcal{C}\left(\mathrm{TF}_{\mathcal{W}^{l+1}}\left(\hat{\mathbf{Z}}_{[N]}^l\right), \epsilon_{l+1}, \|\cdot\|_F\right).
$$

Then we can get:

$$
|\mathcal{C}_{l+1}| = \left|\bigcup_{\hat{\mathbf{Z}}_{[N]}^l \in \mathcal{C}_l} \mathcal{C}\left(\mathrm{TF}_{\mathcal{W}^{l+1}}\left(\hat{\mathbf{Z}}_{[N]}^l\right), \epsilon_{l+1}, \|\cdot\|_F\right)\right| \leq |\mathcal{C}_l|\mathcal{N}_{l+1} \leq \prod_{j=1}^{l+1}\mathcal{N}_j.
$$

We have:

$$
\begin{aligned}
\ln|\mathcal{C}_{l+1}| &\leq \sum_{j=1}^{l+1}\ln\mathcal{N}_j \\
&\leq 4d^2(H+3)\sum_{j=1}^{l+1}\ln\left(1 + \frac{B_j^2 s_j\|\mathbf{Z}_{[N]}^0\|_F}{\epsilon_j}\right)
\end{aligned}
$$

**Step 3:** Next we go to verify that $\mathcal{C}_L$ is a cover of $\mathcal{H}^L$. By Lemma C.13, for any $\mathbf{Z}^L_{[N]} \in \mathcal{H}^L$ we can find a $\hat{\mathbf{Z}}^L_{[N]} \in \mathcal{C}_L$ such that:

$$
\begin{aligned}
\left\| \mathbf{Z}^L_{[N]} - \hat{\mathbf{Z}}^L_{[N]} \right\|_F &= \left\| \mathrm{TF}_{\mathcal{W}^L}\left(\mathbf{Z}^L_{[N]}\right) - \mathrm{TF}_{\hat{\mathcal{W}}^L}\left(\hat{\mathbf{Z}}^L_{[N]}\right) \right\|_F \\
&\le \left\| \mathrm{TF}_{\mathcal{W}^L}\left(\mathbf{Z}^L_{[N]}\right) - \mathrm{TF}_{\mathcal{W}^L}\left(\hat{\mathbf{Z}}^L_{[N]}\right) \right\|_F + \left\| \mathrm{TF}_{\mathcal{W}^L}\left(\hat{\mathbf{Z}}^L_{[N]}\right) - \mathrm{TF}_{\hat{\mathcal{W}}^L}\left(\hat{\mathbf{Z}}^L_{[N]}\right) \right\|_F \\
&\le G_\pi^2(B_L^2 + 1)\left(e^{C_l} B_L^2 H\sqrt{N}md + 1\right) \left\| \mathbf{Z}^{L-1}_{[N]} - \hat{\mathbf{Z}}^{L-1}_{[N]} \right\|_F + \epsilon_L \\
&\le \sum_{l=1}^{L} \prod_{j'=l+1}^{L} G_\pi^2(B_{j'}^2 + 1)\left(e^{C_{j'}} B_{j'}^2 H\sqrt{N}md + 1\right) \epsilon_l.
\end{aligned}
$$

We set $\epsilon_L = \frac{\epsilon}{L}$, and for $1 \le l \le L-1$, we choose $\epsilon_l = \left(L\prod_{j'=l+1}^{L} G_\pi^2(B_{j'}^2 + 1)\left(e^{C_{j'}} B_{j'}^2 H\sqrt{N}md + 1\right)\right)^{-1} \epsilon$. We denote $s_{l+1\to L} := \prod_{j'=l+1}^{L} G_\pi^2(B_{j'}^2 + 1)\left(e^{C_{j'}} B_{j'}^2 H\sqrt{N}md + 1\right)$, then we can get the covering number of $\mathcal{H}^L$ as following:

$$
\begin{aligned}
\ln \mathcal{N}\left(\mathcal{H}^L, \epsilon, \|\cdot\|_F\right) &\le \ln |\mathcal{C}_L| \\
&\le 4d^2(H+3) \sum_{l=1}^{L} \ln\left(1 + \frac{B_l^2 s_l \|\mathbf{Z}^0_{[N]}\|_F}{\epsilon_l}\right) \\
&\le 4d^2(H+3) \sum_{l=1}^{L} \ln\left(1 + \frac{B_l^2 s_l (s_{l+1\to L}) L \|\mathbf{Z}^0_{[N]}\|_F}{\epsilon}\right) \\
&= 4d^2(H+3) \sum_{l=1}^{L} \ln\left(1 + \frac{B_l^2 L s_L \|\mathbf{Z}^0_{[N]}\|_F}{\epsilon}\right).
\end{aligned}
$$

Furthermore, let $\mathbf{X} \in \mathbb{R}^{m \times n_v}$ as the input matrix, and $\mathbf{Z}^0 = \mathrm{Embedding}\,(\mathbf{X})$, by scaling $N$ to $1$, we have:

$$
\ln \mathcal{N}\left(\mathcal{H}_{|\mathbf{X}}, \epsilon, \|\cdot\|_F\right) \le 4d^2(H+3) \sum_{l=1}^{L} \ln\left(1 + \frac{B_l^2 L s'_L \|\mathbf{Z}^0\|_F}{\epsilon}\right)
$$

where $s'_L := \prod_{l\in[L]} G_\pi^2(B_l^2 + 1)\left(e^{C_l} B_l^2 Hmd + 1\right)$. $\qquad\square$

## D. Proof of Theorem 4.22

*Proof.* **Step 1:** We first bound the Rademacher complexity $\tilde{\mathfrak{R}}_D(\mathcal{H})$. For $\mathbf{Z}_i = \left[\mathbf{t}^i_0, \cdots, \mathbf{t}^i_{m-1}\right] \in \mathbb{R}^{m \times n_v}$, note that we do not input each token one by one in order, but input the entire sequence at once, and get a representation matrix $h(\mathbf{Z}_i) \in \mathbb{R}^{m \times d}$. Due to the existence of mask $\mathbf{M}$, each token can only use the information before the current node. We can naturally abstract the above process into using the $j$th token $\mathbf{t}^i_j$ to perform $m$ queries in sequence. Therefore $\tilde{\mathfrak{R}}_D(\mathcal{H})$ can be defined as:

$$
\tilde{\mathfrak{R}}_D(\mathcal{H}) := \mathbb{E}_{\boldsymbol{\epsilon}} \left[ \sup_{h\in\mathcal{H}} \frac{1}{Nm} \sum_{i\in[N]} \sum_{j\in[m]} \varepsilon_{ij} \left\| h\left(\mathbf{t}^i_{j-1}\right) \right\|_F \right].
$$

By Lemma C.7 we know: for any $i \in [N], j \in [m]$, we have $\left\| h\left(\mathbf{t}^i_{j-1}\right) \right\|_F \le \sqrt{d}$. Therefore, according to Lemma C.5, we have:

$$
\tilde{\mathfrak{R}}_D(\mathcal{H}) \le \inf_{\alpha>0} \left( \frac{4\alpha}{\sqrt{Nm}} + \frac{12}{Nm} \int_\alpha^{\sqrt{Nmd}} \sqrt{\ln \mathcal{N}\left(\mathcal{H}_{|D}, \epsilon, \|\cdot\|_2\right)} d\epsilon \right).
$$

We denote $\lambda = 4d^2(H+3)$ and $\rho_L = \sum_{l=1}^L B_l^2$, then by Theorem 4.20, we can get:

$$\int_\alpha^{\sqrt{Nmd}} \sqrt{\ln \mathcal{N}\left(\mathcal{H}_{|D}, \epsilon, \|\cdot\|_2\right)} d\epsilon \leq \sqrt{\lambda L} \int_\alpha^{\sqrt{Nmd}} \left(\sum_{l=1}^L \frac{1}{L} \ln \left(1 + \frac{B_l^2 L s_L \|\mathbf{Z}_{[N]}^0\|_F}{\epsilon}\right)\right)^{\frac{1}{2}} d\epsilon$$

$$\overset{(a)}{\leq} \sqrt{\lambda L} \int_\alpha^{\sqrt{Nmd}} \left(\ln \left(1 + \frac{\rho_L s_L \|\mathbf{Z}_{[N]}^0\|_F}{\epsilon}\right)\right)^{\frac{1}{2}} d\epsilon$$

$$\overset{(b)}{\leq} \sqrt{\lambda L}(\sqrt{Nmd} - \alpha) \left(\ln \left(\frac{1}{\sqrt{Nmd} - \alpha} \int_\alpha^{\sqrt{Nmd}} \left(1 + \frac{\rho_L s_L \|\mathbf{Z}_{[N]}^0\|_F}{\epsilon}\right) d\epsilon\right)\right)^{\frac{1}{2}}$$

$$= \sqrt{\lambda L}(\sqrt{Nmd} - \alpha) \left(\ln \left(1 + \frac{\rho_L s_L \|\mathbf{Z}_{[N]}^0\|_F \ln \frac{\sqrt{Nmd}}{\alpha}}{\sqrt{Nmd} - \alpha}\right)\right)^{\frac{1}{2}}$$

$$\leq \sqrt{\lambda L Nmd} \left(\ln \left(1 + \frac{\rho_L s_L \|\mathbf{Z}_{[N]}^0\|_F \ln \frac{\sqrt{Nmd}}{\alpha}}{\sqrt{Nmd} - \alpha}\right)\right)^{\frac{1}{2}}.$$

Here $(a)$ uses Jensen's inequality, $(b)$ uses Lemma C.6. By setting $\alpha = \frac{1}{\sqrt{Nmd}}$, we have:

$$\tilde{\mathfrak{R}}_D(\mathcal{H}) \leq \inf_{\alpha > 0} \left(\frac{4\alpha}{\sqrt{Nm}} + \frac{12}{Nm} \left(\sqrt{\lambda L Nmd} \left(\ln \left(1 + \frac{\rho_L s_L \|\mathbf{Z}_{[N]}^0\|_F \ln \frac{\sqrt{Nmd}}{\alpha}}{\sqrt{Nmd} - \alpha}\right)\right)^{\frac{1}{2}}\right)\right)$$

$$\leq \frac{4}{Nm\sqrt{d}} + \frac{12\sqrt{\lambda L d}}{\sqrt{Nm}} \left(\ln \left(1 + \frac{\rho_L s_L \|\mathbf{Z}_{[N]}^0\|_F \ln(Nmd)}{\sqrt{Nmd} - 1/\sqrt{Nmd}}\right)\right)^{\frac{1}{2}}$$

$$\overset{(i)}{\lesssim} \frac{\sqrt{\lambda L d}}{\sqrt{Nm}} \sqrt{\ln \left(1 + \rho_L s_L\right)}$$

$$\leq \sqrt{\frac{12 L d^3 (H+3) \ln \left(1 + \rho_L s_L\right)}{Nm}},$$

where $(i)$ uses the fact that $\|\mathbf{Z}_{[N]}^0\|_F$ is of $\sqrt{Nmd}$.

**Step 2:** Next, we consider the Rademacher complexity $\tilde{\mathfrak{R}}_D(\mathcal{G} \circ \hat{h})$:

$$\tilde{\mathfrak{R}}_D(\mathcal{G} \circ \hat{h}) = \mathbb{E}_{\boldsymbol{\varepsilon}} \left[\sup_{\|\mathbf{W}^P\|_F \leq R} \frac{1}{Nm} \sum_{i \in [N]} \sum_{j \in [m]} \varepsilon_{ij} \left\|\text{softmax}\left(\hat{h}\left(\mathbf{t}_{j-1}^i\right) \mathbf{W}^P\right)\right\|_F\right].$$

We consider the set

$$\mathcal{G} \circ \hat{h} := \left\{\text{softmax}\left(\hat{\mathbf{Z}}_{[N]}^L \mathbf{W}^P\right) : \|\mathbf{W}^P\|_F \leq R\right\},$$

where $\hat{\mathbf{Z}}_{[N]}^L \in \mathcal{C}_L$. By Lemma C.4, C.7 and C.9 ,it is easy to have:

$$\ln \mathcal{N}\left(\mathcal{G} \circ \hat{h}, \epsilon, \|\cdot\|_F\right) \leq dn_v \ln \left(1 + \frac{2RG_s \|\hat{\mathbf{Z}}_{[N]}^L\|_F}{\epsilon}\right) \leq dn_v \ln \left(1 + \frac{2RG_s \sqrt{Nmd}}{\epsilon}\right).$$

Then, by Lemma C.5, we can get:

$$\tilde{\mathfrak{R}}_D(\mathcal{G} \circ \hat{h}) \leq \sqrt{\frac{dn_v \ln(1 + 2RG_s)}{Nm}}.$$

**Step 3:** For any $k \in [m]$, let $\mathbf{T}_k \in \mathbb{R}^{k \times n_v}$ be the input token sequence. Then for any $h, \hat{h} \in \mathcal{H}$ and $g \in \mathcal{G}$, we have:

$$
\left\| g\left(h\left(\mathbf{T}_k\right)\right) - g\left(\hat{h}\left(\mathbf{T}_k\right)\right) \right\|_F = \left\| h\left(\mathbf{T}_k\right)\mathbf{W}^P - \hat{h}\left(\mathbf{T}_k\right)\mathbf{W}^P \right\|_F
$$
$$
\leq R \left\| h\left(\mathbf{T}_k\right) - \hat{h}\left(\mathbf{T}_k\right) \right\|_F .
$$

Therefore $G_g = R$, then by Corollary 4.11, we have:

$$
\begin{aligned}
\mathcal{E}_{\mathcal{D}}(\hat{g}, \hat{h}) \leq & \underbrace{6G_\ell G_g \tilde{\mathfrak{R}}_D(\mathcal{H})}_{\textbf{(I)}} + \underbrace{6G_\ell \tilde{\mathfrak{R}}_D(\mathcal{G} \circ \hat{h})}_{\textbf{(II)}} + B_\ell \sqrt{\frac{8\ln\frac{4}{\delta}}{N}} + B_\ell\sqrt{\frac{C_{\varphi,r}\log\frac{2}{\delta}}{2m}} + 4B_\ell \operatorname{disc}(U) \\
\leq & 6G_\ell R \sqrt{\frac{12Ld^3(H+3)\ln\left(1+\rho_L s_L\right)}{Nm}} + 6G_\ell \sqrt{\frac{dn_v \ln(1+2RG_s)}{Nm}} \\
& + B_\ell \sqrt{\frac{8\ln\frac{4}{\delta}}{N}} + B_\ell \sqrt{\frac{C_{\varphi,r}\log\frac{2}{\delta}}{2m}} + 4B_\ell \operatorname{disc}(U) \\
\lesssim & G_\ell R \sqrt{\Theta dH} \sqrt{\frac{\ln\left(1+\rho_L s_L\right)}{Nm}} + G_\ell \sqrt{\frac{dn_v}{Nm}} + B_\ell \sqrt{\frac{8\ln\frac{4}{\delta}}{N}} + B_\ell \sqrt{\frac{C_{\varphi,r}\log\frac{2}{\delta}}{2m}} + 4B_\ell \operatorname{disc}(U).
\end{aligned}
$$

Note that, the number of model parameters is $L(10d^2 + 2Hd^2)$. Since $d$ is usually much larger than $H$, we follow the approach of Kaplan et al. (2020) which use $\Theta \approx 12Ld^2$ approximation to represent the number of model parameters. $\quad\square$

