# OpenReview forum: "On the Generalization Ability of Next-Token-Prediction Pretraining"
_ICML.cc/2025/Conference — ICML 2025 poster_

### Official Review · Reviewer_astS · 2025-03-14

**Overall Recommendation:** 2

**Summary:**

The authors give a generalization error bound for decoder-only transformer language models trained with next-token prediction objective. The bound is a function of the number of training sequences $N$, number tokens $m$ per such sequence, number of model parameters, etc. Within a sequence, they assume that tokens are dependent but follow certain mixing conditions (Definition 4.1). They further justify the error bounds with simulations on a small dataset.

**Recommendation**

The bound has a $O(1/\sqrt{m})$ term. If the sequence length is fixed, then even as the number of sequences $N \rightarrow \infty$, the bound does not tend to 0. This appears to make the bound weaker than bounds in the literature which rely only on the number of sequences (scaling like $O(1/\sqrt{N})$. Therefore, I recommend a reject unless the authors have some strong arguments why their bound is the tightest so far.

**Questions / Comments**

1. It seems the generalization error bound is at least as large as $O(1/\sqrt{m})$, even as the number of training sequences $N\rightarrow \infty$ no matter how large the model is. This seems to make the bound weak in practical scenarios. Specifically, the bound never goes to 0 if the sequence length is fixed. Can the mixing assumption explain this irreducible bound? Is there a worst-case mixing scenario where the generalization error is $O(1/\sqrt{m})$ however large $N$ is?


2. The generalization error bound (in Theorem 4.22) increases as the model size increases. But in practice, larger models have smaller generalization error. How do we understand this apparent contradiction? I see Figure 2 (rightmost) plot supporting the error bound. However, I am not convinced that this is how generalization error scales in general. Is there a certain regime, such as low-$Nm$ where this happens?

3. How do we interpret the discrepancy term $\mathrm{disc}(U)$? One could set $\bar \phi = \frac{1}{N} \sum_{j=1}^N \phi_j$ and think that we draw sequences from the same $\bar \phi$ distribution. Then $\mathrm{disc}(U)$ would be $0$. What other term in the bound increases with this reparametrized $\phi$? The mixing term $\Delta_m$?

4. Minor notation issue: Is the first argument of $\ell$ token probability vector, as suggested by equation (7)? Or is it a token, as suggested by the text right after eq (1)? In Assumption 3.4, the Lipschitz smoothness assumption on $\ell$ needs further clarification based on this answer.

5. Can the authors justify the mixing assumption in Definition 4.1 with some simulations?

6. In the experiments section, the fact that the bounds are in the same order of magnitude as the generalization error is surprising. To compute the generalization bound from Theorem 4.22, what values were used for the various quantities in the bound? Can the authors explain why they train for a relatively high 2000 epochs?

**Claims And Evidence:**

Please see above.

**Essential References Not Discussed:**

It would good to add this reference which also gives error bounds for language models:

Sanae Lotfi, Yilun Kuang, Marc Anton Finzi, Brandon Amos, Micah Goldblum, Andrew Gordon Wilson,

Unlocking Tokens as Data Points for Generalization Bounds on Larger Language Models, 2024, NeurIPS 2024

**Experimental Designs Or Analyses:**

Please see above.

**Methods And Evaluation Criteria:**

Please see above.

**Other Comments Or Suggestions:**

Please see above.

**Other Strengths And Weaknesses:**

Please see above.

**Questions For Authors:**

Please see above.

**Relation To Broader Scientific Literature:**

See above.

**Theoretical Claims:**

Please see above.

---

> ### Author Rebuttal · Authors · 2025-03-31
>
> Thank you for your insightful comments and questions. We have carefully considered them and have added supplementary explanations in the relevant sections of the paper. Additionally, we have incorporated the suggested references into our paper. Below are our responses:
>
> **A1:** This is indeed a significant question, and we fully understand your concerns. We must acknowledge that the sequence length $m$ is constrained by available training resources. However, from a theoretical standpoint, we assume that the sequence length $m$ can approach infinity. Firstly, from the intrinsic nature of human language, language sequences can indeed extend indefinitely, much like time series.
> Furthermore, based on the two papers [1, 2] we are aware of that also consider the generalization bounds of language models at the token level, their bounds include $\mathcal{O}(\sqrt{1/m})$, although they do not specifically target the NTP pre-training task. For instance, in Theorem 4.3 of paper [1], there is a term $\mathcal{O}(\sqrt{\frac{1}{T_p}})$, where $T_p$ represents sequence length, similar to $m$. Paper [2] also considers dependencies between tokens, and thus in their Theorem 1, the generalization bound is $\mathcal{O}(\sqrt{1/m})$. We hope this explanation alleviates your concerns. Additionally, we will enhance the discussion on sequence length in our paper and look forward to reaching a consensus with you.
>
> **A2:** It is true that larger models often perform better in practice. However, according to the scaling laws for large models [3], larger models also require more tokens for pre-training.  In the third experiment depicted in Figure 2 of our paper, we observed that when the total number of tokens is fixed, larger models tend to generalize worse. This is primarily because the limited number of tokens leads to overfitting in larger models. This observation aligns with our theoretical results.
>
> **A3:** For our detailed explanation regarding disc(U), please refer to ​Response **A3** in our reply to **Reviewer axCZ**. Furthermore, $C_{\varphi,r}$ represents the upper bound of $\Delta_{m}$ ( Remark 4.10) and also varies with $\phi$, measuring the diversity of the token sentences. We aim for a smaller $C_{\varphi,r}$ to enable the model to learn more diverse knowledge. We will further explore `disc(U)` and $\Delta_{m}$ in the discussion section of the paper.
>
> **A4:** In Section 3.1 of the paper, we clarify that all tokens $\mathbf{t}^i_j \in \mathbb{R}^{n_v}$ are vectors, meaning that the first parameter of the loss function $\ell$ is the probability vector computed by Equation (7).
>
> **A5:** We apologize for not being able to experimentally demonstrate in a short time frame that human language sequence is a mixing process. However, we provide two theoretical supports: (1) In numerous existing NLP studies, language sequences are often modeled as Markov processes [5,6]. Under certain conditions, Markov chains are indeed mixing processes [7]. (2) Under certain conditions, autoregressive processes are equivalent to mixing processes [8].
>
> **A6:** Regarding the calculation of the error bound, we followed the approach outlined in [9], our computed generalization bound is $\sqrt{\frac{\Theta \tau_{1}}{Nm}}$. (1) We set the $G_{\pi}$ to 1 based on Lemma A.9 in [10]; (2) $B_{l}$ and $C_{l}$ were calculated by extracting the model parameter matrix and the attention matrix during training.
> We set the epoch to 2000 to ensure the model converges as much as possible [11], as smaller datasets require more epochs for generalization [12]. The primary purpose of this experiment was to demonstrate that even with a limited number of training tokens, our generalization bound remains valid and does not collapse as discussed in [9]. Additionally, we are actively working to supplement our study with more experiments.
>
> **Reference：**
>
> [1]Gong.(2025) Towards Auto-Regressive Next-Token Prediction: In-context Learning Emerges from Generalization.
>
> [2]Lotfi.(2024) Unlocking Tokens as Data Points for Generalization Bounds on Larger Language Models.
>
> [3]Open AI.(2020) Scaling Laws for Neural Language Models.
>
> [4]Penedo, M.(2023) The Refined Web Dataset for Falcon LLM: Outperforming Curated Corpora with Web Data, and Web Data Only.
>
> [5]Zhang, Y.(2023) What and how does in-context learning learn? Bayesian model averaging, parameterization, and generalization.
>
> [6]Li, Y.(2023)Transformers as algorithms: Generalization and stability in in-context learning.
>
> [7]Meyn.(2012) Markov chains and stochastic stability.
>
> [8]Krishna.(1986) Mixing properties of harris chains and autoregressive processes.
>
> [9]Galanti.(2023) Norm-based Generalization Bounds for Sparse Neural Networks.
>
> [10]Edelman.(2022) Inductive Biases and Variable Creation in Self-Attention Mechanisms.
>
> [11]Madden.(2024) Upper and lower memory capacity bounds of transformers for next-token prediction.
>
> [12] Power.(2022) Grokking: Generalization beyond overfitting on small algorithmic datasets.

---

> > ### Comment · Reviewer_astS · 2025-04-08
> >
> > > Paper [2] also considers dependencies between tokens, and thus in their Theorem 1, the generalization bound is $O(\sqrt{1/m})$...
> >
> > Here $m$ is the number of tokens in the whole training set, not the length of one sequence, right? Are you saying that if you consider token dependencies, their bound reduces to $\mathcal{O}(\sqrt{1/\text{sequence length})$?
> >
> > > A3: For ...
> >
> > I did not get your answer. Can you please clarify what terms in Theorem 4.9 become large if you set all $\phi$'s equal like in my question?
> >
> > > A6:
> >
> > Other terms in the bound from Theorem 4.22 that are not mentioned in your comment, also contribute to the error, right? If some terms are excluded from the bound in calculation, the usefulness of the comparison is limited. It should be made clear how the bounds are computed at least in the supplement. Otherwise, I would say the figures are misleading.
> >
> > We also care more about the generalization error towards the "end" of the training.

---

> > > ### Author Response · Authors · 2025-04-08
> > >
> > > We are delighted to receive your feedback. Below are our detailed responses to the issues you raised:
> > >
> > > **A1:** Regarding the issues with Question 1 and paper [2]: First, we acknowledge that in paper [2], $m$ indeed refers to the total number of tokens in the entire training set. They concatenate all token sequences into a single sequence using the "EOT" token for the NTP task, effectively reducing the sequence count $N$ to 1. In our bound, when $N=1$, the order is also $\sqrt{\frac{1}{m}}$. Thus, in a sense, their bound can be seen as a special case of ours when $N=1$. The additional constant terms in our bound arise because we consider a more complex excess risk bound. If we, like [2], only considered the generalization error bound, these constants could also be omitted. From an engineering perspective, neither $N$ nor $m$ (or the total number of training tokens) can realistically reach infinity. Therefore, $N$ and $m$ can only approach infinity in theoretical analysis scenarios, making it reasonable to assume $m$ can approach infinity in our analysis. We hope the reviewer can understand this point. Finally, compared to [2], our bound better reflects the relationship between model parameter size and training token volume, aligning with the Scaling Laws [1], which suggest that training token volume should increase in tandem with model parameter size.
> > >
> > > **A2:** We apologize for any previous oversight on this issue. Here is a detailed explanation: When all $\phi$ are equal, disc(U) in Theorem 4.9 becomes 0. The term $||\Delta_{m}||\_{\infty}$ might increase, but this is not necessarily the case, as $||\Delta_{m}||\_{\infty}$ depends on the specific distribution of $\phi$ and is not significantly affected by whether all $\phi$ are equal. As previously explained, disc(U) primarily depends on the quality of data cleaning. The distribution differences between high-quality sample sequences are minimal and can often be ignored, as any two high-quality human statements can be connected by transitional phrases or conjunctions to form a coherent statement. In contrast, low-quality sentences, such as those with grammatical errors or misspellings, are difficult to merge with high-quality sentences (even if merged, they are hard for the model to comprehend). Therefore, distribution differences mainly stem from differences in sample quality. Based on this, we can further assume that there are only two distinct distributions: $\phi\_{\text{good}}$ for high-quality samples and $\phi\_{\text{bad}}$ for low-quality samples, similar to labeling each piece of data as "good" or "bad" during data cleaning. Thus, when data cleaning is thorough and no low-quality samples exist, only $\phi\_{\text{good}}$ remains, meaning all $\phi$ are equal. In this case, due to the overall high quality of the dataset, the model's generalization performance improves.
> > >
> > > **A3:** Thank you for your suggestion. We have explicitly stated the calculation formula for the bound in the experimental section of the paper. We acknowledge that discarding other terms may lose some insights, but since the focus of this paper is not on discussing the impact of data distribution or probabilistic factors on generalization, but rather on analyzing the influence of $\Theta$, $N$, and $m$, discarding these terms helps us focus on these parameters. This is the main reason for our choice, and we hope for your understanding. Additionally, the results shown in Figure 2 are from the end of training, not during the training process, and we have clarified this in the text.
> > >
> > > We sincerely thank the reviewer for taking the time and effort to provide feedback. We hope our responses clarify the issues mentioned. There is currently very little theoretical research related to LLM pre-training, and we are confident that this work, focusing on the theoretical analysis of NTP pre-training, is meaningful for the exploration and development of language models. We would be deeply grateful for your support. Best wishes!
> > >
> > > **Reference:**
> > >
> > > [1]Open AI.(2020) Scaling Laws for Neural Language Models.
> > >
> > > [2]Lotfi.(2024) Unlocking Tokens as Data Points for Generalization Bounds on Larger Language Models.

---

### Official Review · Reviewer_fzti · 2025-03-21

**Overall Recommendation:** 4

**Summary:**

This paper presents new Theoretical results to study the generalization of decoder-only transformer based LLM models. The paper revolves around using a Rademacher complexity argument to bound the generalization risk of a multi layer transformer model with fixed position encoding used for tokens. This work focuses on NTP pretraining of LLMs while taking into account the dependence between tokens. The paper then provides results for the covering number of their decoder model and finally bound the pretrained model's generalization using a Racdemacher complexity upper bound derived using the covering number. Large portion of the paper's work is regarding the derivation of the covering number, taking a step by step approach from each component of attention and ffn and building up to the full transformer decoder model.

The result provide a upper bound for the model with an experimental result to confirm the validity. As expected, increasing number of tokens and context size help close the test vs train generalization gap. Additionally the experimental results regarding fixing datapoints and increasing number of parameters suggest that the model tends to overfit on the limited data, suggesting that any scaling on parameters should be followed by a scaling in number of datapoints.

**Claims And Evidence:**

For the most part the claims in the paper are theoretical, followed by solid proofs and foundations. The main result that can be supported with experiments are the ones w.r.t Thm 4.22 on the generalization. In this regard, while the results are promising for the real data experiments, I would like to see further ablation studies or perhaps additional experiments with synthetic data. Given that the upper bounds are not tight, it would be beneficial to include additional toy experimental results. However, I don't believe this is necessary given the highly theoretical aspects of the paper and the limited assumptions made on model/data.

**Essential References Not Discussed:**

None to the best of my knowledge.

**Experimental Designs Or Analyses:**

As mentioned previously, the experimental results provided on the real data setup confirms the theoretical findings. However, I do believe providing more results with synthetic data could help improve my confidence.

regarding the experimental design, I do have a number of minor questions:

Q6: Why the inclusion of dorpout ?

Q7: What do the authors mean when they suggest grid search for $N$ and $m$ ? I just want to confirm that means changing the parameters not any optimization concern.

With regards to the results analysis,

Q8: For the conclusion from the first figure to the left, does having larger $m$ imply seeing more tokens ? Its a bit difficult for me to justify the conclusion drawn, given that based on the text, longer context length implies more total tokens seen given that the number of iterations for training are kept the same for all experiments (the same 2000 epochs)

Q9: Given some famous results such as double descent, I do wonder what the implications are where increasing model size hurts generalization performance. Does this mean the model is under parametrized w.r.t to the dataset ?

**Methods And Evaluation Criteria:**

More of a theoretical paper so evaluation criteria for comparison to other methods and models is not as critical.

**Other Comments Or Suggestions:**

I have given this work a 4 as I can't see any major issues w.r.t the claims made theoretically and given the novelty of the work (w.r.t analyzing NTP pretraining). I would like to see more experimental results and some further discussion regarding the observations.

**Other Strengths And Weaknesses:**

I would like sections in the appendix to further discuss the comparison of their to other bounds for LLMs, such as the ones suggested in table 1. I do understand that the previous work studies different training setups however I do believe such comparisons of bounds could help find potential aspects that impact different training setups.

**Questions For Authors:**

Q10: I would like to have a better understanding of what of what $k$ represents for the Def 4.1. To clarify, given a single sequence, $k$ determines how far off the two sample $A,B$ sequence are from one another. Is that correct ? and this is used to help ease the non-iid'ness of sequenced tokens.

Q11: Could the authors expand upon the $disc(U)$ and how it reflects on the quality of data. From my understanding, this term carries the bulk of non-iid'ness of tokens and their relation to each other within one sequence.

**Relation To Broader Scientific Literature:**

The authors do provide comparison to previous generalization bound for LLM models, having mentioned that their method is the first to consider the NTP pretraining regime.

**Theoretical Claims:**

I have gone through the proofs of the paper for the most part. From my understanding I haven't found significant issues, but I would like some clarifications on certain parts of the proofs. Namely:

Q1: In lemma C6, when the authors discuss "continuous concave (downward)", what condition on the function are they applying ? This is later applied to the function $ln(1+x/e)^{0.5}$ for the proof of Thm 4.22.

Q2: For the proof of C9, on line 1055, shouldn't it be an inequality ? I don't think it has major significance but just asking to make sure I didn't miss anything.

Q3: For the proof of C10, on line 1133 I missed how $||Z^*||_F$ into the bound. Could you please elaborate ?

Q4: For the proof of Thm 4.22, can the authors elaborate on the comment made on line 1582 regarding the masking of queries. I believe this is an important part of the proof and I don't exactly understand the argument. I don't understand how the current formulation incorporates the auto regressive nature ?

Q5: Proof of Thm 4.22, could the authors please elaborate on why the inequality is valid for line 1621 when they set$\alpha=\frac{1}{\sqrt{Nmd}}$ ? Specifically for the second term.


To be clear, whenever authors use results or lemmas from previous work, I took it for the most part at face value.

---

> ### Author Rebuttal · Authors · 2025-03-31
>
> First, thank you very much for your recognition and support of our work. We have incorporated your valuable suggestions by adding a comparative discussion of related past and recent work in the appendix and have made every effort to supplement our experiments. Additionally, we are more than willing to address any questions you have about this work:
>
> **A1:** The expression $\ln(1+x/e)^{0.5}$ can be seen as $g=f^{0.5}$ where $f=\ln(1+x/e)$. The function $g$ is "continuously concave (downward)."
>
> **A2:** This uses a property of the softmax function: $\sum\_{j\ne i} e\_j=1-e\_i$, so the expression inside the parentheses becomes $2(1-e\_i)$.
>
> **A3:** Please note the definitions of $\mathcal{C}\_{Q}$ and $\mathcal{C}\_{K}$ in line 1093, where the input data is any $\mathbf{Z}$. Here, $||\mathbf{Z}^*||\_F$ corresponds to $||\mathbf{X}||\_F$ in Lemma C.4, as we take the maximum value of the norm for the upper bound. Meanwhile, $||\mathbf{Z}\_{[N]}||\_F$ results from substituting the value of $\epsilon\_{Q}$.
>
> **A4:** This is an excellent question! Since we are discussing the pre-training phase, as described in our paper and the paper[6], this phase is not autoregressive. Autoregression is the mechanism used during model reasoning. In the pre-training phase, $m$ queries (tokens) are input simultaneously, and $m$ output tokens are produced simultaneously. Due to the masking mechanism, each query can only use preceding information, mimicking the autoregressive mechanism without seeing future information.
>
> **A5:** In line 1618, the right side of the inequality is a lower bound. Theoretically, $\alpha$ should be chosen to minimize this expression, i.e., where the derivative is zero. However, since deriving the expression with respect to $\alpha$ is complex, we set $\alpha=\frac{1}{\sqrt{Nmd}}$. Although this does not reach the lower bound, it simplifies derivation and analysis.
>
> **A6:** The training data used is too limited, so dropout is added to prevent overfitting, similar to the experimental setup in [1].
>
> **A7:** This does not involve any parameter optimization issues; it is solely to analyze the impact of parameter changes on model performance based on experimental results. We have clarified this in the paper.
>
> **A8:** Thank you for your question. To explore the optimal selection of parameters $N$ and $m$ under a fixed total number of tokens, as you suggested, we should vary the sequence length while keeping the total number of tokens constant. Our experiments aimed to verify theoretical results, but indeed, the total number of tokens increased. We are working on this exploratory experiment and aim to present the results in the final version.
>
> **A9:** Many empirical studies indicate that increasing model parameters excessively, while keeping the number of pre-training tokens constant, can harm model performance and generalization ability. Both Chinchilla's Law [2] and Scaling Laws [3] suggest that token numbers should increase alongside model parameters. This may be because larger models with fixed training tokens are more prone to over-parameterization and overfitting. As research in Nature paper [5] shows, for simple tasks with fewer tokens, large models may "memorize" noise or spurious patterns in the training data rather than learning underlying general rules, leading to poorer performance.
>
> **A10:** Your understanding is correct. For example, in the sentence "I went to the restaurant today, the food was delicious, and after eating, I went to the park, which was crowded," let event A = "I went to the restaurant today," B = "the food was delicious," and C = "the park was crowded." According to the definition of $\varphi$-mixing, the time step $k\_1$ between A and B is much smaller than the time step $k\_2$ between A and C, so we have $\varphi(k\_1)>\varphi(k\_2)$, indicating that the dependency between A and B is stronger than between A and C. The dependency decreases as the time step $k$ increases.
>
> **A11:** For the response regarding `disc(U)`, please refer to response **A3** in our reply to **Reviewer axCZ.**
>
> **Reference:**
>
> [1]Edelman, B.(2022) Inductive biases and variable creation in self-attention mechanisms.
>
> [2]DeepMind.(2022) Training Compute-Optimal Large Language Models.
>
> [3]Open AI.(2020) Scaling Laws for Neural Language Models.
>
> [4]Penedo, M.(2023) The Refined Web Dataset for Falcon LLM: Outperforming Curated Corpora with Web Data, and Web Data Only.
>
> [5]Zhou, L.(2024) Larger and more instructable language models become less reliable.
>
> [6]Bachmann, G.(2024) The pitfalls of next-token prediction.

---

> > ### Comment · Reviewer_fzti · 2025-04-05
> >
> > I would like to thank the authors for their diligent response. Most of my questions with respect to the theory and experimental results have been addressed in the rebuttal and I have looked through other reviews for the paper as well. I believe this work deserves the previous score I have given it. However, I have the following comments:
> >
> > A8 and A9: I believe the conclusions drawn from the rightmost results in Fig. 2 are still somewhat vague. As the authors mentioned, the distinction between "tokens seen" and "number of steps" remains unclear. Given their experimental setup, a model trained on more tokens for the same number of epochs has undergone more iterations. This could potentially explain some of the implications regarding overparameterization and overfitting discussed in relation to the scaling laws, as mentioned by the authors.
> >
> > Q11: I appreciate the authors’ explanation. However, I still find the concept of $disc(U)$ somewhat difficult to grasp. For example, based on the authors’ explanation, does the distribution $\tau_k$ represent examples in the language that include misspellings or grammatical issues?

---

> > > ### Author Response · Authors · 2025-04-06
> > >
> > > We greatly appreciate the reviewer's feedback and are very pleased to have your continued support. We are happy to hear that most of your previous questions have been resolved. Below are our responses to the two remaining issues:
> > >
> > > **A1:** Thank you for your insightful analysis! In the experiment shown on the far right of Figure 2, we fixed the total number of pre-training tokens and the total training steps, meaning that with constant data volume and computational resources, increasing the model's parameter size leads to poorer generalization performance. This experiment aims to demonstrate that as the model size increases, its capacity also grows, enhancing its ability to fit more data. Therefore, the pre-training token count (and computational resources) should be increased simultaneously to prevent overfitting.
> > >
> > > **A2:** Your understanding is entirely correct! When low-quality data, such as those with grammatical errors, exist in the pre-training dataset due to inadequate data cleaning, the distribution of this low-quality data differs significantly from that of other high-quality data. Thus, $\phi_k$ should represent the distribution of the low-quality data. In this paper, we assume each token sequence $\mathbf{X}_i$ follows a specific $\phi_i$ distribution, but we can further assume there are only two distinct distributions: $\phi\_{good}$ for high-quality data and $\phi\_{bad}$ for low-quality data. This is akin to labeling each piece of data as "good" or "bad" during data cleaning. Therefore, as long as the data cleaning quality is high enough to ensure no low-quality data exists, disc(U) can be considered negligible.
> > >
> > > We hope our responses address your concerns. Once again, we sincerely thank you for all your support, suggestions, and questions regarding this work. We hold your careful, patient, and responsible approach to reviewing in the highest regard and extend our best wishes to you.

---

### Official Review · Reviewer_axCZ · 2025-03-23

**Overall Recommendation:** 3

**Summary:**

This work derives new bounds on the generalization power of multi-layer multi-head transformer models pertained through Next-Token-Prediction mechanism. The first theorem in the paper shows that the generalization error is bounded by the Rademacher complexity of the class of $\mathcal{G}(\mathcal{H})$ (where $\mathcal{G}$ is the  token predictor (decoder) and $\mathcal{H}$ is representation learner) and other additive terms that decay with the number of training examples and the sequence length. The next main result bounds the above Rademacher complexity through covering number and shows that it is bounded by a terms that is increasing with the total number of parameters of the model and dimension, and decays with the number of training examples and the sequence length. The combination of these two gives a generalization bound that is mildly deteriorating with $'L'$ number of layers (only linear compared to previous works which was either exponential or quadratic) and decreases with the sequence length as $1/\sqrt{m}$. The method used in this work also leverages $\Phi$-mixing to account for the inter-token dependencies and allows for the masking operation.

**Claims And Evidence:**

Yes. Theoretical results are rigorous, proofs are provided in the appendix and real-world experiments align with the theoretical claims.

**Essential References Not Discussed:**

No.

**Experimental Designs Or Analyses:**

they sound valid. however, code is not provided by the authors.

**Methods And Evaluation Criteria:**

The results are sound.

**Other Comments Or Suggestions:**

typo(?): how is $||W_Q,W_K,W_V||_F$ defined in lemma 4.16?

typo in line 186: samples

the title "experiments" is line 427 seems redundant.

In the related work section, it seems appropriate to include comparisons with previous studies and clarify how your results differ from them.

**Other Strengths And Weaknesses:**

Strengths:

The considered model is quite sophisticated and close to what can be used in practice. The method is rigorous with few assumptions on the model and also considers inter-token dependencies via $\Phi$-mixing. The authors use an interesting combination of known methods to bound the Rademacher complexity of both the representation learner and the token selection part of the transformer through bounding the covering number of networks with bounded Frobenius norm.

Weaknesses:

The paper doesn't discuss in detail about the new/challenging steps in the proof compared to previous works. the authors mention they're the first to consider mask matrix but it is not discussed how it affects the approach or the final results. Considering NTP paradigm and obtaining bounds decreasing in terms of sequence length are interesting contributions of this work but the paper lacks a high-level discussion on the challenges.

It seems Lemma 4.15 which bounds the covering number of Frobenius-norm bounded weights (by $a$) and is only logarithmic in $\epsilon,a$ and data norm is essential for obtaining improved results in terms of number of layers. This is comparable to Lemma 9 in (Deng et al.2024) that obtains a bound on the covering number of $\ell_{q,s}$-norm bounded weights and has a polynomial dependence on $a$ and data norm but with better dependence on $d$. Can the authors comment on the worse dependence in $d$ in their result compared to Deng et al.2014?


The effect of $disc(U)$ is not discussed in detail in the paper and it's not really clear how it quantitatively affects the bound, except that it's zero if the distributions in $U$ are equal.

**Questions For Authors:**

please see sections above.

**Relation To Broader Scientific Literature:**

The work presents a theoretical upper bound for the transformer DOM. This is closely related to Zhang et al. 2023 and Deng et al. 2024 which also utilizes Rademacher complexity. Compared to closely related work of (Deng et al. 2024) which considered $\mathcal{H}$ as fixed (cf. summary section), here the Rademacher complexity is computed over both $\mathcal{H}$ and $\mathcal{G}$. Also, the dependency on the number of layers in the resulting bound is improved.

**Theoretical Claims:**

No.

---

> ### Author Rebuttal · Authors · 2025-03-31
>
> We are delighted to receive your recognition and support for our work, and we appreciate your careful attention to details we might have overlooked. We have made corrections to the relevant sections of the paper based on your suggestions. Additionally, we plan to release the code upon acceptance of the paper. Below are our responses to your questions, which we hope will address any concerns:
>
> **A1:** Our work primarily focuses on the proof details, and we initially lacked a thorough technical comparison with previous works. We have now enhanced these sections. Compared to earlier works such as [1,2], our main technical challenges and innovations include:
>
> (1) Considering both the independence between sequences and the dependency between tokens within sequences. Unlike previous works that only considered independent and identically distributed token sequences, we introduced the $\varphi$-mixing tool for analysis. This increased the complexity of error decomposition but also enhanced the interpretability in the NTP pre-training scenario, specifically the bound's decrease with sequence length.
>
> (2) We introduced the Rademacher complexity of composite function spaces and proposed its decomposition theorem (Proposition 1), enabling separate analysis of different function spaces. This greatly simplifies the generalization analysis when changing DOMs task heads or transferring pre-trained DOMs to different downstream tasks.
>
> (3) We performed a detailed analysis of the covering number for masked-self-attention-based decoder-only models. Our approach maintains a high structural consistency with the original Transformer [3] and analyzes under the F-norm, which was not discussed in previous works. The main challenge in considering the mask matrix is defining the upper bound of the attention matrix's norm, as seen in Lemma C.8. This results in an additional $\sqrt{\ln(m)}$ factor compared to analyses that do not consider the mask matrix's norm upper bound.
>
>
> **A2:** Thank you for your question. We compare our work in two aspects, denoting the input data matrix as $\mathbf{X}_{[N]} \in \mathbb{R}^{Nmd}$: (1) Regarding the dependence on $\epsilon$ and data norm, their bound is $\sqrt{\frac{||\mathbf{X}\_{[N]}||^2}{N}}$, while ours is $\sqrt{\frac{\ln(||\mathbf{X}\_{[N]}||\_{F}/\sqrt{Nmd})}{Nm}}$. Given that the order of $||\mathbf{X}\_{[N]}||$ is $\sqrt{Nmd}$, their bound has a greater dependence on the data norm, whereas our bound almost eliminates this dependence. (2) Regarding the dependence on model dimension $d$, since the model parameter count $\Theta=\mathcal{O}(d^2)$, their bound has a logarithmic dependence on $\Theta$ due to its logarithmic dependence on $d$. Our bound has a polynomial dependence on $\Theta$. According to the Scaling Laws for language models [4], the dataset size should grow sub-linearly with the model parameter count, making our bound more consistent with the Scaling Laws.
>
> **A3:** In our work, `disc(U)` primarily measures the overall quality of pre-training data. We assume all sequences are generated by a $\varphi$-mixing process, defined as an infinitely long sequence, implying that most sequences may originate from the same mixing process. On one hand, two different sequences might be extracted from the same corpus or even the same article. On the other hand, due to the nature of human language, even seemingly unrelated sentences like "Large language models benefit humanity" and "The weather is nice today" can appear in the same context when read together. This indicates that any high-quality sentence conforming to human language rules can be interconnected through language. Conversely, sequences like gibberish, grammatical errors, unclear expressions, or incorrect knowledge struggle to establish connections with high-quality human language. This underscores the importance of data cleaning quality [5]. Fewer low-quality sequences in the pre-training data result in a smaller `disc(U)`, thereby enhancing the model's generalization performance.
>
> **A4：** Regarding Lemma 4.16, we acknowledge that the notation $||W_Q,W_K,W_V||\_F \le B$ was indeed a space-saving simplification in the main text. The complete formulation, specifying $||W_Q||\_F ≤ B$, $||W_K||\_F ≤ B$, and $||W_V||\_F ≤ B$ individually, is provided in Lemma C.10. We have now added explicit clarification of this point in our paper. Additionally, we confirm that the issues raised regarding line 186 and line 427 have been addressed. We thank the reviewer again for bringing these important details to our attention!
>
> **Reference:**
>
> [1]Edelman, B.(2022) Inductive biases and variable creation in self-attention mechanisms.
>
> [2]Deng, Y.(2024) On the generalization ability of unsupervised pertaining.
>
> [3]Vaswani,A.(2017) Attention is all you need.
>
> [4]Open AI.(2020) Scaling Laws for Neural Language Models.
>
> [5]Penedo, M.(2023) The Refined Web Dataset for Falcon LLM: Outperforming Curated Corpora with Web Data, and Web Data Only.

---

### Official Review · Reviewer_cRs1 · 2025-03-26

**Overall Recommendation:** 3

**Summary:**

This paper investigates the generalization properties of Next-Token Prediction (NTP) pre-training. The derived generalization bounds highlight the influence of key model parameters, such as the number of token sequences and the maximum sequence length. Specifically, the contributions include: establishing a Rademacher complexity bound for the excess risk, deriving covering number bounds for the function space of transformer-decoder models, and providing a generalization bound for decoder-only models (DOMs) in the context of NTP pre-training.

## update after rebuttal
The authors have addressed my questions, and I will maintain my current score.

**Claims And Evidence:**

The claims in this paper—particularly the generalization bounds—are well-supported and convincing.

**Essential References Not Discussed:**

The related works discussion is sufficient.

**Experimental Designs Or Analyses:**

This is a theoretical work, the experiments are valid to verify the theoretical results.

**Methods And Evaluation Criteria:**

This is a theoretical work, and while the simulation settings are relatively simple, they help reinforce the theoretical findings.

**Other Comments Or Suggestions:**

N/A

**Other Strengths And Weaknesses:**

As mentioned earlier, my main concern is the lack of discussion on how the proofs in this paper relate to prior work. For instance, Assumption 4.14 also appears in Edelman et al. (2022) and Deng et al. (2024). However, it is unclear how the proofs in this paper build upon or differ from those studies, and what specific challenges the authors faced. As a result, the novelty of the proof remains unclear. Additionally, Assumption 4.2 appears to play an important role, but it is unclear how this assumption contributes to the development of the proofs.

**Questions For Authors:**

See comments from Other Strengths And Weaknesses.

**Relation To Broader Scientific Literature:**

This work provides a thorough discussion of related literature and compares its results with prior studies in Section 2. The authors also strengthen their assumptions by referencing established works. However, my main concern is the lack of discussion on how their proofs relate to those in previous research. For example, Assumption 4.14 was also used in Edelman et al. (2022) and Deng et al. (2024). While the authors compare their bounds with those in these two papers, they offer limited insight into how their proof techniques differ from or build upon those earlier works.

**Theoretical Claims:**

The theorems and proofs provided in this paper are convincing.

---

> ### Author Rebuttal · Authors · 2025-03-31
>
> Thank you for your recognition and support of our work. Below, we provide a detailed comparative analysis of our work with the two most relevant previous studies, [1] and [2]. We sincerely hope this will address your concerns. All of the following content has been added to the discussion section of our paper:
>
> Our approach shares the overall methodology and thought process with [1] and [2], where we first decompose the excess risk error to derive an upper bound on the Rademacher complexity, and then obtain a more refined generalization error bound by analyzing the model capacity, i.e., the covering number. However, there are several key differences:
>
> 1. **Task Context:** We focus on NTP pre-training, using $\varphi$-mixing to model token sequences to capture dependencies between tokens within a sequence, whereas [1] and [2] only consider independence between sequences.
>
> 2. **Rademacher Complexity:** Since DOM can be viewed as composed of two parts, its function space is considered as a composite of two function spaces. Therefore, we introduced the Rademacher complexity of composite function spaces and derived the corresponding decomposition theorem, Proposition 1. Compared to the direct analysis of a single function space in [1] and [2], our method facilitates the analysis of pre-training generalization when changing DOM task heads and makes it easier to transfer to different downstream tasks.
>
> 3. **Covering Number Analysis:** Like [1] and [2], we analyzed the covering number of Transformers, which is why we share Assumption 4.14. Similar assumptions are common in covering number analyses, such as in [3] and [4]. However, [1] and [2] analyze the covering number of encoder-only models under the spectral norm, while we analyze decoder-only models under the F-norm. Our challenge includes considering the masked self-attention mechanism, with the main impact of the mask matrix detailed in Lemma C.8.
>
> 4. **Final Results:** [1] and [2] show polynomial dependence on data norms and parameter bounds, and logarithmic dependence on the number of model parameters, which is the opposite of our findings. Our results nearly eliminate dependence on data norms and show polynomial dependence on model parameters, aligning more closely with the Scaling Laws for large models [5], which suggest linear growth of data size with model parameter count. This is mainly because we cleverly applied the convexity and concavity of functions, as stated in Lemma C.6, to Lemma C.5, with detailed applications in Appendix D.
>
> Additionally, Assumption 4.2 is crucial for considering dependencies between tokens within a sequence. Based on Assumption 4.2, we can use Lemma B.4 to analyze the model's generalization ability on a single token sequence.
>
> **Reference:**
>
> [1]Edelman, B.(2022) Inductive biases and variable creation in self-attention mechanisms.
>
> [2]Deng, Y.(2024) On the generalization ability of unsupervised pertaining.
>
> [3]Bartlett, P.(2017)  Spectrallynormalized margin bounds for neural networks.
>
> [4]Lin, S. and Zhang, J. Generalization bounds for convolutional neural networks.
>
> [5]Open AI.(2020) Scaling Laws for Neural Language Models.

---

### Decision · Program_Chairs · 2025-05-01

**Decision:**

Accept (poster)

**Comment:**

The authors establish a Rademacher complexity bound for the excess risk of NTP training and derive covering number bounds for the function space of transformer-decoder models, thus yielding a generalization bound for decoder models. Within a sequence, they assume that tokens are dependent but follow certain mixing conditions.

Three reviewers remain positive about the paper after raising several technical questions which have been successfully addressed by the authors. The fourth reviewer raised concerns regarding suboptimality of the bound, particularly with respect to previous works, in that the bound scales as 1/\sqrt{m} where m is the size of the sequence and thus won't go down as the number of sequences N increases. The authors have in my opinion successfully clarified that results in referenced previous works are of similar nature and have also clarified that compared to previous works they here tackle the challenge of sequence dependencies (albeit using standard assumptions on mixing conditions).

Overall, I suggest that the authors take care in revising their manuscript according to the feedback. Please include all additional explanations given during rebuttal to make the results more interpretable and clarify the technical contributions compared to previous works. Edge cases such as those asked by Rev. astS on what happens when all \phi_i are equal are useful and should be included in the revision. Additional discussions regarding the experiments as included in the rebuttal should also be added.

That said, study of pretraining generalization properties is an understudied topic and this paper makes a meaningful step in this direction. Given this and the overall positive feedback, I recommend acceptance.